# SubDyve: Subgraph-Driven Dynamic Propagation for Virtual Screening Enhancement Controlling False Positive

**Jungseob Yi[1]    Seoyoung Choi[2]    Sun Kim[1,2,3,4]    Sangseon Lee[5]**

[1]Interdisciplinary Program in Artificial Intelligence, Seoul National University
[2]Department of Computer Science and Engineering, Seoul National University
[3]Interdisciplinary Program in Bioinformatics, Seoul National University
[4]AIGENDRUG Co., Ltd., Seoul
[5]Department of Artificial Intelligence, Inha University

## Abstract

Virtual screening (VS) aims to identify bioactive compounds from vast chemical libraries, but remains difficult in low-label regimes where only a few actives are known. Existing methods largely rely on general-purpose molecular fingerprints and overlook class-discriminative substructures critical to bioactivity. Moreover, they consider molecules independently, limiting effectiveness in low-label regimes. We introduce SubDyve, a network-based VS framework that constructs a subgraph-aware similarity network and propagates activity signals from a small known actives. When few active compounds are available, SubDyve performs iterative seed refinement, incrementally promoting new candidates based on local false discovery rate. This strategy expands the seed set with promising candidates while controlling false positives from topological bias and overexpansion. We evaluate SubDyve on ten DUD-E targets under zero-shot conditions and on the CDK7 target with a 10-million-compound ZINC dataset. SubDyve consistently outperforms existing fingerprint or embedding-based approaches, achieving margins of up to +34.0 on the BEDROC and +24.6 on the $EF_{1\%}$ metric.

## 1 Introduction

The chemical space in drug discovery is vast, comprising more than $10^{60}$ synthetically accessible drug-like molecules (Virshup et al., 2013). Exhaustive exploration is infeasible, making virtual screening (VS) a key tool for identifying promising compounds small enough for experimental validation. However, in early stage discovery, most protein targets lack substantial ligand data; researchers often start with only a few known active molecules (Deng et al., 2024; Jiang et al., 2024; Chen et al., 2024; Scott et al., 2016). The central challenge in such low-data settings is to retrieve additional actives from billions of candidates, given only a target protein and sparse activity labels.

Deep learning approaches to virtual screening fall into two main categories: supervised models and foundation models. The first trains on large, balanced datasets using graph neural networks or 3D molecular representations, but requires extensive labeled data and often overfits in low-data settings. The second leverages foundation models (FMs) pre-trained on large-scale unlabeled molecular corpora to support inference with minimal supervision. Representative FMs include ChemBERTa (Ahmad et al., 2022), MolBERT (Fabian et al., 2020), MoLFormer (Ross et al., 2022), GROVER (Rong et al., 2020), and AMOLE (Lee et al., 2024), which support zero-shot VS. Protein-language-model pipelines (Lam et al., 2024) show similar promise in structure-free contexts. However, FM-based methods screen compounds independently, failing to capture molecular dependencies.

An orthogonal line of approach addresses these limitations with network-based label-efficient learning (Yi et al., 2023; Saha et al., 2024; Ma et al., 2024). Among these, network propagation (NP) has emerged as a promising and effective strategy. NP treats known actives as seed nodes in a molecular graph, diffusing influence across networks to prioritize candidates based on global connectivity to the seed set. This framework captures higher-order molecular associations and naturally supports generalization from few labeled molecules.

Despite its promise, two critical limitations of NP remain to be resolved. First, VS tasks often hinge on substructural variations between closely related molecules (Ottanà et al., 2021; Stumpfe et al., 2019). Yet standard NP relies on similarity graphs uses general-purpose fingerprints (e.g., ECFP), which fail to encode fine-grained subgraph features that distinguish actives from inactive molecules (Yi et al., 2023), often blurring critical activity-relevant distinctions. Second, NP inherits the topological bias of the underlying graph: nodes in dense clusters may be ranked highly due to connectivity alone, inflating false positives (FP), particularly when the seed set is small (Picart-Armada et al., 2021).

To address these limitations, we propose SubDyve, a graph-based virtual screening framework for label-efficient compound prioritization. Rather than relying on generic molecular fingerprints, SubDyve builds a subgraph fingerprint graph using class-discriminative substructures mined via supervised subgraph selection (Lim et al., 2023). It then performs iterative seed refinement guided by local false discovery rate (LFDR) estimates (Efron, 2005), expanding high-confidence compounds as new seeds while controlling false positives. This process is integrated into a joint learning framework that trains a graph neural network with objectives for classification, ranking, and contrastive embedding. By combining subgraph-aware graph construction with uncertainty-calibrated propagation, SubDyve improves precision and generalization under sparse supervision. We evaluate SubDyve on the DUD-E benchmark and a 10M-compound ZINC/PubChem dataset for CDK7 target, where it achieves strong early enrichment using substantially fewer labels than deep learning and mining-based baselines. Our contributions are as follows:

- We demonstrate that SubDyve achieves state-of-the-art performance on public and large-scale datasets under severe label constraints.
- We propose a subgraph fingerprint graph construction method that identifies class-discriminative subgraphs, preserving subtle activity-defining features that are overlooked by conventional fingerprints.
- We introduce an LFDR-based seed refinement mechanism that overcomes graph-induced bias and enhances screening specificity while controlling false positive rates.

## 2 Related Work

**Representation-Centric Virtual Screening** Traditional VS methods use fixed fingerprints (e.g., ECFP, MACCS) or 3D alignments with shallow classifiers, but often miss substructural patterns critical to bioactivity. Recent deep learning approaches embed molecular structures into task-optimized latent spaces. PharmacoMatch (Rose et al., 2025) frames pharmacophore screening as neural subgraph matching over 3D features. PSICHIC (Koh et al., 2024) and BIND (Lam et al., 2024) integrate protein sequence embeddings with ligand graphs. Large-scale pretrained encoders like ChemBERTa (Ahmad et al., 2022), MoLFormer (Ross et al., 2022), and AMOLE (Lee et al., 2024) reduce label demands via foundation model generalization. However, these methods treat compounds independently, ignoring higher-order molecular dependencies.

**Label-Efficient Network Propagation** Network propagation (NP) enables label-efficient VS by diffusing activity signals over molecular similarity graphs. Yi et al. (Yi et al., 2023) build target-aware graphs from multiple similarity measures to rank candidates from known actives. GRAB (Yoo et al., 2021) applies positive unlabeled (PU) learning to infer soft labels from few positives, demonstrating robustness in low-supervision settings. While NP-based methods capture higher-order dependencies, they often rely on generic fingerprints (e.g., ECFP) that overlook discriminative substructures and suffer from topological bias, inflating false positives under sparse or uneven labeling (Picart-Armada et al., 2021; Hill et al., 2019).

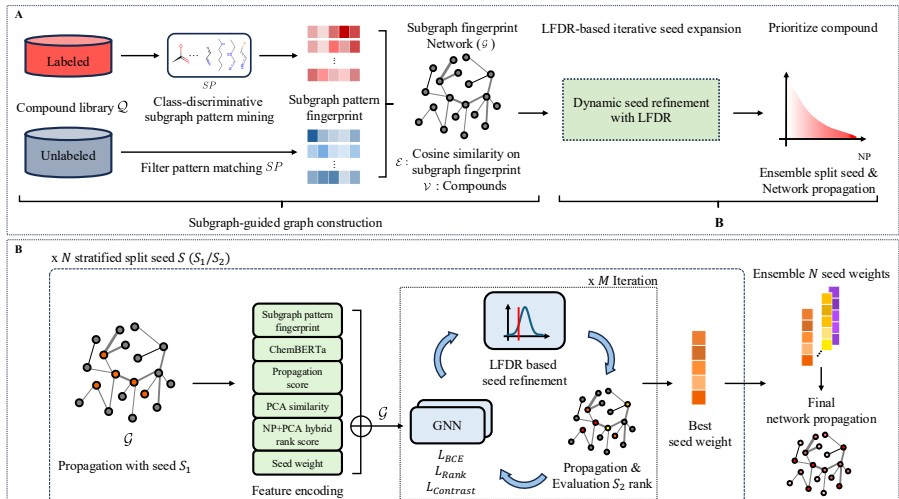

Figure 1: Architecture of SubDyve Framework. (A) Overall process of SubDyve. Consists of subgraph-similariy network construction, dynamic seed refinement with LFDR, and prioritization. (B) Dynamic seed refinement with LFDR: seed weights are iteratively updated within each stratified split and aggregated for final prioritization.

**Substructure-Aware Similarity Graphs**   Recent work enhances molecular graphs with substructure-aware representations to capture subtle activity-relevant patterns. Supervised Subgraph Mining (SSM) (Lim et al., 2023) identifies class-specific motifs that improve prediction and reveal mechanistic effects like toxicity. ACANet (Shen et al., 2024) applies attention over local fragments to detect activity cliffs. While effective in property prediction, such methods remain underexplored in virtual screening, where subgraph-aware graphs could better resolve activity-specific features.

## 3   METHODOLOGY

In this section, we present the architecture of SubDyve, a virtual screening framework for settings with few known actives. SubDyve first constructs a subgraph fingerprint network using class-discriminative subgraph patterns (Figure 1A). Based on this network, it performs dynamic seed refinement guided by LFDR to iteratively update seed weights (Figure 1B). To ensure robustness, refinement is repeated across $N$ settings, and the ensembled seed weights are used for a final network propagation to prioritize unlabeled compounds.

### 3.1   PROBLEM FORMULATION

We define virtual screening as the problem of ranking a large set of unlabeled candidate compounds, especially under a low-label regime. Let $\mathcal{Q}$ be the set of candidate molecules, and $\mathcal{C} \subset \mathcal{Q}$ the subset of known actives against a target protein $p$. The goal is to assign a relevance score $r(q)$ to each $q \in \mathcal{Q}$ such that compounds in $\mathcal{C}$ are ranked higher than inactives. In low-label settings, a small subset $\mathcal{S}_{train}, \mathcal{S}_{test} \subset \mathcal{C}$ of seed actives is available ($\mathcal{S}_{train} \cap \mathcal{S}_{test} = \emptyset$). We assume access to a compound similarity graph $\mathcal{G} = (\mathcal{V}, \mathcal{E})$ over $\mathcal{Q}$, where each node $v \in \mathcal{V}$ is a compound and each edge $(i, j) \in \mathcal{E}$ encodes structural similarity. The task is to propagate activity signals from $\mathcal{S}_{train}$ over $\mathcal{G}$ and assign relevance scores $r(q)$ to all $q \in \mathcal{Q}$, prioritizing $\mathcal{S}_{test}$.

### 3.2   SUBGRAPH FINGERPRINT NETWORK CONSTRUCTION

We first mine class-discriminative subgraph patterns $\mathcal{SP}$ from the labeled seed set $\mathcal{S}_{train}$ using the SSM algorithm (Lim et al., 2023) with curated negative molecules. Each molecule is then encoded as a $d$-dimensional *subgraph pattern fingerprint*, where each dimension reflects the frequency of a discriminative subgraph combination (DiSC) (see Appendix B.2.1-B.2.2).

We filter the candidate set $\mathcal{Q}$ to retain only compounds that match at least one subgraph in $\mathcal{SP}$, forming a reduced set $\mathcal{Q}'$. A subgraph fingerprint graph $\mathcal{G}$ is then constructed over $\mathcal{Q}'$ using pairwise cosine similarity between subgraph pattern fingerprints (Appendix B.2.3). This graph serves as the foundation for network propagation and compound ranking.

## 3.3 Dynamic Seed Refinement with LFDR Estimation

Using $\mathcal{S}_{train}$ as seed nodes, we perform initial network propagation over $\mathcal{G}$ to assign soft relevance scores to all compounds. While this provides a baseline prioritization, its effectiveness is limited by the small size of $\mathcal{S}_{train}$. Signals tend to diffuse broadly or become biased toward topologically dense regions, resulting in reduced specificity and inflated false positives.

To address this, SubDyve introduces a dynamic seed refinement procedure that iteratively improves the seed set using GNN-based inference and local false discovery rate (LFDR) estimation. To enable robust screening, we stratify $\mathcal{S}_{train}$ into disjoint subsets $\mathcal{S}_1$ and $\mathcal{S}_2$, where $\mathcal{S}_1$ initiates the initial network propagation, and $\mathcal{S}_2$ guides seed refinement via iterative loss updates in both the GNN and propagation modules. This mechanism enables confident expansion of the supervision signal while suppressing propagation-induced errors.

### 3.3.1 Feature Encoding for GNN

Before refinement, we compute feature vectors for all compounds using SMILES (Weininger, 1988) and graph-derived descriptors. Each compound $i \in \mathcal{Q}'$ is encoded as:

$$\mathbf{x}_i = [w_i, n_i^{\mathrm{NP}}, \mathbf{f}_i^{\mathrm{FP}}, s_i^{\mathrm{PCA}}, h_i^{\mathrm{hyb}}, \mathbf{e}_i^{\mathrm{PT\text{–}CB}}], \tag{1}$$

where $w_i$ denotes a weight of seed $\mathcal{S}_1$ for network propagation. $n_i^{\mathrm{NP}}$ is a network propagation score using $w_i$. $s_i^{\mathrm{PCA}}$ is a RBF similarity to seed $\mathcal{S}_1$ in PCA latent space, and $h_i^{\mathrm{hyb}}$ is a hybrid ranking computed as the average of the PCA and NP ranks, $(\mathrm{rank}(s_i^{\mathrm{PCA}}) + \mathrm{rank}(n_i^{\mathrm{NP}}))/2$. $\mathbf{e}_i^{\mathrm{PT\text{–}CB}}$ and $\mathbf{f}_i^{\mathrm{FP}}$ encode semantic and substructural properties, respectively. Details of the feature encoding are described in Appendix B.3.

### 3.3.2 Iterative Seed Refinement with LFDR Estimation

Building on the initial propagation with $\mathcal{S}_1$, SubDyve performs iterative seed refinement over hyperparameter $M$ iterations. In each iteration, the model is trained to recover held-out actives in $\mathcal{S}_2$ through three steps: (1) GNN training with a composite loss, (2) LFDR-guided seed refinement, and (3) network propagation and evaluation.

**(1) GNN Training.** A graph neural network processes the subgraph fingerprint graph $\mathcal{G}$ to produce logits $\hat{l}_i$ and embeddings $z_i$ for compound $i$. The model is trained using a composite loss that combines three objectives:

$$\begin{aligned}\mathcal{L}_{\mathrm{total}} = (1 - \lambda_{\mathrm{rank}}) \cdot \mathcal{L}_{\mathrm{BCE}} + \lambda_{\mathrm{rank}} \cdot \mathcal{L}_{\mathrm{RankNet}} \\ + \lambda_{\mathrm{contrast}} \cdot \mathcal{L}_{\mathrm{Contrast}}.\end{aligned} \tag{2}$$

Here, $\mathcal{L}_{\mathrm{BCE}}$ is a binary cross-entropy loss that adjusts for class imbalance by weighting active compounds more heavily according to their low prevalence. $\mathcal{L}_{\mathrm{RankNet}}$ is a pairwise ranking loss that encourages known actives in the held-out set $\mathcal{S}_2$ to be ranked above unlabeled candidates. $\mathcal{L}_{\mathrm{Contrast}}$ is a contrastive loss on the held-out set $\mathcal{S}_2$, where each compound pairs with its most similar member, treating the rest as negatives. The coefficients $\lambda_{\mathrm{rank}}$ and $\lambda_{\mathrm{contrast}}$ are hyperparameters controlling the contribution of each loss term. Full loss definitions and model optimization are described in Appendix B.4.

**(2) LFDR-Guided Seed Refinement.** Given logits $\hat{l}_i$, we compute a standardized score:

$$z_i = \frac{\hat{l}_i - \mu}{\sigma}, \quad q_i = \mathrm{LFDR}(z_i), \tag{3}$$

with $\mu, \sigma$ estimated from all logits in $\mathcal{Q}'$. Details of LFDR algorithm is described in Algorithm 3 at Appendix B. The seed set $\mathcal{S}_{t+1}$ and its weights are then updated as

$$\mathcal{S}_{t+1} = \{i \in \mathcal{Q}' : q_i < \tau_{\mathrm{FDR}}\}, \quad w_i^{(t+1)} = w_i^{(t)} + \beta(\sigma(z_i) - \pi_0), \ i \in \mathcal{S}_{t+1}, \tag{4}$$

so that only compounds satisfying $q_i < \tau_{\mathrm{FDR}}$ remain as seeds while others are removed, where $\sigma(\cdot)$ is the sigmoid function, $\pi_0$ the prior null probability, and $\beta$ controls the update rate. This procedure ensures a provable upper bound on the false discovery rate (Efron, 2005), as detailed in Proposition 1.

**Proposition 1.** *Let $Z_1, \ldots, Z_m$ follow the two–group mixture $f(z) = \pi_0 f_0(z) + \pi_1 f_1(z)$ and define the selection rule*

$$\mathcal{R}_\alpha = \{\, i : \mathrm{lfdr}(Z_i) \leq \alpha \,\}, \qquad 0 < \alpha < 1.$$

*Denote $R_\alpha = |\mathcal{R}_\alpha|$ and $V_\alpha = \sum_{i=1}^{m} I\{i \in \mathcal{R}_\alpha\} H_i$. Then*

$$\mathrm{mFDR}(\mathcal{R}_\alpha) = \frac{\mathbb{E}[V_\alpha]}{\mathbb{E}[R_\alpha]} \leq \alpha, \tag{1}$$

$$\mathrm{FDR}(\mathcal{R}_\alpha) = \mathbb{E}\!\left[\frac{V_\alpha}{R_\alpha \vee 1}\right] \leq \alpha. \tag{2}$$

*Proof is provided in Appendix A.*

**(3) Network Propagation and Evaluation.** We apply network propagation with the updated seed weights and evaluate performance on $\mathcal{S}_2$ using enrichment factor at early ranking thresholds. While $\mathcal{S}_2$ contributes to $\mathcal{L}_{\mathrm{total}}$, early stopping relies solely on intermediate enrichment scores, preventing bias from direct training objectives. If enrichment improves, the iteration continues; otherwise, early stopping is triggered. Among all iterations, we retain the seed weights that yield the highest performance on $\mathcal{S}_2$ for final ensemble aggregation.

### 3.4 Final Aggregation and Prioritization

To improve robustness under limited supervision, we perform $N$ stratified splits of the initial seed set $\mathcal{S}_{\mathrm{train}}$ into disjoint subsets $\mathcal{S}_1$ and $\mathcal{S}_2$. From each split, we retain the best-performing seed weights on $\mathcal{S}_2$ and aggregate them across all $N$ splits using element-wise max pooling to construct the final ensemble seed vector. Using this ensembled vector, we perform a final round of network propagation over $\mathcal{G}$ to score all compounds in $\mathcal{Q}'$, producing the final compound ranking used for virtual screening evaluation.

## 4 Experiments

In this section, we evaluate the SubDyve framework on a set of virtual screening tasks, including 10 benchmark targets from the DUD-E dataset and a real-world case study curated using ZINC20 (Irwin et al., 2020) compounds and PubChem (Kim et al., 2023) active compounds. This evaluation highlights the applicability of the framework in both standardized benchmarks and real-world drug discovery environments. Detailed experimental setup and hyperparameter configurations are provided in Appendix C. We present comprehensive comparisons with state-of-the-art methods, alongside extensive ablation studies, interpretation analyzes, and case studies to highlight the effectiveness and robustness of each component.

### 4.1 Virtual Screening Performance

#### 4.1.1 Zero-Shot Screening on DUD-E Targets

**Task.** The goal of this evaluation is to assess whether SubDyve can effectively generalize in zero-shot virtual screening by prioritizing active compounds without target-specific training.

**Dataset & Evaluation.** We follow the evaluation protocol of PharmacoMatch (Rose et al., 2025), using the same 10 protein targets from the DUD-E benchmark (Mysinger et al., 2012). Since SubDyve requires a small number of active molecules to construct subgraph newtork and initiate network propagation, directly using actives from the test targets would violate the zero-shot assumption and compromise fairness. To ensure fair comparison, we curated related proteins from PubChem using MMseqs2 (Steinegger & Söding, 2017) tool with similarity thresholds of 0.9 to filter out proteins in PubChem. In addition, we additionally

Table 1: Performance comparison of SubDyve (threshold=0.9) and baselines on the ten DUD-E targets. The top results are shown in **bold**, and the second-best are underlined, respectively. Confidence intervals are reported with 100 bootstrap resamples (DiCiccio & Efron, 1996). The complete results, including all baselines and metrics are at Appendix F.1.1.

| Protein Target | SubDyve (Ours) | | PharmacoMatch (Rose et al., 2025) | | CDPKit (Seidel, 2024) | | DrugCLIP (Gao et al., 2023) | | MoLFormer (Ross et al., 2022) | | AutoDock Vina (Eberhardt et al., 2021) | |
|---|---|---|---|---|---|---|---|---|---|---|---|---|
| | BEDROC | $EF_{1\%}$ | BEDROC | $EF_{1\%}$ | BEDROC | $EF_{1\%}$ | BEDROC | $EF_{1\%}$ | BEDROC | $EF_{1\%}$ | BEDROC | $EF_{1\%}$ |
| ACES | **86±2** | **57.0±2.4** | 18±1 | 8.4±1.4 | 16±2 | 5.5±1.3 | 52±2 | 32.4±1.7 | 24±2 | 8.3±0.7 | 33±1 | 13.87±0.5 |
| ADA | 83±4 | 50.6±5.3 | 44±4 | 16.7±4.1 | 82±3 | 53.6±4.3 | 82±3 | **60.2±5.3** | 72±1 | 48.3±0.9 | 7±2 | 1.05±1.7 |
| ANDR | **72±2** | **37.1±2.1** | 33±2 | 15.8±1.9 | 26±2 | 12.6±2.1 | 64±3 | 34.3±2.4 | 9±1 | 3.0±0.1 | 34±1 | 18.41±0.6 |
| EGFR | **86±2** | **60.0±2.3** | 11±1 | 3.1±0.7 | 26±2 | 12.2±1.6 | 40±2 | 28.7±2.1 | 75±2 | 48.1±2.8 | 14±1 | 3.68±0.7 |
| FA10 | 58±2 | 46.8±1.7 | 1±1 | 0.2±0.2 | 6±1 | 0.0±0.0 | **86±1** | **51.2±1.8** | 66±0 | 36.7±0.4 | 41±1 | 15.77±0.8 |
| KIT | 44±3 | 13.8±2.6 | 4±1 | 0.0±0.0 | 9±2 | 1.1±0.8 | 10±2 | 5.2±1.7 | **66±1** | **36.8±0.9** | 18±2 | 2.97±1.9 |
| PLK1 | **85±3** | **51.7±4.0** | 9±2 | 1.5±1.3 | 39±3 | 5.7±2.3 | 66±4 | 45.0±4.0 | 69±0 | 35.2±4.0 | 13±1 | 1.83±0.3 |
| SRC | **61±2** | **35.0±1.8** | 27±1 | 6.0±1.0 | 28±1 | 11.1±1.2 | 16±1 | 8.1±1.3 | 48±1 | 21.5±1.5 | 13±1 | 4.00±0.5 |
| THRB | 61±2 | 36.6±2.0 | 22±1 | 5.9±1.0 | 35±2 | 11.8±1.5 | **83±1** | **46.9±1.7** | 6±1 | 1.2±0.1 | 25±1 | 4.31±1.0 |
| UROK | 37±3 | 25.6±2.4 | 4±1 | 0.6±0.7 | 55±3 | 24.5±2.8 | **73±3** | **48.1±3.1** | 36±2 | 10.0±1.5 | 28±1 | 7.90±0.7 |
| Avg. rank | 1.6 | 1.6 | 4.5 | 4.4 | 3.6 | 3.7 | 2.4 | 2.0 | 3.0 | 3.3 | 4.0 | 4.0 |

considered a stricter configuration with threshold of 0.5, which forces seeds to be drawn from more distant homologous proteins. Bioactive compounds associated with these PubChem proteins were then used as seed molecules. Detailed filtering criteria and dataset statistics are provided in Appendix D.1. All baseline models were evaluated using the best hyperparameter settings reported in their respective papers to ensure a fair comparison. For evaluation, we measure early recognition metrics: BEDROC ($\alpha = 20$), $EF_{N\%}$, as well as AUROC for completeness. Full metric panels are in Appendix C.3.

**Performance.** Table 1 reports BEDROC ($\alpha = 20$) and $EF_{1\%}$ scores, highlighting Sub-Dyve's early recognition performance on threshold 0.9 setting. Full results, including AUROC and per-target metrics, are provided in Appendix F.1.1. SubDyve achieves the highest average rank across all metrics; 1.6 for BEDROC and 1.6 for $EF_{1\%}$. For example, on EGFR and PLK1, two pharmacologically important targets known for their conformational flexibility and multiple ligand binding modes (Zhao et al., 2018; Murugan et al., 2011), SubDyve achieved BEDROC scores of 86 and 85, substantially higher than those of MoLFormer (75 and 69). Even for structurally challenging targets such as ACES, which features a deep, narrow binding gorge that imposes strict shape complementarity and limits ligand accessibility (Mishra & Basu, 2013), SubDyve yields meaningful enrichment ($EF_{1\%} = 57.0$), substantially higher than DrugCLIP (32.4) and AutoDock Vina (13.87). These results highlight SubDyve's robustness across diverse targets and its strength in early active identification.

To examine whether SubDyve relies on close homologs when constructing seed sets, we repeated the experiment using a stricter MMseqs2 identity threshold of 0.5. Due to the increased number of homolog candidates when lowering the MMseqs2 identity threshold from 0.9 to 0.5, we conducted this stricter analysis on a subset of three targets (EGFR, PLK1, SRC). Despite receiving less informative and more diverse seed molecules, SubDyve maintained strong early-recognition performance and remained better or competitive with all baselines on both BEDROC and $EF_{1\%}$ (Appendix F.1.2). These findings indicate that SubDyve does not depend on highly similar homologs and remains effective even when only distant protein–ligand annotations are available.

**Multi-Target Screening Analysis.** To evaluate SubDyve beyond the single-target setting, we conducted a multi-target analysis on three tyrosine kinases (EGFR, KIT, SRC), which share related molecular functions and signaling roles. Rather than constructing separate subgraph networks, we identified homologous PubChem proteins common to the three targets and performed a single subgraph-mining step using their associated bioactive compounds. The resulting unified subgraph fingerprint network was reused across all evaluations without additional target-specific preprocessing. Details are summarized in Appendix F.1.3.

We assessed SubDyve under four multi-target activity definitions (shared actives across all three targets or each target pair). SubDyve maintained strong early-recognition performance. For example, on the EGFR–SRC pair, SubDyve achieved a BEDROC of 84.11 and $EF_{1\%}$ of 41.21, outperforming DrugCLIP (BEDROC 13.14, EF1% 7.26). Even in the three-target setting (EGFR–KIT–SRC), SubDyve reached a BEDROC of 69.52 compared to DrugCLIP's 4.55. Full results are provided in Appendix Table 16. These results suggest that for biologically related targets with shared ligand annotations, SubDyve's extracted structural patterns transfer effectively, enabling the preprocessing cost of subgraph mining to be amortized over multiple screening tasks.

Table 2: Performance comparison of SubDyve and baselines on the PU dataset. The top results are shown in **bold**, and the second-best are underlined, respectively. The complete results, including all baselines are at Appendix F.2.

| Method | BEDROC (%) | EF | | | | |
|---|---|---|---|---|---|---|
| | | 0.5% | 1% | 3% | 5% | 10% |
| **Deep learning-based** | | | | | | |
| BIND (Lam et al., 2024) | - | - | - | - | - | 0.04 ± 0.08 |
| AutoDock Vina (Eberhardt et al., 2021) | 1.0 ± 1.3 | - | 0.2 ± 0.3 | 0.6 ± 0.7 | 1.1 ± 0.6 | 1.2 ± 0.5 |
| DrugCLIP (Gao et al., 2023) | 2.7 ± 1.26 | 1.63 ± 1.99 | 1.63 ± 0.81 | 2.45 ± 1.02 | 2.53 ± 1.35 | 2.69 ± 0.62 |
| PSICHIC (Koh et al., 2024) | 9.37 ± 3.08 | 4.07 ± 2.58 | 6.92 ± 3.30 | 7.48 ± 2.47 | 7.02 ± 1.80 | 5.35 ± 0.94 |
| GRAB (Yoo et al., 2021) | 40.68 ± 10.60 | 44.22 ± 8.35 | 45.21 ± 5.63 | 29.78 ± 1.38 | 18.69 ± 0.47 | **10.00 ± 0.00** |
| **Data mining-based** | | | | | | |
| avalon + NP (Yi et al., 2023) | 77.59 ± 1.72 | 135.76 ± 6.44 | 87.58 ± 2.9 | 31.55 ± 0.54 | 19.67 ± 0.4 | 9.88 ± 0.16 |
| estate + NP (Yi et al., 2023) | 52.44 ± 6.19 | 94.4 ± 13.68 | 57.87 ± 7.15 | 22.71 ± 2.7 | 15.92 ± 0.85 | 8.24 ± 0.38 |
| fp4 + NP (Yi et al., 2023) | 69.62 ± 3.69 | 122.76 ± 13.02 | 75.01 ± 4.21 | 28.96 ± 1.34 | 18.36 ± 1.0 | 9.59 ± 0.29 |
| graph + NP (Yi et al., 2023) | 75.86 ± 3.99 | 126.72 ± 10.05 | 84.73 ± 3.74 | 31.68 ± 0.92 | 19.1 ± 0.47 | 9.75 ± 0.24 |
| maccs + NP (Yi et al., 2023) | 75.44 ± 4.85 | 135.72 ± 12.7 | 79.82 ± 4.76 | 31.0 ± 1.41 | 18.93 ± 0.66 | 9.67 ± 0.21 |
| pubchem + NP (Yi et al., 2023) | 63.48 ± 5.16 | 99.17 ± 10.17 | 69.3 ± 7.08 | 30.87 ± 1.27 | 18.77 ± 0.9 | 9.84 ± 0.15 |
| rdkit + NP (Yi et al., 2023) | 79.04 ± 1.96 | 148.69 ± 4.25 | 89.24 ± 2.08 | 31.68 ± 0.92 | 19.02 ± 0.55 | 9.55 ± 0.3 |
| standard + NP (Yi et al., 2023) | 72.42 ± 3.51 | 121.97 ± 15.51 | 84.34 ± 5.56 | 31.27 ± 0.96 | 19.01 ± 0.33 | 9.71 ± 0.24 |
| SubDyve (Ours) | **83.44 ± 1.44** | **155.31 ± 6.38** | **97.59 ± 1.44** | **33.01 ± 0.60** | **19.90 ± 0.18** | **10.00 ± 0.00** |
| Statistical Significance (p-value) | ** | - | ** | * | - | - |

### 4.1.2 PU-Style Screening on CDK7 Target

**Task.** This task simulates a realistic virtual screening scenario with few known actives for a given target. We mask a portion of actives to evaluate the model's ability to rank them highly among many unlabeled candidates.

**Dataset & Evaluation.** We construct a PU (positive-unlabeled) dataset consisting of 1,468 CDK7-active compounds from PubChem and 10 million unlabeled molecules from ZINC. To ensure efficiency, we apply a subgraph-based reduction strategy that retains only compounds containing task-relevant substructures, yielding a filtered subset of approximately 30,000 ZINC compounds. We randomly select 30% of the actives for $\mathcal{S}_{\mathrm{train}}$, from which only 10% are used as a held-out set $\mathcal{S}_2$. These $\mathcal{S}_2$ compounds are excluded from subgraph generation to prevent leakage. Each experiment is repeated five times with different random seeds. We report BEDROC ($\alpha = 85$) and EF scores at various thresholds to assess early recognition. Detailed statistics and explanation of the evaluation settings are provided in Appendix D.2.

**Baselines.** We compare SubDyve with multiple baselines: (i) data mining-based methods using 12 standard molecular fingerprints (Yi et al., 2023), (ii) BIND (Lam et al., 2024), a foundation model trained on 2.4 million BindingDB interactions, (iii) PSICHIC (Koh et al., 2024), which learns protein-ligand interaction fingerprints, and (iv) GRAB (Yoo et al., 2021), a PU learning algorithm. For the data mining baselines, we construct a chemical similarity network and apply one round of propagation.

**Performance.** Table 2 presents the screening results. SubDyve achieves the highest BEDROC score (83.44) and consistently leads across enrichment thresholds, including $EF_{0.5\%}$ (155.31), $EF_{1\%}$ (97.59), and $EF_{3\%}$ (33.01). Compared to GRAB (Yoo et al., 2021), a PU learning baseline, SubDyve improves $EF_{1\%}$ by more than 2× (98.0 vs. 45.2) and BEDROC by over 80%. Against PSICHIC (Koh et al., 2024), which leverages interpretable interaction fingerprints, SubDyve improves $EF_{0.5\%}$ by over 38× and achieves a BEDROC nearly 9× higher. The foundation model BIND (Lam et al., 2024), despite being trained on millions of interactions, performs poorly in this setting ($EF_{10\%} = 0.04$), likely due to distribution mismatch. Notably, SubDyve not only achieves superior accuracy but also requires substantially less runtime than most baselines (e.g., AutoDock Vina: 13,343s; DrugCLIP: 2,985s; GRAB: 1,040s; RDKit: 957s; SubDyve: 1,088s), with full details summarized in Appendix D.3. Notably, SubDyve not only achieves superior accuracy but also runs approximately 12.3× faster than AutoDock Vina and requires 15.06× less memory than DrugCLIP, with complete runtime and memory comparisons provided in Appendix F.9. These results highlight SubDyve's strength in prioritizing true actives under minimal supervision and its robustness across compound representations and model classes.

Table 4: Ablation study on the number of seed compounds in the PU dataset. For each seed size (50, 150, 250), the first and second rows show the average and best-performing of general fingerprint baselines. Best values are in **bold**, second-best are underlined. Full results are in Appendix F.4.

| No. of Seeds | Method | BEDROC (%) | EF | | | | |
| --- | --- | --- | --- | --- | --- | --- | --- |
| | | | 0.30% | 0.50% | 1% | 3% | 5% |
| 50 | pubchem + NP | $41.13 \pm 4.46$ | $44.69 \pm 14.09$ | $45.51 \pm 7.91$ | $41.97 \pm 6.91$ | $25.7 \pm 1.99$ | $17.14 \pm 1.00$ |
| | maccs + NP | $47.02 \pm 3.83$ | $56.77 \pm 15.24$ | $52.81 \pm 9.24$ | $50.92 \pm 3.15$ | $27.74 \pm 2.04$ | $17.05 \pm 1.20$ |
| | Subgraph + NP | $46.33 \pm 1.26$ | $37.79 \pm 21.22$ | $31.81 \pm 12.68$ | **$53.93 \pm 4.97$** | $27.61 \pm 1.47$ | $17.27 \pm 0.51$ |
| | SubDyve | **$51.78 \pm 3.38$** | **$69.5 \pm 11.81$** | **$62.53 \pm 14.84$** | $52.66 \pm 5.91$ | **$29.48 \pm 2.37$** | **$18.15 \pm 0.90$** |
| 150 | rdkit + NP | $50.82 \pm 3.79$ | $52.69 \pm 6.75$ | $54.62 \pm 10.48$ | $54.62 \pm 7.24$ | $29.5 \pm 1.59$ | $17.79 \pm 0.95$ |
| | maccs + NP | $55.22 \pm 4.39$ | **$79.99 \pm 15.80$** | $71.65 \pm 13.30$ | $60.69 \pm 6.59$ | $30.6 \pm 1.29$ | $18.85 \pm 0.48$ |
| | Subgraph + NP | $55.08 \pm 1.52$ | $44.39 \pm 22.83$ | $61.29 \pm 10.07$ | **$67.17 \pm 7.24$** | $30.07 \pm 1.38$ | $18.22 \pm 0.93$ |
| | SubDyve | **$59.07 \pm 2.25$** | $74.67 \pm 7.46$ | **$73.55 \pm 10.51$** | $66.72 \pm 5.29$ | **$32.26 \pm 1.04$** | **$19.73 \pm 0.36$** |
| 250 | fp2 + NP | $56.88 \pm 5.26$ | $67.45 \pm 16.53$ | $75.57 \pm 15.28$ | $65.15 \pm 8.30$ | $30.19 \pm 1.26$ | $18.52 \pm 0.61$ |
| | avalon + NP | $61.29 \pm 2.44$ | $97.18 \pm 13.25$ | **$86.96 \pm 9.16$** | $68.05 \pm 4.42$ | $31.14 \pm 0.52$ | $19.51 \pm 0.48$ |
| | Subgraph + NP | $61.96 \pm 3.24$ | $41.01 \pm 13.89$ | $86.31 \pm 11.97$ | **$80.31 \pm 4.60$** | $30.20 \pm 1.44$ | $18.49 \pm 0.85$ |
| | SubDyve | **$66.73 \pm 2.71$** | **$97.69 \pm 16.55$** | $85.44 \pm 12.82$ | $78.19 \pm 3.38$ | **$32.85 \pm 0.60$** | **$19.72 \pm 0.36$** |

## 4.2 Ablation Study

To demonstrate the effectiveness of SubDyve components, we conduct ablation studies: (1) impact of subgraph-based similarity network and LFDR seed refinement, (2) initial seed set size, (3) behavior of LFDR refinement, (4) false positive (FP)-pressure via LFDR threshold, (5) subgraph pattern size, (6) impact of GNN features. Due to the limited space, experimental result of (4), (5), (6) is reported in Appendix F.6, F.7, and F.8, respectively.

### 4.2.1 Effects of Subgraph Network and LFDR Seed Refinement

We conduct an ablation study on the PU dataset to assess the impact of SubDyve's two main components: (i) the subgraph-based similarity network and (ii) LFDR-guided seed refinement. The experiments on the 10 DUDE targets are provided in Appendix F.3.

Table 3 shows that combining both components achieves the best performance (BEDROC 83.44, $EF_{1\%}$ 97.59), with improvements over all partial variants being statistically significant ($p < 0.01$, paired t-test). Using LFDR without subgraph features leads to a substantial drop in both BEDROC and EF, indicating that accurate refinement depends on the quality of the underlying network. Applying subgraph features without LFDR yields only modest im-

Table 3: Ablation results for the effect of subgraph fingerprints and LFDR-guided refinement with significance testing on the PU dataset. Top and second-best results are in **bold** and underline, respectively.

| Subgraph | LFDR | BEDROC | $EF_{1\%}$ |
| --- | --- | --- | --- |
| | | $79.04 \pm 1.96$ | $89.24 \pm 2.08$ |
| | ✓ | $63.78 \pm 11.43$ | $67.22 \pm 16.61$ |
| ✓ | | $78.68 \pm 2.87$ | $89.68 \pm 3.53$ |
| ✓ | ✓ | **$83.44 \pm 1.44^{**}$** | **$97.59 \pm 1.44^{**}$** |

provements, suggesting most gains come from their interaction. These results highlight that chemically meaningful network construction and uncertainty-aware refinement are complementary and essential for robust virtual screening in low-label settings.

### 4.2.2 Performance under Varying Seed Set Sizes

We conduct an ablation study to evaluate the effect of seed set size on the PU dataset. For each setting (50, 150, 250 seeds), we compare SubDyve against baselines using general-purpose molecular fingerprints and subgraph fingerprint network (Subgraph+NP). Detailed experimental settings are described in Appendix D.3.

Table 4 shows that SubDyve outperform across all seed sizes, demonstrating strong early enrichment even under limited supervision. While the best-performing general fingerprints vary by seed size (e.g., MACCS at 50, Avalon at 250), SubDyve achieves best performance without fingerprint-specific tuning. Notably, Subgraph+NP, using a network constructed from class-discriminative patterns, performs comparably to the best baselines, highlighting the effectiveness of substructure-aware graph construction. These results suggest that

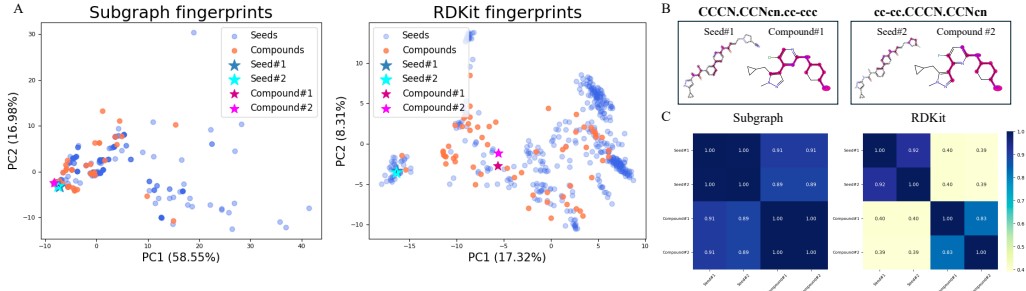

Figure 2: **Case study of seed–hit patterns from SubDyve vs. RDKit.** (A) PCA visualization of top 1% ranked compounds and seeds under each method, illustrating that SubDyve produces more coherent clustering in subgraph fingerprint space than RDKit. (B) Examples of structurally similar seed–hit pairs prioritized only by SubDyve, highlighting its ability to recover compounds with shared functional substructures. (C) Heatmaps of pairwise fingerprint similarity between seeds and retrieved hits, showing stronger seed–hit consistency with SubDyve fingerprints.

SubDyve combines robustness and adaptability across diverse label regimes without requiring task-specific fingerprint optimization.

### 4.2.3 Stability and Calibration Behavior of LFDR Refinement

We investigated the stability of SubDyve's LFDR-guided refinement by varying the calibration threshold while fixing the number of refinement iterations to six. Thresholds ($\tau \in 0.05, 0.10, 0.30, 0.50$) cover settings that emphasize either high-confidence updates or mid-confidence regions where many candidates fall between ambiguous structural cues. As summarized in Table 5, performance remained highly consistent: deviations in BEDROC, $EF_{1\%}$, and $EF_{5\%}$ were all within approximately 0.4% of

Table 5: Sensitivity analysis of SubDyve under varying LFDR thresholds ($\tau \in \{0.05, 0.10, 0.30, 0.50\}$). Each entry reports the relative performance difference (%) from the baseline configuration ($\tau = 0.10$).

| $\tau$ | $\Delta$ BEDROC | $\Delta EF_{1\%}$ | $\Delta EF_{5\%}$ |
|---|---|---|---|
| 0.05 | +0.04 | +0.40 | -0.48 |
| 0.10 (base) | +0.00 | +0.00 | +0.00 |
| 0.30 | -0.07 | -0.33 | -0.04 |
| 0.50 | +0.44 | +0.27 | +0.05 |

the default configuration ($\tau = 0.10$). These results indicate that LFDR refinement is not overly sensitive to the choice of threshold and maintains stable ranking behavior even when the calibration boundary is shifted toward more uncertain regions. A full two-dimensional sweep over thresholds and iteration counts is provided in Appendix F.5.

### 4.3 Case study

To further demonstrate the interpretability and structural behavior of SubDyve, we conduct four case studies: (1) a comparison with RDKit fingerprints on CDK7 to assess local substructure similarity; (2) ranking gap for structurally similar molecules on DUD-E targets; (3) an analysis of active/decoy recovery patterns on DUD-E targets with varying seed sizes; (4) characterization of subgraph patterns from augmented seeds on CDK7; and (5) pharmacophoric relevance assessment of the mined subgraphs on CDK7. Due to the limited space, experimental results of (3), (4), (5) are reported in Appendix G.1, G.3, G.4, respectively.

### 4.3.1 Substructure Similarity in CDK7-Target Compound Retrieval

To evaluate the representational advantage of SubDyve over general-purpose molecular fingerprints, we compare its retrieval behavior with RDKit on a pair of structurally similar seed compounds on PU dataset. Specifically, we visualize the compounds prioritized in the top 1% by each method, alongside their seed compounds, using PCA projections of their respective fingerprint spaces (Figure 2A). SubDyve's retrieved compounds form a tight cluster around the seeds in the subgraph fingerprint space, indicating high local consistency and

shared substructural motifs. In contrast, RDKit-prioritized compounds are more scattered, despite being selected from the same ranking percentile, highlighting the method's weaker capacity to preserve functional substructure similarity.

A closer inspection of two representative seed–hit pairs (Figure 2B) reveals that SubDyve successfully prioritizes compounds containing key substructures, such as penta-1,3-diene(cc-ccc) or butadiene groups(cc-cc), that align well with the seeds. These hits were not retrieved by RDKit, likely due to its reliance on predefined global fingerprints that overlook localized structural alignment.

To further quantify this effect, Figure 2C presents similarity heatmaps between seeds and retrieved compounds. Subgraph fingerprint similarities remain consistently high across pairs, while RDKit similarities are notably lower, even for structurally related hits.

These results suggest that SubDyve offers superior sensitivity to activity-relevant substructures, making it especially well-suited for discovering functionally analogous compounds in early-stage virtual screening.

### 4.3.2 Ranking gap analysis for structurally similar pairs

To evaluate the ranking efficiency of SubDyve, we perform a ranking gap comparison analysis with general-purpose molecular fingerprints on structurally similar molecule pairs with activity differences. For the FA10 target in the DUD-E benchmark, we extract active–decoy pairs with high structural similarity (Tanimoto similarity $\geq 0.85$) and find that SubDyve significantly ranks actives higher and decoys lower than the RDKit + NP model, as shown in the Figure 3. Similar trends are observed for other DUD-E targets, as shown in Appendix G.2.

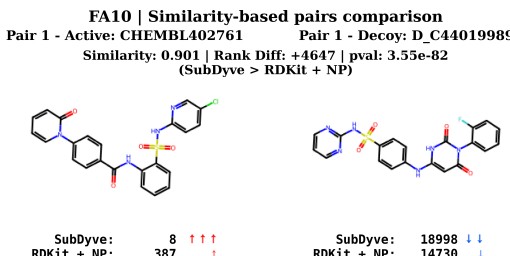

Figure 3: Ranking gap and visualization of highly similar active–decoy pairs for FA10 on DUD-E targets shown with model relevance ranks. Statistical significance is reported as Wilcoxon signed-rank p-value.

## 5 Conclusion

We present SubDyve, a label-efficient virtual screening framework that constructs a task-adaptive subgraph fingerprint network by mining class-discriminative substructures from bioactivity-labeled compounds. Built upon these chemically meaningful patterns, SubDyve performs iterative seed refinement using LFDR-guided calibration to prioritize candidate molecules with minimal supervision. Our method achieves more than a twofold improvement in average $EF_{1\%}$ on the zero-shot DUD-E benchmark and delivers strong BEDROC and EF performance in large-scale screening on the PU dataset. These results demonstrate that integrating substructure-similarity network construction with uncertainty-aware propagation offers a scalable and effective solution for virtual screening in low-label regimes, advancing the feasibility of early-phase hit discovery.

Although SubDyve is formulated as a target-specific screening framework, its design suggests potential for broader applicability. The subgraph-based similarity network can incorporate protein-informed or structure-derived descriptors (Koh et al., 2024; Desaphy et al., 2013), while the LFDR refinement remains compatible with such extensions because it operates only on propagated scores. Our study focuses on ligand-only benchmarks, and applying SubDyve to settings that integrate protein information or involve multiple related targets remains a limitation; however, the observed correspondence between mined substructures and pocket-relevant chemotypes indicates that extending the framework toward richer protein–ligand contexts represents a meaningful direction for future work.

## 6 ACKNOWLEDGEMENT

This research was supported by the Institute of Information & Communications Technology Planning & Evaluation (IITP) grant funded by the Korean government (MSIT) for the Artificial Intelligence Graduate School Program at Seoul National University (RS-2021-II211343) and the Artificial Intelligence Convergence Innovation Human Resources Development Program at Inha University (RS-2022-00155915), as well as by the National Research Foundation of Korea (NRF) grants funded by the Korean government (MSIT) (RS-2023-00257479), the Bio & Medical Technology Development Program (RS-2022-NR067933), and the Basic Science Research Program funded by the Ministry of Education (RS-2023-00246586).

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

APPENDIX

## A    PROOF FOR PROPOSITION

*Proof.* of proposition 1.

[local–FDR $\leq \alpha$ implies FDR $\leq \alpha$]

Let $Z_1, \ldots, Z_m$ be test statistics following the two–group mixture

$$f(z) = \pi_0 f_0(z) + \pi_1 f_1(z), \qquad \pi_0 + \pi_1 = 1, \ \pi_1 > 0.$$

Define the *local false–discovery rate* (Efron, 2005) (Efron, 2005)

$$\mathrm{lfdr}(z) = \Pr(H = 0 \mid Z = z) = \frac{\pi_0 f_0(z)}{f(z)}.$$

Choose hypotheses by

$$\mathcal{R}_\alpha = \big\{ i : \mathrm{lfdr}(Z_i) \leq \alpha \big\}, \qquad 0 < \alpha < 1.$$

Let $R_\alpha = |\mathcal{R}_\alpha|$ and $V_\alpha = \sum_{i=1}^m I\{i \in \mathcal{R}_\alpha\} H_i$.
Then

$$\mathrm{mFDR} = \frac{\mathbb{E}[V_\alpha]}{\mathbb{E}[R_\alpha]} \leq \alpha, \qquad \mathrm{FDR} = \mathbb{E}\Big[\frac{V_\alpha}{R_\alpha \vee 1}\Big] \leq \alpha.$$

Write $\mathbb{E}[V_\alpha] = \sum_i \mathbb{E}\big[\mathrm{lfdr}(Z_i)\, I\{i \in \mathcal{R}_\alpha\}\big]$. Because $\mathrm{lfdr}(Z_i) \leq \alpha$ whenever $i \in \mathcal{R}_\alpha$,

$$\mathbb{E}[V_\alpha] \leq \alpha\, \mathbb{E}[R_\alpha].$$

Dividing both sides gives $\mathrm{mFDR} \leq \alpha$. Since $V_\alpha \leq R_\alpha$, Jensen's inequality yields $\mathrm{FDR} \leq \mathrm{mFDR} \leq \alpha$. $\qquad\square$

**mFDR (Marginal FDR)**

$$\mathrm{mFDR} = \frac{\mathbb{E}[V]}{\mathbb{E}[R]}$$

Marginal FDR takes expectations of the numerator and denominator separately, providing a *mean* proportion of false discoveries. It is always defined (no $0/0$ issue when $R = 0$) and satisfies $\mathrm{FDR} \leq \mathrm{mFDR}$.

## B    MODEL ARCHITECTURE AND LOSS DETAILS

### B.1    PSEUDOCODE OF SUBDYVE

Algorithm 1 outlines the full SubDyve framework for virtual screening. The procedure consists of three main stages: (1) Subgraph fingerprint network construction, (2) Dynamic seed refinement with LFDR calibration, and (3) Final compound prioritization via network propagation. In Step 1, we mine class-discriminative substructures and use them to construct a molecular similarity graph (details in Appendix B.2). Step 2 performs $N$-fold stratified refinement using GNN model with a composite loss and iteratively expands the seed set based on LFDR calibration. For each split, the seed weights from the best-performing iteration are retained. Step 3 aggregates the $N$ seed weight vectors via max pooling and performs final propagation to produce the ranked list of candidate compounds. The LFDR refinement step is described in detail in Algorithm 2.

---

**Algorithm 1** SUBDYVE FRAMEWORK FOR VIRTUAL SCREENING

**Require:**

Initial labeled set $\mathcal{S}_{\text{train}}$, unlabeled pool $\mathcal{Q}'$

Number of stratified splits $N$, number of iterations $M$

Hyper-parameters $(\lambda_{\text{rank}}, \lambda_{\text{con}}, \gamma_{\text{np}}, \tau, \beta, \theta)$

1: **// Step 1: Subgraph fingerprint Network Construction (see Appendix B.2**)
2: Mine class-discriminative subgraph patterns from $\mathcal{S}_{\text{train}}$ using a supervised subgraph mining algorithm
3: Construct subgraph pattern fingerprints for all $v \in \mathcal{Q}'$
4: Compute pairwise cosine similarity between fingerprints to construct the subgraph fingerprint graph $\mathcal{G} = (\mathcal{V}, \mathcal{E}, \mathbf{w}_e)$

5: **// Step 2: Dynamic Seed Refinement with LFDR**
6: **for** $n = 1$ **to** $N$ **do**                                  ▷ Stratified bootstraps of $\mathcal{S}_{\text{train}}$
7: $\quad(\mathcal{S}_1, \mathcal{S}_2) \leftarrow \texttt{Split}(\mathcal{S}_{\text{train}}, \text{ratio})$
8: $\quad$Compute node features $\mathbf{x}(v)$ for all $v \in \mathcal{V}$                        ▷ Appendix B.3
9: $\quad$Initialize augmented seeds $\mathcal{S}_{\text{aug}} \leftarrow \varnothing$, seed weight map $\mathbf{s}$
10: $\quad$**for** $m = 1$ **to** $M$ **do**                                        ▷ Iteration loop
11: $\quad\quad\mathbf{X} \leftarrow [\mathbf{x}(v)]_{v \in \mathcal{V}}$
12: $\quad\quad(\boldsymbol{\ell}, \mathbf{z}) \leftarrow \mathcal{M}_\theta(\mathbf{X}, \mathcal{E}, \mathbf{w}_e)$                   ▷ Logits $\boldsymbol{\ell}$ and embeddings $\mathbf{z}$ from GNN
13: $\quad\quad\mathcal{L}_{\text{BCE}} \leftarrow \texttt{WeightedBCE}(\boldsymbol{\ell}, \mathcal{S}_2, \gamma_{\text{np}})$
14: $\quad\quad\mathcal{L}_{\text{Rank}} \leftarrow \texttt{PairwiseRankNet}(\boldsymbol{\ell}, \mathcal{S}_2)$
15: $\quad\quad\mathcal{L}_{\text{Con}} \leftarrow \texttt{InfoNCE}(\mathbf{z}, \mathcal{S}_2)$
16: $\quad\quad\mathcal{L}_{\text{total}} \leftarrow (1 - \lambda_{\text{rank}}) \cdot \mathcal{L}_{\text{BCE}} + \lambda_{\text{rank}} \cdot \mathcal{L}_{\text{Rank}} + \lambda_{\text{con}} \cdot \mathcal{L}_{\text{Con}}$            ▷ Appendix B.4
17: $\quad\quad\texttt{Update}$ model parameters via $\nabla \mathcal{L}_{\text{total}}$
18: $\quad\quad(\mathcal{S}_{\text{aug}}, \mathbf{s}) \leftarrow \text{ALGORITHM } 2(\boldsymbol{\ell}, \mathcal{S}_1, \mathcal{S}_{\text{aug}}, \mathbf{s}, \tau, \beta)$    ▷ LFDR-guided Seed Refinement
19: $\quad$**end for**
20: $\quad$Save $\mathbf{s}_n$ from best-performing iteration on $\mathcal{S}_2$
21: **end for**

22: **// Step 3: Final Prioritization**
23: Aggregate $\{\mathbf{s}_n\}_{n=1}^N$ via element-wise max pooling to obtain ensembled seed vector $\mathbf{s}^*$
24: Perform final network propagation over $\mathcal{G}$ using $\mathbf{s}^*$ to score $\mathcal{Q}'$
25: **return** Final ranked list of compounds based on propagation scores

---

## B.2    DETAILS OF SUBGRAPH FINGERPRINT NETWORK CONSTRUCTION

This section describes the full pipeline for constructing a subgraph fingerprint network. The objective is to extract class-discriminative substructures, enabling more effective propagation and compound ranking. The process consists of three main stages: (1) mining class-discriminative subgraph patterns, (2) generating continuous subgraph pattern fingerprints, and (3) constructing the subgraph fingerprint network.

### B.2.1    MINING CLASS-DISCRIMINATIVE SUBGRAPH PATTERNS

We adopt the Supervised Subgraph Mining (SSM) algorithm (Lim et al., 2023) to identify substructures that differentiate active and inactive compounds. We curate activity-labeled data from the PubChem dataset by extracting compounds annotated as bioactive against the target of interest. Candidate subgraphs are generated using a supervised random walk strategy: for each node $v \in \mathcal{V}(\mathcal{G})$, a fixed-length walk is repeated multiple times to sample a diverse set of subgraphs. Each subgraph is decomposed into atom-pair doublets to estimate class-specific transition preferences. These preferences iteratively refine the walk policy, guiding subsequent sampling toward class-informative regions.

The mined subgraphs are evaluated using a classifier, and the subgraph set that yields the highest predictive performance (e.g., in AUC) is selected as the final set $Sub^{\text{opt}}$. In parallel,

---

**Algorithm 2** LFDR-guided Seed Refinement

---

**Require:** Ranking logits $l_i$ for all $i \in \mathcal{V}$, initial train seeds $\mathcal{S}_1$, current augmented seeds $\mathcal{S}_{\mathrm{aug}}$, seed weight $w_i \in$ seed weight map $\mathbf{s}$, LFDR thresholds $\tau_{\mathrm{FDR}}$, update rate $\beta$, baseline $b$

1: Compute z-scores: $z_i \leftarrow \texttt{zscore}(l_i)$
2: Estimate local FDR: $\mathrm{LFDR}_i \leftarrow \texttt{local\_fdr}(z_i)$ ▷ Algorithm 3
3: **for** each node $i \in \mathcal{V}$ **do**
4:    **if** $i \notin \mathcal{S}_{\mathrm{aug}}$ and $\mathrm{LFDR}_i < \tau_{\mathrm{FDR}}$ **then**
5:        $\mathcal{S}_{\mathrm{aug}} \leftarrow \mathcal{S}_{\mathrm{aug}} \cup \{i\}$ ▷ Add high-confidence node
6:        $w_i \leftarrow 1.0$
7:    **else if** $i \in \mathcal{S}_{\mathrm{aug}} \setminus \mathcal{S}_1$ and $\mathrm{LFDR}_i > \tau_{\mathrm{FDR}}$ **then**
8:        $\mathcal{S}_{\mathrm{aug}} \leftarrow \mathcal{S}_{\mathrm{aug}} \setminus \{i\}$ ▷ Remove low-confidence node
9:        $w_i \leftarrow 0$
10:    **else if** $i \in \mathcal{S}_{\mathrm{aug}}$ **then**
11:        $w_i \leftarrow w_i + \beta \cdot (\sigma(l_i) - b)$ ▷ Update existing seed weight
12:    **end if**
13: **end for**
14: **return** Updated $S_{\mathrm{aug}}$, $s$

---

**Algorithm 3** LFDR Estimation

---

**Require:** Observed Z-scores $Z = \{Z_i\}_{i=1}^{\mathcal{V}}$, null density $f_0(z)$, bin count $B$, polynomial degree $d$, regularization parameter $\alpha \geq 0$, null proportion $\pi_0$
**Ensure:** Local FDR estimates $\widehat{\mathrm{lfdr}}(Z_i)$ for all $i = 1, \ldots, \mathcal{V}$

1: Partition the range $[\min Z, \max Z]$ into $B$ equal-width bins $(b_{j-1}, b_j]$ with centers $z_j = \frac{1}{2}(b_{j-1} + b_j)$
2: Count samples in each bin: $N_j \leftarrow \#\{Z_i \in (b_{j-1}, b_j]\}$ for $j = 1, \ldots, B$
3: Construct design matrix $X \in \mathbb{R}^{B \times (d+1)}$ with $X_{jk} = z_j^{k-1}$ for $k = 1, \ldots, d+1$
4: Fit Poisson distribution:

$$\hat{\boldsymbol{\beta}} \leftarrow \arg\min_{\boldsymbol{\beta}} \left\{ -\sum_{j=1}^{B} \left[ N_j \cdot (\mathbf{x}_j^\top \boldsymbol{\beta}) - \exp(\mathbf{x}_j^\top \boldsymbol{\beta}) \right] \right.$$
$$\left. + \frac{\alpha}{2} \|\boldsymbol{\beta}\|_2^2 \right\}$$

5: **for** each $i = 1, \ldots, \mathcal{V}$ **do**
6:    Construct polynomial features $\mathbf{x}(Z_i) = \left( Z_i^0, Z_i^1, \ldots, Z_i^d \right)^\top$
7:    Estimate marginal density: $\widehat{f}(Z_i) \leftarrow \exp\left( \mathbf{x}(Z_i)^\top \hat{\boldsymbol{\beta}} \right)$
8:    Compute null density: $f_0(Z_i) \leftarrow \texttt{null\_pdf}(Z_i)$
9:    Compute LFDR:
$$\widehat{\mathrm{lfdr}}(Z_i) \leftarrow \frac{\pi_0 \cdot f_0(Z_i)}{\widehat{f}(Z_i)}$$
10:    Clip to $[0, 1]$: $\widehat{\mathrm{lfdr}}(Z_i) \leftarrow \min(1, \max(0, \widehat{\mathrm{lfdr}}(Z_i)))$
11: **end for**
12: **return** $\{\widehat{\mathrm{lfdr}}(Z_i)\}_{i=1}^{\mathcal{V}}$

---

we identify single-subgraph structural alerts (SAs) by computing feature importance scores using a random forest model. Subgraphs with importance above 0.0001 and entropy below 0.5 are retained as interpretable indicators of activity.

### B.2.2 Generating Subgraph Pattern Fingerprints

To capture higher-order structure, we construct Discriminative Subgraph Combinations (DiSCs)—co-occurring subgraph sets that frequently appear in actives. Starting with 1-mer subgraphs, we iteratively build $k$-mer combinations using a branch-and-bound search with SMARTS-based pattern grouping. Candidates are scored using an entropy-based metric $1 - \text{Entropy}(\text{Supp}_{pos}, \text{Supp}_{neg})$, and only those with sufficient support ($\geq 2\%$) and discriminative power are retained. Entropy filtering is not applied to 1-mers to preserve informative small motifs.

The top-$d$ DiSCs are selected based on entropy rank and used to construct a $d$-dimensional fingerprint vector, where each entry encodes the frequency of a specific subgraph combination within the molecule. These fingerprint vectors serve as task-aware molecular representations for graph construction.

### B.2.3 Constructing Molecular Similarity Networks

We construct a similarity graph $\mathcal{G} = (\mathcal{V}, \mathcal{E}, \mathbf{w}_e)$ by computing pairwise cosine similarity between subgraph pattern fingerprints. Each compound in $\mathcal{Q}'$ is represented as a node, and weighted edges reflect structural proximity in the DiSC space.

### B.3 Details of Feature Encoding for GNN

In the dynamic seed refinement step, we use a two-layer GNN to predict activity scores over the subgraph fingerprint network. Each compound $i \in \mathcal{Q}'$ is encoded as:

$$\mathbf{x}_i = [w_i, n_i^{\text{NP}}, \mathbf{f}_i^{\text{FP}}, s_i^{\text{PCA}}, h_i^{\text{hyb}}, \mathbf{e}_i^{\text{PT–CB}}] \tag{5}$$

The components of the feature vector are described below:

- $w_i$: Weight of seed to use for network propagation. Initially set $w_{i \in \mathcal{S}_1}$ to 1, otherwise set to 0.
- $n_i^{\text{NP}}$: Network propagation score drive from $w_i$.
- $\mathbf{f}_i^{\text{FP}}$: Class-discriminative substructure features extracted from subgraph pattern fingerprints.
- $s_i^{\text{PCA}}$: RBF Similarity to seed compounds in a PCA-projected latent space based on subgraph pattern fingerprints $\mathbf{f}_i^{\text{FP}}$.
- $h_i^{\text{hyb}}$: Hybrid ranking score computed as a weighted average of the rankings of $s_i^{\text{PCA}}$ and $n_i^{\text{NP}}$.
- $\mathbf{e}_i^{\text{PT–CB}}$: Semantic features derived from a pretrained ChemBERTa model, representing molecular sequence semantics.

Each GNN layer is followed by a residual connection and LayerNorm, with the second layer reducing the hidden dimension by half. The model outputs a scalar logit $\hat{l}_i$ computed via a linear layer for ranking, along with a 32-dimensional embedding vector for representation regularization.

### B.4 Details of Composite Loss for GNN

Following (Lin et al., 2024), SubDyve jointly optimizes classification, ranking, and representation learning objectives within the GNN during seed refinement. The final loss is a weighted sum of three components: binary cross-entropy $\mathcal{L}_{\text{BCE}}$, pairwise ranking loss $\mathcal{L}_{\text{RankNet}}$, and contrastive loss $\mathcal{L}_{\text{Contrast}}$. Each component is designed to enhance model performance under sparse supervision.

### 1. Binary Cross-Entropy Loss (BCE)

We employ a class-balanced BCE loss to accommodate severe class imbalance. Additionally, compound-level weights modulated by network propagation scores enhance robustness to

noisy supervision:

$$\mathcal{L}_{\text{BCE}} = \frac{1}{|\mathcal{Q}'|} \sum_{i=1}^{|\mathcal{Q}'|} w_i \cdot \left[ y_i \cdot \log \sigma(\hat{l}_i) \right.$$

$$\left. + \text{PW} \cdot (1 - y_i) \cdot \log(1 - \sigma(\hat{l}_i)) \right], \tag{6}$$

$$\sigma(\hat{l}_i) = \frac{1}{1 + e^{-\hat{l}_i}}, \quad w_i = 1 + \gamma_{\text{np}} \cdot n_i^{\text{NP}}$$

where $\hat{l}_i$ is the predicted logit for compound $i$, and $y_i \in \{0, 1\}$ is the ground truth label indicating whether $i \in \mathcal{S}_2$ (active) or not (inactive). $n_i^{\text{NP}}$ is the NP score from initial propagation. $\gamma_{\text{np}}$ is set to 5. The term PW balances class skew by weighting the active class more heavily: $\text{pos\_weight} = \frac{|\{i|y_i=0\}|}{|\{i|y_i=1\}|+\epsilon}$.

## 2. Pairwise RankNet Loss

To improve early recognition, we adopt a pairwise margin-based loss that encourages higher scores for known actives in $\mathcal{S}_2$ relative to likely inactives:

$$\mathcal{L}_{\text{RankNet}} = \frac{1}{C} \sum_{(i,j)} \max\left(0, m - (\hat{l}_i - \hat{l}_j)\right), \tag{7}$$

$$i \in \mathcal{S}_2, ; j \in \mathcal{Q}' \setminus \mathcal{S}_2.$$

Here, $m$ is a margin hyperparameter and $C$ denotes the number of valid $(i, j)$ pairs.

## 3. Contrastive Loss (InfoNCE)

This loss promotes intra-class consistency in the learned embeddings. For each compound $i \in \mathcal{Q}'$, we select its most similar positive compound $z_{i+}$ from $\mathcal{S}_2$ based on subgraph pattern fingerprint similarity, and treat the remaining compounds in $\mathcal{S}_2$ as $z_{i-}$.

$$\mathcal{L}_{\text{Contrast}} = \frac{1}{|\mathcal{S}2|} \times$$

$$\sum_{i \in \mathcal{S}2} -\log\left(\frac{\exp\left(\frac{z_i^\top zi^+}{\tau}\right)}{\exp\left(\frac{z_i^\top z_{i+}}{\tau}\right) + \sum_k \exp\left(\frac{z_i^\top z_{i-}^{(k)}}{\tau}\right)}\right) \tag{8}$$

where $\tau$ is a temperature parameter.

## 4. Total Composite Loss

The total loss is a weighted combination:

$$\mathcal{L}_{\text{total}} = (1 - \lambda_{\text{rank}}) \cdot \mathcal{L}_{\text{BCE}} +$$
$$\lambda_{\text{rank}} \cdot \mathcal{L}_{\text{RankNet}} + \tag{9}$$
$$\lambda_{\text{contrast}} \cdot \mathcal{L}_{\text{Contrast}}.$$

where $\lambda_{\text{rank}} = 0.3$ and $\lambda_{\text{contrast}} = 0.6$ fixed across all experiments. The GNN is trained using Adam optimizer (Kingma & Ba, 2014) with a fixed learning rate of $8 \times 10^{-4}$ and weight decay of $1.57 \times 10^{-5}$. Hyperparameter selection is discussed in Appendix C.2. The entire code and reproduced experiments is available at `https://github.com/J-Sub/SubDyve`.

## C  Implementation & Evaluation Details

### C.1  Network Propagation Algorithm

Network propagation (NP) is used to prioritize candidate compounds by diffusing signals from a small number of known actives across a chemical similarity network. This approach has been shown to effectively integrate relational structure for large-scale inference (Cowen et al., 2017). NP iteratively balances the initial bioactivity signal carried by $S$ with the topological context supplied by the graph, allowing evidence to flow along indirect paths and uncovering nodes that are not immediate neighbors of the seeds:

$$P^{(t+1)} = (1 - \alpha) W_{\mathcal{N}} P^{(t)} + \alpha P^{(0)}, \tag{10}$$

where $P^{(0)}$ is a one-hot vector encoding compounds $S$, $W_{\mathcal{N}}$ is the column-normalized adjacency matrix, and $\alpha \in [0, 1]$ controls the restart probability. Over iterations, the score vector $P^{(t)}$ converges to a stationary distribution that captures both local and global connectivity, thereby ranking compounds in $Q$ by their network proximity to $S$. By integrating signals along multiple paths rather than relying solely on direct neighbors, NP effectively highlights previously unconnected yet pharmacologically relevant candidates, making it well suited for large-scale virtual screening task.

Network propagation (NP) prioritizes candidate compounds by diffusing activity signals from a small set of known actives across a chemical similarity network. This method effectively incorporates both local and global graph structure, enabling inference over indirect molecular relationships (Cowen et al., 2017).

The propagation is formulated as an iterative update:

$$P^{(t+1)} = (1 - \alpha) W_{\mathcal{N}} P^{(t)} + \alpha P^{(0)}, \tag{11}$$

where $W_{\mathcal{N}}$ is the column-normalized adjacency matrix of the molecular graph, and $\alpha \in [0, 1]$ is the restart probability. The initial vector $P^{(0)}$ encodes seed activity, typically assigning 1 to known actives and 0 elsewhere.

As iterations proceed, $P^{(t)}$ converges to a stationary distribution that reflects both direct and indirect connectivity to the seed set. This enables the identification of structurally distant yet functionally related candidates, making NP a suitable backbone for large-scale virtual screening under sparse supervision.

### C.2  Hyperparameter Search Space

We perform hyperparameter optimization in two phases depending on the parameter type. GNN model architecture and loss-related parameters are tuned using Bayesian Optimization (100 iterations) (Appendix Table 5). Hyperparameters related to iteration process on dynamic seed refinement are searched via random search (Appendix Table 6).

### C.3  Evaluation Metrics

To evaluate the performance of early retrieval in virtual screening, we adopt the BEDROC and Enrichment Factor (EF) metrics.

**BEDROC.** The Boltzmann-Enhanced Discrimination of ROC (BEDROC) is designed to emphasize early recognition by assigning exponentially decreasing weights to lower-ranked active compounds. It is defined as:

$$\begin{aligned}
\text{BEDROC}_\alpha &= \left( \frac{1 - e^{-\alpha}}{1 - e^{-\alpha/N}} \right) \left( \frac{1}{n} \sum_{i=1}^{n} e^{-\alpha r_i / N} \right) \times \\
&\left( \frac{\sinh(\alpha/2)}{\cosh(\alpha/2) - \cosh(\alpha/2 - \alpha R_\alpha)} \right) + \frac{1}{1 - e^{\alpha(1 - R_\alpha)}}
\end{aligned} \tag{12}$$

Table 5: Hyperparameters related to GNN model

| Parameter | Search Space | Selected Value |
|---|---|---|
| Hidden dimension | {16, 32, 64, 128} | 64 |
| Embedding dimension | {8, 16, 32, 64} | 32 |
| $\lambda_{\text{rank}}$ | [0.0, 1.0] | 0.3 |
| $\lambda_{\text{contrast}}$ | [0.0, 1.0] | 0.6 |
| Margin for RankNet loss | [0.0, ..., 1.0] | 0.5 |
| Weight decay | $[10^{-6}, 10^{-4}]$ | $1.57 \times 10^{-5}$ |
| $\beta$ (seed weight update rate) | [0.1, 1.0] | 0.7 |
| Learning rate | $[10^{-4}, 10^{-2}]$ | 0.0008 |
| $\gamma_{\text{NP}}$ (NP score weight) | {0.0, ..., 5.0} | 5.0 |
| GNN layer type | {GCN, GIN, GAT} | GCN |

Table 6: Hyperparameters related to iteration of seed refinement

| Parameter | Search Space | Selected Value |
|---|---|---|
| Training epochs | {10, 20, 30, 40, 50} | 50 |
| Max iterations ($M$) | {3, 4, 5, 6, 7} | 6 |
| Early stopping patience of iterations | {1, 2, 3} | 3 |
| Stratified split ($N$) | {10, 5, 3, 2} | 2 |
| LFDR threshold $\tau_{\text{FDR}}$ | {0.03, 0.05, 0.1, 0.3, 0.5} | 0.1 |

$n$ is the number of active compounds, $N$ is the total number of molecules, $r_i$ is the rank of the $i$-th active compound, and $R_\alpha = n/N$. Following prior work (Truchon & Bayly, 2007), we set $\alpha = 85$ to prioritize early retrieval.

**Enrichment Factor (EF).** The EF quantifies the proportion of actives retrieved within the top-ranked subset relative to a random distribution. It is computed as:

$$\text{EF}_{x\%} = \frac{n_a/N_{x\%}}{n/N} \tag{13}$$

$n$ is the number of active compounds, $N$ is the total number of molecules, $N_{x\%}$ is the number of molecules in the top $x\%$ of the ranking, and $n_a$ is the number of actives within that portion. Higher EF values indicate better prioritization of active compounds in the early ranks.

## D    Experiment Details

### D.1    Zero-Shot Virtual Screening Setup on Ten DUD-E Targets

For each DUD-E target, we curate a high-quality dataset of bioactive compounds from PubChem while preserving the zero-shot setting. To avoid data leakage, we filter protein homologs using MMseqs2 (Steinegger & Söding, 2017), excluding any proteins with sequence identity greater than 90% relative to the target. From the remaining homologs (identity $\leq 0.9$), we retrieve associated compounds and their bioactivity annotations, retaining only those labeled as active or inactive with valid potency measurements. Duplicate entries and records with missing values are removed to ensure data reliability. We additionally compare the FM-based baseline model with ChemBERTa (Ahmad et al., 2022) and MoLFormer (Ross et al., 2022) models for further comparison. Using the pre-trained models, we calculate performance by averaging the embeddings of bioactive compounds using the pre-trained models and taking the active and inactive compounds from DUD-E and ranking them according to their cosine similarity to each compound. For the ADA target, where PubChem

Table 7: Summary of PubChem-augmented data for DUD-E targets, including similarity ranges, and number of seed molecules used in propagation.

| Target | PDB code | Active Ligands | Decoy Ligands | PubChem Total (Act/Inact) | Similarity Range | Seed Count |
|--------|----------|----------------|---------------|---------------------------|------------------|------------|
| ACES | 1e66 | 451 | 26198 | 1604 (1502/102) | 0.647–0.9 | 1277 |
| ADA | 2e1w | 90 | 5448 | 444 (386/58) | 0.909–0.953 | 335 |
| ANDR | 2am9 | 269 | 14333 | 918 (822/96) | 0.56–0.9 | 755 |
| EGFR | 2rgp | 541 | 35001 | 576 (427/149) | 0.478–0.9 | 374 |
| FA10 | 3kl6 | 537 | 28149 | 261 (237/24) | 0.845–0.9 | 195 |
| KIT | 3g0e | 166 | 10438 | 3299 (3164/135) | 0.537–0.9 | 771 |
| PLK1 | 2owb | 107 | 6794 | 353 (191/162) | 0.61–0.9 | 174 |
| SRC | 3el8 | 523 | 34407 | 1232 (827/405) | 0.88–0.9 | 624 |
| THRB | 1ype | 461 | 26894 | 3126 (2071/1055) | 0.833–0.9 | 477 |
| UROK | 1sqt | 162 | 9837 | 825 (750/75) | 0.489–0.9 | 615 |

annotations are sparse, a slightly higher identity threshold (up to 0.953) is used to enable sufficient subgraph extraction.

Appendix Table 7 summarizes the number of actives/inactives and the protein similarity thresholds used per target. For downstream propagation, we construct a target-specific subgraph fingerprint network comprising PubChem actives and DUD-E molecules (including actives and decoys). PubChem actives with $IC_{50} \leq 500$ nM are selected as seed molecules, while the remaining actives are incorporated as unlabeled nodes. Subgraph patterns are extracted via a Murcko-scaffold split and encoded into 2000-dimensional subgraph fingerprints, which serves as the molecular representation for each node in the graph.

**Baseline Models Training Summary**   We evaluate SubDyve against two structure-based virtual screening baselines: CDPKit (Alignment) (Seidel, 2024) and PharmacoMatch (Rose et al., 2025) (Table 8). CDPKit is a geometric alignment algorithm that performs unsupervised pharmacophore matching without model training, relying solely on spatial fit. In contrast, PharmacoMatch is a self-supervised learning framework trained on 1.2 million drug-like compounds from ChEMBL((Davies et al., 2015; Zdrazil et al., 2024)). It learns a joint embedding space for pharmacophore graphs using contrastive loss and an order embedding objective, enabling similarity-based retrieval of actives without direct supervision.

Table 8: Key characteristics for baseline methods.

| Model | Training Data Description |
|-------|--------------------------|
| PharmacoMatch | 1.2M small molecules from ChEMBL after Lipinski filtering and duplication removal; trained in a self-supervised fashion on 3D pharmacophore graphs using contrastive loss and order embeddings. |
| CDPKit (Alignment) | Unsupervised alignment of 3D pharmacophores using geometric fit (no training). |
| ChemBERTa | ChemBERTa-2 is a model based on ChemBERTa that optimises pre-training performance through large-scale pre-training and multi-task-self-supervised learning comparisons using up to 77 million molecular data. |
| MoLFormer | MoLFormer is a transformer-based molecular language model trained using efficient linear attentions and rotational positional embeddings on 1.1 billion molecules (SMILES) data, outperforming traditional graph and language-based models in many of the 10 benchmarks. |
| DrugCLIP | DrugCLIP is a dense retrieval-based contrastive learning model that solves the virtual screening problem by learning the similarity between a protein pocket and a molecule. The model leverages extensive data, including over 120,000 protein-molecule pairs and more than 17,000 complex structures from PDBbind, BioLip, and ChEMBL datasets, and utilizes the HomoAug data augmentation method to maximize the diversity of the training set. |

**Robustness to distant homologs Setup**   To examine whether SubDyve depends on closely related homologs when operating in the zero-shot setting, the sequence-identity threshold in MMseqs2 used for seed generation was lowered from 0.9 to 0.5. This adjustment ensures that seeds originate from substantially more distant PubChem proteins, thereby creating a more challenging and realistic zero-shot condition in which homologs often exhibit considerable variation in binding-site geometry and domain composition (McNutt et al., 2024; Lam et al., 2024; Pourmirzaei et al., 2025).

In this setting, experiments were constructed using three DUD-E targets for which sufficiently low-similarity homologs were available. Appendix Table 9 summarizes the number of compounds utilized per target as a function of protein similarity. To enhance enrich-

Table 9: Summary of PubChem data for DUD-E targets with similarity below 0.5, including similarity ranges.

| Target | PDB code | Active Ligands | Decoy Ligands | PubChem Total (Act/Inact) | Similarity Range |
|--------|----------|----------------|---------------|---------------------------|------------------|
| EGFR | 2rgp | 541 | 35001 | 428 (413/15) | 0.272–0.294 |
| PLK1 | 2owb | 107 | 6794 | 391 (225/166) | 0.292–0.306 |
| SRC | 3el8 | 523 | 34407 | 974 (612/362) | 0.394–0.438 |

ment performance, all available active compounds were incorporated into the downstream propagation process.

## D.2 PU-Style Screening Setup on PU dataset

To evaluate the screening effectiveness of SubDyve in a realistic compound discovery scenario, we construct a dataset using molecules from the ZINC20 and PubChem databases. Specifically, we retrieve 10,082,034 compounds from ZINC20 (`https://zinc20.docking.org/tranches/home/`) and select CDK7 as the target protein. From PubChem, we obtain 1,744 unique compounds annotated for CDK7 after deduplication, of which 1,468 are labeled as active based on curated assay data.

To simulate sparse supervision, we randomly select 30% of the active compounds for $\mathcal{S}_{\text{train}}$. From this subset, we designate 10% as a held-out set $\mathcal{S}_2$ and use the remainder as initial seed nodes $\mathcal{S}_1$. The other 70% of actives are included in the screening graph as unlabeled nodes, emulating the presence of under-characterized actives in large-scale libraries. This setup ensures that the test set remains completely unseen and that the majority of actives do not contribute label information during propagation.

For fair comparison across network propagation (NP)-based baselines, we use the same seed sets across all runs, providing identical supervision to all models. We extract subgraph patterns using the SSM algorithm (Appendix B.2.1), excluding all test compounds to prevent leakage. A total of 100 discriminative patterns are used to construct subgraph-based fingerprints. To assess whether the difference in performance between methods was statistically significant, we applied the paired t-test over results from five independent runs. The hyperparameter settings follow Appendix Table 6, except the stratified split $N$ is set to 5.

## D.3 Details of Experimental Setup for Varying Seed Set Size

To demonstrate that SubDyve can effectively rank the 10% held-out set $\mathcal{S}_2$ specified in the PU-style screening setup created with the PU dataset with much fewer seeds, we conduct an ablation study with much fewer $\mathcal{S}_1$. Each seed count was randomly selected, and performance reported over five runs. This setup creates a much harsher situation where the test set is completely unseen and a much larger amount of active is prevented from contributing label information during propagation.

For fairness, we use the same number of seeds in the network propagation-based baseline and perform five runs on a randomly selected $\mathcal{S}_2$ as same in Appendix D.2. When extracting subgraph patterns with the SSM algorithm, we exclude all test compounds to prevent leakage.

## E Compute Resources and Time Profiling

**Training Environments.** Appendix Table 10 presents the system and software environment used for all experiments. The setup includes a high-memory server equipped with a single NVIDIA RTX A6000 GPU, dual Intel Xeon processors, and 512GB of RAM, running Ubuntu 22.04.4 with CUDA 11.1. All experiments are conducted on a single server.

**Time Profiling of Components in SubDyve.** Appendix Table 11 summarizes the per-module runtime of SubDyve across ten DUD-E targets. Subgraph mining is the dominant cost, requiring approximately 108 seconds per iteration, whereas similarity computation,

network construction, and propagation remain lightweight (sub-millisecond per pair or edge, and tens of seconds per graph). LFDR refinement, which includes a small GNN training loop, adds about 1.27 seconds per epoch on average. Notably, computing chemical similarity for one million compound pairs takes approximately 5 hours. *Note that* profiling time of LFDR-guided seed refinement integrates the GNN training time.

To contextualize the preprocessing overhead, Appendix Table 12 provides the end-to-end comparison against a standard RDKit+NP pipeline. While RDKit preprocessing averages 9.27 minutes per target, SubDyve's full network-building pipeline, comprised of SubDyve's subgraph mining, similarity computation, and graph construction, averages 17.88 minutes. The additional cost arises primarily from SubDyve's discriminative subgraph mining step, which extracts target-adaptive substructures that conventional fingerprint methods do not generate. In practice, this one-time mining overhead is comparable to the cumulative cost of evaluating multiple fingerprints in standard workflows, while enabling a more expressive and task-specific representation. Overall, SubDyve maintains practical preprocessing time while offering richer structural abstraction than traditional fingerprint-based pipelines.

Table 10: System and software environment.

| Component | Specification |
|---|---|
| GPU | $1 \times$ NVIDIA RTX A6000 (49GB) |
| CPU | Intel Xeon Gold 6248R @ 3.00GHz (48 cores) |
| RAM | 512 GB |
| OS | Ubuntu 22.04.4 LTS |
| CUDA | 11.1 |
| Python | 3.9.16 |
| PyTorch | 1.10.1 + cu111 |
| PyTorch Geometric | 2.0.4 |
| scikit-learn | 1.6.1 |
| scipy | 1.10.1 |

Table 11: Profiling time of each module.

| Module | Profiling Time |
|---|---|
| Subgraph Mining (SSM) | $108.55 \pm 15$ sec / iteration |
| Subgraph Pattern Similarity Computation | $0.4 \pm 0.2$ ms / compound pair |
| Subgraph fingerprint Network Construction | $0.1 \pm 0.1$ ms / edge |
| Network Propagation | $16 \pm 23$ sec / graph |
| Dynamic Seed Refinement with LFDR (incl. GNN training) | $1.27 \pm 0.56$ sec / epoch |

Table 12: End-to-end timing comparison of subgraph network building (SubDyve) versus baseline preprocessing (RDKit+NP) on DUD-E targets. Times are reported in minutes. SubDyve's network building includes subgraph mining and network construction (including similarity computation). RDKit preprocessing includes network construction (fingerprint generation and similarity computation).

| Target | SubDyve Network Building | | | RDKit |
|---|---|---|---|---|
| | Subgraph Mining | Network Construction | Total | Preprocessing |
| **Average** | 10.68 | 7.20 | **17.88** | **9.27** |

# F    Additional Experimental Results

## F.1    Zero-Shot Virtual Screening Results on Ten DUD-E Targets

### F.1.1    Comprehensive Evaluation

Appendix Table 13 shows a comprehensive evaluation of SubDyve and baseline methods across ten DUD-E targets. Metrics include AUROC, BEDROC, $EF_{1\%}$, $EF_{5\%}$, and $EF_{10\%}$, with confidence intervals estimated via 100 resampling trials. SubDyve achieves the best average performance across all five metrics, consistently outperforming other methods in early recognition and enrichment, while maintaining high AUROC. These results support the effectiveness of combining subgraph fingerprint network construction with LFDR-based refinement for zero-shot virtual screening.

Table 13: Comprehensive evaluation of SubDyve and baseline methods on ten DUD-E targets. Confidence intervals were estimated via bootstrapping (DiCiccio & Efron, 1996), using 100 resampled datasets to compute the standard deviations. AUROC and BEDROC are reported as percentages. The best and second-best scores per metric are in **bold** and underline, respectively.

| Protein Target | SubDyve (Ours) | | | | | PharmacoMatch | | | | | CDPKit | | | | | DrugCLIP | | | | |
|---|---|---|---|---|---|---|---|---|---|---|---|---|---|---|---|---|---|---|---|---|
| | AUROC | BEDROC | $EF_{1\%}$ | $EF_{5\%}$ | $EF_{10\%}$ | AUROC | BEDROC | $EF_{1\%}$ | $EF_{5\%}$ | $EF_{10\%}$ | AUROC | BEDROC | $EF_{1\%}$ | $EF_{5\%}$ | $EF_{10\%}$ | AUROC | BEDROC | $EF_{1\%}$ | $EF_{5\%}$ | $EF_{10\%}$ |
| ACES | 91±1 | 86±2 | 57.0±2.4 | 17.1±0.4 | 8.8±0.2 | 58±2 | 18±1 | 8.4±1.4 | 3.5±0.3 | 2.2±0.2 | 55±1 | 16±2 | 5.5±1.3 | 3.0±0.3 | 2.1±0.2 | 80±1 | 52±2 | 32.4±1.7 | 10.2±0.5 | 5.9±0.2 |
| ADA | 90±3 | 83±4 | 50.6±5.3 | 16.6±0.8 | 8.4±0.4 | 83±3 | 44±4 | 16.7±4.1 | 9.5±1.0 | 5.7±0.4 | 94±1 | 82±3 | 53.6±4.3 | 15.9±0.9 | 8.4±0.4 | 96±1 | 82±3 | 60.2±5.3 | 16.2±0.8 | 8.4±0.3 |
| ANDR | 87±2 | 72±2 | 37.1±2.1 | 14.8±0.5 | 8.0±0.2 | 76±1 | 33±2 | 15.8±1.9 | 6.0±0.5 | 4.3±0.3 | 71±2 | 26±2 | 12.6±2.1 | 4.4±0.5 | 3.7±0.3 | 91±1 | 64±3 | 34.3±2.4 | 12.7±0.6 | 7.5±0.3 |
| EGFR | 94±1 | 86±2 | 60.0±2.3 | 17.0±0.3 | 8.6±0.2 | 63±1 | 11±1 | 3.1±0.7 | 2.0±0.3 | 1.6±0.2 | 76±1 | 26±2 | 12.2±1.6 | 4.6±0.3 | 3.7±0.2 | 69±1 | 40±2 | 28.7±2.1 | 7.4±0.4 | 4.4±0.2 |
| FA10 | 79±1 | 58±2 | 46.8±1.7 | 10.6±0.4 | 5.5±0.2 | 47±1 | 1±1 | 0.2±0.2 | 0.1±0.1 | 0.2±0.1 | 55±1 | 6±1 | 0.0±0.0 | 0.7±0.2 | 1.2±0.1 | 94±1 | 86±1 | 51.2±1.8 | 17.0±0.3 | 9.1±0.1 |
| KIT | 82±2 | 44±3 | 13.8±2.6 | 11.3±0.7 | 6.1±0.4 | 56±2 | 4±1 | 0.0±0.0 | 0.4±0.2 | 0.7±0.2 | 63±2 | 9±2 | 1.1±0.8 | 1.2±0.4 | 1.8±0.3 | 30±3 | 10±2 | 5.2±1.7 | 1.8±0.5 | 1.2±0.3 |
| PLK1 | 94±2 | 85±3 | 51.7±4.0 | 17.7±0.6 | 9.0±0.3 | 62±3 | 9±2 | 1.5±1.3 | 0.7±0.3 | 1.8±0.3 | 75±3 | 39±3 | 5.7±2.3 | 10.2±0.9 | 5.5±0.5 | 88±2 | 66±4 | 45.0±4.0 | 12.8±0.9 | 7.3±0.4 |
| SRC | 82±1 | 61±2 | 35.0±1.8 | 11.3±0.4 | 6.9±0.2 | 79±1 | 27±1 | 6.0±1.0 | 5.3±0.4 | 4.6±0.2 | 80±1 | 28±1 | 11.1±1.2 | 5.3±0.4 | 4.3±0.2 | 59±2 | 16±1 | 8.1±1.3 | 2.9±0.3 | 2.0±0.2 |
| THRB | 78±1 | 61±2 | 36.6±2.0 | 11.9±0.5 | 6.0±0.2 | 70±1 | 22±1 | 5.9±1.0 | 4.8±0.4 | 3.3±0.2 | 79±1 | 35±2 | 11.8±1.5 | 7.2±0.4 | 4.5±0.2 | 97±0 | 83±1 | 46.9±1.7 | 17.2±0.3 | 9.3±0.1 |
| UROK | 55±3 | 37±3 | 25.6±2.4 | 8.0±0.7 | 4.1±0.3 | 60±2 | 4±1 | 0.6±0.7 | 0.5±0.2 | 0.4±0.2 | 91±1 | 55±3 | 24.5±2.8 | 10.4±0.9 | 8.2±0.4 | 93±1 | 73±3 | 48.1±3.1 | 14.7±0.7 | 8.1±0.3 |
| Avg. rank | 2.7 | 1.6 | 1.6 | 1.6 | 1.7 | 5.5 | 5.9 | 5.6 | 5.6 | 5.7 | 3.8 | 4.2 | 4.5 | 4.3 | 4.1 | 2.9 | 2.5 | 2.0 | 2.6 | 2.8 |
| Final rank | 1 | 1 | 1 | 1 | 1 | 7 | 7 | 7 | 7 | 7 | 4 | 4 | 4 | 4 | 4 | 2 | 2 | 2 | 2 | 2 |

| Protein Target | AutoDock Vina | | | | | ChemBERTa | | | | | MolFormer | | | | |
|---|---|---|---|---|---|---|---|---|---|---|---|---|---|---|---|
| | AUROC | BEDROC | $EF_{1\%}$ | $EF_{5\%}$ | $EF_{10\%}$ | AUROC | BEDROC | $EF_{1\%}$ | $EF_{5\%}$ | $EF_{10\%}$ | AUROC | BEDROC | $EF_{1\%}$ | $EF_{5\%}$ | $EF_{10\%}$ |
| ACES | 77±0.0 | 33±1.1 | 13.87±0.5 | 6.47±0.2 | 4.32±0.1 | 53±0 | 9±1 | 1.9±0.9 | 1.5±0.2 | 1.3±0.1 | 74±2 | 24±2 | 8.3±0.7 | 4.3±0.6 | 3.7±0.4 |
| ADA | 57±0.0 | 7±2.7 | 1.05±1.7 | 0.42±0.7 | 2.12±0.4 | 76±1 | 15±3 | 4.2±1.6 | 1.9±0.3 | 2.6±0.6 | 89±0 | 72±1 | 48.3±0.9 | 13.9±0.3 | 7.2±0.2 |
| ANDR | 64±0.0 | 34±1.2 | 18.41±0.6 | 6.89±0.3 | 4.07±0.2 | 39±0 | 5±1 | 1.9±0.4 | 0.9±0.2 | 0.8±0.2 | 56±1 | 9±1 | 3.0±0.1 | 1.6±0.3 | 1.5±0.3 |
| EGFR | 64±0.0 | 14±1.4 | 3.68±0.7 | 2.76±0.3 | 2.17±0.1 | 77±1 | 35±1 | 16.4±0.6 | 7.0±0.1 | 5.0±0.0 | 93±1 | 75±2 | 48.1±2.8 | 15.2±0.4 | 8.4±0.2 |
| FA10 | 84±0.0 | 41±1.7 | 15.77±0.8 | 7.28±0.3 | 5.05±0.2 | 73±1 | 28±3 | 12.9±1.6 | 5.4±0.5 | 3.4±0.2 | 93±0 | 66±0 | 36.7±0.4 | 13.0±0.2 | 7.6±0.1 |
| KIT | 78±0.0 | 18±2.4 | 2.97±1.9 | 3.23±0.5 | 3.11±0.3 | 62±1 | 16±1 | 4.9±3.7 | 2.8±0.0 | 2.5±0.1 | 93±0 | 66±1 | 36.8±0.9 | 13.6±0.4 | 7.7±0.4 |
| PLK1 | 64±0.0 | 13±1.8 | 1.83±0.3 | 1.85±0.3 | 2.22±0.4 | 60±3 | 15±1 | 4.9±1.4 | 2.9±0.2 | 1.9±0.3 | 89±1 | 69±0 | 35.2±4.0 | 14.4±0.0 | 8.0±0.1 |
| SRC | 66±0.0 | 13±1.2 | 4.00±0.5 | 2.36±0.2 | 1.96±0.1 | 64±2 | 15±1 | 3.3±0.7 | 3.1±0.2 | 2.6±0.4 | 82±1 | 48±1 | 21.5±1.5 | 10.4±0.4 | 6.5±0.1 |
| THRB | 81±0.0 | 25±1.8 | 4.31±1.0 | 4.80±0.3 | 3.98±0.2 | 79±0 | 34±2 | 14.5±2.2 | 6.3±0.4 | 4.7±0.1 | 59±1 | 6±1 | 1.2±0.1 | 0.9±0.3 | 0.9±0.1 |
| UROK | 80±0.0 | 28±1.3 | 7.90±0.7 | 5.88±0.3 | 3.92±0.2 | 62±3 | 5±1 | 0.6±0.0 | 0.3±0.1 | 1.0±0.3 | 79±3 | 36±2 | 10.0±1.5 | 7.6±0.5 | 5.2±0.4 |
| Avg. rank | 4.2 | 4.9 | 5.1 | 4.9 | 4.9 | 5.4 | 5.4 | 5.4 | 5.4 | 5.2 | 3.3 | 3.4 | 3.7 | 3.4 | 3.3 |
| Final rank | 5 | 5 | 5 | 5 | 5 | 6 | 6 | 6 | 6 | 6 | 3 | 3 | 3 | 3 | 3 |

In addition, recent advances in protein–ligand prediction increasingly leverage explicit 3D structural information. To complement the main results and to investigate SubDyve against such structure-aware approaches, we further evaluated two representative 3D methods, DiffDock-NMDN (Xia et al., 2025) and DynamicBind (Lu et al., 2024), under the same DUD-E evaluation protocol.

As shown in Appendix Table 14, across ten DUD-E targets SubDyve consistently achieved substantially higher BEDROC and $EF_{1\%}$ scores than both 3D generative approaches and the physics-based AutoDock-Vina scoring function. While AutoDock-Vina provides moderate enrichment, both DiffDock-NMDN and DynamicBind exhibit notably low early-recognition performance under this evaluation setting.

A likely explanation is the mismatch between the objectives used to train modern 3D generative models and the requirements of ligand ranking. Most structure-aware deep models are optimized for accurate pose generation—rewarding geometrically plausible arrangements of ligand and protein atoms—rather than for discriminating true binders from decoys. As a result, these models often assign high confidence to plausible poses even for non-binding decoys, yielding weak enrichment. This gap between pose-generation accuracy and screening effectiveness has also been observed in recent analyses of 3D docking and diffusion-based models (Li et al., 2025; Buttenschoen et al., 2024).

Moreover, DUD-E decoys are intentionally constructed to mimic drug-like physicochemical properties, and the docked complexes provided in DUD-E contain non-negligible geometric noise. Structure-aware generative models, which are trained on curated datasets such as PDBbind, tend to be more sensitive to such noise, whereas physics-based scoring functions

Table 14: Performance comparison including 3D structure-aware model on the ten DUD-E targets. The top results are shown in **bold**, and the second-best are underlined, respectively. Confidence intervals are reported with 100 bootstrap resamples (DiCiccio & Efron, 1996).

| Protein Target | SubDyve (Ours) | | DiffDock-NMDN (Xia et al., 2025) | | DynamicBind (Lu et al., 2024) | | AutoDock Vina (Eberhardt et al., 2021) | |
|---|---|---|---|---|---|---|---|---|
| | BEDROC | $EF_{1\%}$ | BEDROC | $EF_{1\%}$ | BEDROC | $EF_{1\%}$ | BEDROC | $EF_{1\%}$ |
| ACES | **86±2** | **57.0±2.4** | 4±1 | 1.0±0.4 | 11±1 | 3.8±0.9 | 33±1 | 13.9±0.5 |
| ADA | **83±4** | **50.6±5.3** | 20±2 | 3.9±1.2 | 9±2 | 0.9±1.1 | 7±2 | 1.1±1.7 |
| ANDR | **72±2** | **37.1±2.1** | 7±1 | 0.8±0.4 | 8±1 | 0.7±0.5 | 34±1 | 18.4±0.6 |
| EGFR | **86±2** | **60.0±2.3** | 5±1 | 0.5±0.2 | 17±1 | 1.7±0.6 | 14±1 | 3.7±0.7 |
| FA10 | **58±2** | **46.8±1.7** | 3±0 | 0.5±0.2 | 11±1 | 0.0±0.0 | 41±1 | 15.8±0.8 |
| KIT | **44±3** | **13.8±2.6** | 5±1 | 1.2±0.7 | 9±2 | 1.0±0.7 | 18±2 | 3.0±1.9 |
| PLK1 | **85±3** | **51.7±4.0** | 6±1 | 0.9±0.8 | 5±1 | 0.0±0.0 | 13±1 | 1.8±0.3 |
| SRC | **61±2** | **35.0±1.8** | 6±1 | 0.4±0.2 | 25±1 | 6.0±1.0 | 13±1 | 4.0±0.5 |
| THRB | **61±2** | **36.6±2.0** | 2±0 | 0.7±0.2 | 4±1 | 0.4±0.3 | 25±1 | 4.3±1.0 |
| UROK | **37±3** | **25.6±2.4** | 5±1 | 1.1±0.5 | 4±1 | 0.7±0.6 | 28±1 | 7.9±0.7 |
| Avg. rank | 1.0 | 1.0 | 3.6 | 3.2 | 3.0 | 3.6 | 2.4 | 2.2 |

Table 15: Performance comparison across EGFR, PLK1, and SRC with similarity below 0.5. AUROC and BEDROC are reported as percentages. Confidence intervals are estimated via bootstrapping (DiCiccio & Efron, 1996), using 100 resampled datasets to compute the standard deviations.

| | EGFR | | PLK1 | | SRC | |
|---|---|---|---|---|---|---|
| Model | BEDROC | $EF_{1\%}$ | BEDROC | $EF_{1\%}$ | BEDROC | $EF_{1\%}$ |
| SubDyve$_{0.9}$ | **86 ± 2** | **60.0 ± 2.3** | **85 ± 3** | 51.7 ± 4.0 | **61 ± 2** | **35.0 ± 1.8** |
| SubDyve$_{0.5}$ | 79 ± 2 | 60.0 ± 2.2 | 78 ± 4 | **60.6 ± 3.6** | 41 ± 2 | 16.2 ± 1.6 |
| PharmacoMatch | 11 ± 1 | 3.1 ± 0.7 | 9 ± 2 | 1.5 ± 1.3 | 27 ± 1 | 6.0 ± 1.0 |
| CDPKIT | 26 ± 2 | 12.2 ± 1.6 | 39 ± 3 | 5.7 ± 2.3 | 28 ± 1 | 11.1 ± 1.2 |
| DrugCLIP | 40 ± 2 | 28.7 ± 2.1 | 66 ± 4 | 45.0 ± 4.0 | 16 ± 2 | 8.1 ± 1.3 |
| MoLFormer | 75 ± 2 | 48.1 ± 2.8 | 69 ± 0 | 35.2 ± 4.0 | 48 ± 1 | 21.5 ± 1.5 |
| AutoDock Vina | 14 ± 1 | 3.68 ± 0.7 | 13 ± 1 | 1.83 ± 0.3 | 13 ± 1 | 4.00 ± 0.5 |

like AutoDock-Vina exhibit greater robustness—explaining its relatively better performance compared to learned 3D models.

Taken together, these observations suggest that current 3D generative pipelines may require additional calibration or hybrid scoring strategies to achieve reliable ligand ranking under this experiment setting. A deeper examination of structure-aware models remains an important direction for future work.

### F.1.2 ROBUSTNESS TO DISTANT HOMOLOGS

Based on the experiment setting in Appendix D.1 "Robustness to distant homologs Setup", Appendix Table 15 summarizes the performance of SubDyve and baseline models under the restricted similarity condition. Even under this more challenging setting with less informative seeds, SubDyve maintained strong early-recognition performance and remained better or competitive with all baselines on both BEDROC and EF1%.

Notably, for EGFR, the BEDROC score decreases by less than seven points relative to the 0.9 sequence-identity setting, while still maintaining a large performance gap over pharmacophore-based, FM-based and deep learning-based baselines. For PLK1, performance remains stable under the restricted similarity threshold and exceeds the results observed at 0.9 similarity. These observations reinforce that SubDyve remains effective even when only distant homologs with sparse annotations are available.

### F.1.3 MULTI-TARGET SCALABILITY EVALUATION ON THREE DUD-E KINASES

To investigate whether SubDyve can amortize preprocessing costs across biologically related targets, we conducted a multi-target evaluation using three tyrosine kinases from DUD-E: EGFR, KIT, and SRC. These targets share regulatory roles in signaling pathways and possess overlapping ligand–protein annotation profiles (Tomuleasa et al., 2024; Tatton et al., 2003; Rao et al., 2015). Leveraging this structure, we identified homologous proteins common to the three targets using PubChem annotations and collected the corresponding compounds.

From this joint compound set, we performed a single supervised subgraph-mining step, extracted class-discriminative patterns, and constructed a unified subgraph-based fingerprint

network that was reused for all analyses. SubDyve's network propagation with LFDR refinement was then applied under four activity definitions: (i) ligands active across all three targets, and (ii–iv) ligands active across each target pair (EGFR–KIT, EGFR–SRC, KIT–SRC). We limited the number of seeds for multi-target to 189 to account for multi-target scalability and low-label regimes. The number of known active compounds targeting the tyrosine kinases is limited: EGFR–SRC and KIT–SRC share 27 and 15 compounds, respectively, while only three compounds are active across all three targets and EGFR–KIT. All other hyperparameter settings are configured identically to the single-target DUD-E experiments.

Across all configurations, SubDyve consistently demonstrated strong early-recognition performance and substantially outperformed DrugCLIP on BEDROC and EF metrics (Appendix Table 16). These results indicate that subgraph patterns mined once from shared homologous annotations can transfer across related kinase targets, suggesting that SubDyve's preprocessing cost can be partially amortized when targets exhibit biological or ligand-based similarity.

While this experiment does not constitute full multi-target joint training nor address scalability for highly heterogeneous target sets, it provides evidence that subgraph-level representations extracted by SubDyve remain informative across related protein families. Extending the approach toward broader transfer-learning or multi-target VS frameworks is a promising direction for future work.

Table 16: Comparison of SubDyve (Ours) and DrugCLIP across multi target combinations.

| Combination | SubDyve (Ours) | | | | DrugCLIP | | | |
|---|---|---|---|---|---|---|---|---|
| | BEDROC | $EF_{1\%}$ | $EF_{5\%}$ | $EF_{10\%}$ | BEDROC | $EF_{1\%}$ | $EF_{5\%}$ | $EF_{10\%}$ |
| EGFR-KIT-SRC | **69.52** | **35.93** | **19.38** | **9.69** | 4.55 | 0.00 | 0.00 | 2.22 |
| EGFR-KIT | **79.37** | **57.27** | **18.18** | **9.09** | 5.19 | 0.00 | 0.00 | 1.66 |
| EGFR-SRC | **84.11** | **41.21** | **19.20** | **9.99** | 13.14 | 7.26 | 2.95 | 1.66 |
| KIT-SRC | **55.76** | **20.25** | **14.40** | **8.81** | 1.66 | 0.00 | 0.00 | 0.33 |

Table 17: Complete performance comparison of SubDyve and baselines on the PU dataset. The top results are shown in **bold**, and the second-best are underlined, respectively.

| Method | BEDROC (%) | EF | | | | |
|---|---|---|---|---|---|---|
| | | 0.5% | 1% | 3% | 5% | 10% |
| **Deep learning-based** | | | | | | |
| BIND (BIB, 2024) (Lam et al., 2024) | - | - | - | - | - | 0.04 ± 0.08 |
| AutoDock Vina (J. Chem. Inf. Model.) (Eberhardt et al., 2021) | 1.0 ± 1.3 | - | 0.2 ± 0.3 | 0.6 ± 0.7 | 1.1 ± 0.6 | 1.2 ± 0.5 |
| DrugCLIP (NeurIPS) (Gao et al., 2023) | 2.7 ± 1.26 | 1.63 ± 1.99 | 1.63 ± 0.81 | 2.45 ± 1.02 | 2.53 ± 1.35 | 2.69 ± 0.62 |
| PSICHIC (Nat MI) (Koh et al., 2024) | 9.37 ± 3.08 | 4.07 ± 2.58 | 6.92 ± 3.30 | 7.48 ± 2.47 | 7.02 ± 1.80 | 5.35 ± 0.94 |
| GRAB (ICDM) (Yoo et al., 2021) | 40.68 ± 10.60 | 44.22 ± 8.35 | 45.21 ± 5.63 | 29.78 ± 1.38 | 18.69 ± 0.47 | **10.00 ± 0.00** |
| **Data mining-based** | | | | | | |
| avalon + NP (Yi et al., 2023) | 77.59 ± 1.72 | 135.76 ± 6.44 | 87.58 ± 2.9 | 31.55 ± 0.54 | 19.67 ± 0.4 | 9.88 ± 0.16 |
| cdk-substructure + NP (Yi et al., 2023) | 66.56 ± 2.89 | 125.4 ± 11.28 | 69.67 ± 2.98 | 28.15 ± 0.92 | 17.22 ± 0.79 | 9.22 ± 0.42 |
| estate + NP (Yi et al., 2023) | 52.44 ± 6.19 | 94.4 ± 13.68 | 57.87 ± 7.15 | 22.71 ± 2.7 | 15.92 ± 0.85 | 8.24 ± 0.38 |
| extended + NP (Yi et al., 2023) | 73.7 ± 3.3 | 136.73 ± 6.83 | 83.54 ± 5.21 | 31.28 ± 0.97 | 18.85 ± 0.55 | 9.63 ± 0.2 |
| fp2 + NP (Yi et al., 2023) | 72.68 ± 3.77 | 129.06 ± 11.89 | 85.49 ± 3.89 | 30.86 ± 0.69 | 18.69 ± 0.6 | 9.51 ± 0.36 |
| fp4 + NP (Yi et al., 2023) | 69.62 ± 3.69 | 122.76 ± 13.02 | 75.01 ± 4.21 | 28.96 ± 1.34 | 18.36 ± 1.0 | 9.59 ± 0.29 |
| graph + NP (Yi et al., 2023) | 75.86 ± 3.99 | 126.72 ± 10.05 | 84.73 ± 3.74 | 31.68 ± 0.92 | 19.1 ± 0.47 | 9.75 ± 0.24 |
| hybridization + NP (Yi et al., 2023) | 75.4 ± 5.18 | 135.15 ± 17.78 | 80.25 ± 5.88 | 31.14 ± 1.0 | 18.69 ± 0.6 | 9.63 ± 0.15 |
| maccs + NP (Yi et al., 2023) | 75.44 ± 4.85 | 135.72 ± 12.7 | 79.82 ± 4.76 | 31.0 ± 1.41 | 18.93 ± 0.66 | 9.67 ± 0.21 |
| pubchem + NP (Yi et al., 2023) | 63.48 ± 5.16 | 99.17 ± 10.17 | 69.3 ± 7.08 | 30.87 ± 1.27 | 18.77 ± 0.9 | 9.84 ± 0.15 |
| rdkit + NP (Yi et al., 2023) | 79.04 ± 1.96 | 148.69 ± 4.25 | 89.24 ± 2.08 | 31.68 ± 0.92 | 19.02 ± 0.55 | 9.55 ± 0.3 |
| standard + NP (Yi et al., 2023) | 72.42 ± 3.51 | 121.97 ± 15.51 | 84.34 ± 5.56 | 31.27 ± 0.96 | 19.01 ± 0.33 | 9.71 ± 0.24 |
| SubDyve | **83.44 ± 1.44** | **155.31 ± 6.38** | **97.59 ± 1.44** | **33.01 ± 0.60** | **19.90 ± 0.18** | **10.00 ± 0.00** |
| Statistical Significance (p-value) | ** | - | ** | * | - | - |

Table 18: Description of various molecular fingerprints used in virtual screening.

| Fingerprint | Description |
|---|---|
| Standard | Based on the presence or absence of specific functional groups or atoms in a molecule. Simple and efficient but may lack specificity. |
| Extended | Similar to standard fingerprints but include additional features such as bond counts and stereochemistry. |
| Graph | Derived from the topological structure of a molecule; includes atom/bond counts, ring sizes, and branching patterns. |
| MACCS | A set of 166 predefined molecular keys from the MACCS project indicating presence/absence of specific substructures. |
| PubChem | Developed by NIH; based on predefined substructure paths in a molecule. |
| Estate | Encodes topological and electrostatic properties of a molecule. |
| Hybridization | Encodes the hybridization states of atoms in a molecule. |
| CDK-substructure | Captures the presence or absence of specific chemical substructures. |
| RDKit | Fingerprints generated using the RDKit toolkit; used for similarity searches and cheminformatics applications. |
| Avalon | Path-based fingerprints representing features derived from atomic paths within molecules. |
| FP2 | Developed by OpenEye; uses topological and pharmacophoric information for similarity search and screening. |
| FP4 | Also from OpenEye; incorporates topological, pharmacophoric, and electrostatic features for molecular comparison. |

## F.2 PU-Style Screening Results on PU dataset

To supplement the results in Section 4.1.2, we report additional baseline results for the PU-style virtual screening experiment on the PU dataset. Table 17 extends Table 2 by including all evaluated general-purpose molecular fingerprints. These are combined with the same network propagation pipeline as in SubDyve, allowing a controlled comparison of representation effectiveness. Descriptions of the 12 fingerprints used are provided in Table 18.

As shown in Table 17, SubDyve achieves the best performance across all BEDROC and EF metrics, outperforming deep learning models and general fingerprint-based baselines. While some fingerprints (e.g., rdkit, Graph) perform competitively under certain thresholds, they fall short in consistency across metrics. These results support the advantage of task-specific subgraph representations combined with uncertainty-aware refinement for robust screening under sparse supervision.

### F.3 ABLATION STUDY: IMPACT OF SUBGRAPH PATTERN FINGERPRINT NETWORK AND LFDR-GUIDED SEED REFINEMENT ON TEN DUD-E TARGETS

We conduct an ablation study on 10 DUD-E targets to evaluate the individual and joint contributions of two core components of SubDyve: (1) the subgraph-based similarity network and (2) the LFDR-based seed refinement. In Table 19, combining both components yields the best results, with the highest $EF_{1\%}$ on all 10 targets and top BEDROC scores on 9. This highlights SubDyve's robustness beyond the PU dataset and its screening performance on DUD-E targets.

We also observed that applying LFDR refinement alone without the subgraph-based similarity network often degrades or remains unchanged, while using both components together consistently improves performance. This finding highlights the complementarity of chemically meaningful network construction and uncertainty-awareness, both essential for robust and generalizable virtual screening under low supervision.

Table 19: Ablation study results for the effect of subgraph fingerprint network and LFDR-guided seed refinement on the 10 DUD-E dataset. The top results are shown in **bold**, and the second-best are underlined, respectively.

| Target | Subgraph | LFDR | BEDROC | $EF_{1\%}$ |
|---|---|---|---|---|
| ACES | | | $64 \pm 2$ | $38.7 \pm 2.4$ |
| | | ✓ | $62 \pm 2$ | $38.5 \pm 2.0$ |
| | ✓ | | $76 \pm 1$ | $35.7 \pm 1.7$ |
| | ✓ | ✓ | **$86 \pm 2$** | **$57.0 \pm 2.4$** |
| ADA | | | $87 \pm 2$ | $41.1 \pm 4.1$ |
| | | ✓ | **$87 \pm 2$** | $45.2 \pm 4.1$ |
| | ✓ | | $76 \pm 3$ | $36.3 \pm 4.9$ |
| | ✓ | ✓ | $83 \pm 4$ | **$50.6 \pm 5.3$** |
| ANDR | | | $27 \pm 3$ | $18.9 \pm 2.4$ |
| | | ✓ | $24 \pm 2$ | $18.4 \pm 2.1$ |
| | ✓ | | $45 \pm 3$ | $23.1 \pm 2.7$ |
| | ✓ | ✓ | **$72 \pm 2$** | **$37.1 \pm 2.1$** |
| EGFR | | | $40 \pm 2$ | $30.9 \pm 1.7$ |
| | | ✓ | $33 \pm 2$ | $18.2 \pm 1.6$ |
| | ✓ | | $79 \pm 2$ | $53.2 \pm 2.0$ |
| | ✓ | ✓ | **$86 \pm 2$** | **$60.0 \pm 2.3$** |
| FA10 | | | $17 \pm 2$ | $11.8 \pm 1.3$ |
| | | ✓ | $6 \pm 1$ | $1.0 \pm 0.4$ |
| | ✓ | | $56 \pm 2$ | $46.8 \pm 2.0$ |
| | ✓ | ✓ | **$58 \pm 2$** | **$47.0 \pm 1.7$** |
| KIT | | | $11 \pm 2$ | $3.7 \pm 1.5$ |
| | | ✓ | $11 \pm 2$ | $2.9 \pm 1.3$ |
| | ✓ | | $37 \pm 3$ | $5.8 \pm 1.9$ |
| | ✓ | ✓ | **$44 \pm 3$** | **$13.8 \pm 2.6$** |
| PLK1 | | | $61 \pm 4$ | $43.2 \pm 4.4$ |
| | | ✓ | $57 \pm 5$ | $32.2 \pm 3.6$ |
| | ✓ | | $78 \pm 3$ | $49.5 \pm 4.7$ |
| | ✓ | ✓ | **$85 \pm 3$** | **$51.7 \pm 4.0$** |
| SRC | | | $56 \pm 2$ | $28.5 \pm 1.7$ |
| | | ✓ | $39 \pm 2$ | $12.6 \pm 1.4$ |
| | ✓ | | $25 \pm 2$ | $9.4 \pm 1.3$ |
| | ✓ | ✓ | **$61 \pm 2$** | **$35.0 \pm 1.8$** |
| THRB | | | $28 \pm 2$ | $20.3 \pm 1.9$ |
| | | ✓ | $21 \pm 2$ | $10.7 \pm 1.4$ |
| | ✓ | | $32 \pm 2$ | $21.2 \pm 1.7$ |
| | ✓ | ✓ | **$61 \pm 2$** | **$36.6 \pm 2.0$** |
| UROK | | | $35 \pm 3$ | $22.2 \pm 2.9$ |
| | | ✓ | $30 \pm 3$ | $13.0 \pm 2.6$ |
| | ✓ | | $30 \pm 3$ | $11.1 \pm 2.6$ |
| | ✓ | ✓ | **$37 \pm 3$** | **$25.6 \pm 2.4$** |

### F.4 ABLATION STUDY: VARYING SEED SET SIZES

To provide a more comprehensive evaluation of PU-style virtual screening on the PU dataset, we present additional baseline results for the number of seeds (50, 150, 250) in Table 20, which expands upon the findings reported in Section 4.2.2 and Table 4. This extended table includes a wider range of general-purpose molecular fingerprints, each integrated into the same network propagation framework used by SubDyve, ensuring a fair and controlled comparison of representational capabilities. Additionally, we introduce Subgraph + NP as a control variant, applying standard propagation over subgraph-derived networks without LFDR-based refinement.

Across all seed sizes, SubDyve consistently achieves superior performance, particularly in BEDROC, $EF_{3\%}$, and $EF_{5\%}$. Subgraph + NP advantage also extends to $EF_{1\%}$, highlighting the strength of subgraph-based representations in capturing bioactive chemical features beyond those accessible to general fingerprints.

Although certain baselines—such as MACCS and Avalon—exhibit strong results at specific enrichment thresholds, their performance lacks consistency across evaluation metrics, demonstrating the robustness of SubDyve's approach. These results suggest that subgraph patterns and LFDR-based refinement have screening power over other pre-defined fingerprints, even in harsh environments with much sparser seed.

Beyond the settings considered in the main text, we further examined the extreme case in which only very small seed sets are available (5, 10, or 15 compounds). As summarized in Table 21, SubDyve maintains competitive early-recognition quality even at these minimal seed sizes, consistently outperforming all generic fingerprints integrated with the same propagation framework. Notably, while overall performance naturally decreases as the seed set becomes extremely sparse, subgraph-driven representations continue to provide meaningful enrichment, and SubDyve remains the top-performing method across most metrics and seed sizes. These results indicate that the structural patterns extracted by subgraph mining retain useful discriminative power even under highly limited supervision, and that LFDR-guided refinement further stabilizes ranking performance in such challenging regimes.

Table 20: Ablation study on the number of seed compounds on the PU dataset. For each seed size (50, 150, 250), the baseline of all generic fingerprint performance is shown. For each number, the best value is highlighted in **bold**, and the second-best is underlined.

| No. of Seeds | Method | BEDROC (%) | EF | | | | |
|---|---|---|---|---|---|---|---|
| | | | 0.30% | 0.50% | 1% | 3% | 5% |
| 50 | avalon + NP (Yi et al., 2023) | 46.18 ± 3.95 | 54.02 ± 15.47 | 52.83 ± 12.07 | 48.9 ± 7.93 | 28.96 ± 0.68 | 18.28 ± 0.87 |
| | cdk-substructure + NP (Yi et al., 2023) | 40.61 ± 3.4 | 59.58 ± 8.86 | 53.72 ± 7.0 | 42.36 ± 5.22 | 22.85 ± 1.4 | 14.85 ± 0.95 |
| | estate + NP (Yi et al., 2023) | 34.87 ± 3.38 | 37.8 ± 15.17 | 39.92 ± 11.07 | 37.51 ± 4.77 | 20.53 ± 2.29 | 13.87 ± 0.63 |
| | extended + NP (Yi et al., 2023) | 44.74 ± 4.41 | 36.61 ± 10.1 | 47.19 ± 10.52 | 49.29 ± 11.34 | 27.73 ± 0.79 | 17.55 ± 0.77 |
| | fp2 + NP (Yi et al., 2023) | 43.51 ± 5.4 | 39.32 ± 13.17 | 43.07 ± 11.06 | 47.64 ± 10.63 | 27.2 ± 0.61 | 17.06 ± 0.6 |
| | fp4 + NP (Yi et al., 2023) | 40.46 ± 3.45 | 54.11 ± 12.07 | 52.82 ± 7.24 | 42.38 ± 4.73 | 22.98 ± 1.98 | 15.59 ± 1.11 |
| | graph + NP (Yi et al., 2023) | 45.08 ± 4.62 | 51.23 ± 18.33 | 55.28 ± 9.08 | 49.34 ± 9.06 | 27.06 ± 1.84 | 16.81 ± 0.87 |
| | hybridization + NP (Yi et al., 2023) | 43.76 ± 4.14 | 41.99 ± 22.79 | 51.19 ± 14.22 | 48.49 ± 8.79 | 26.11 ± 1.03 | 16.89 ± 0.76 |
| | maccs + NP (Yi et al., 2023) | 47.02 ± 3.83 | 56.77 ± 15.24 | 52.81 ± 9.24 | 50.92 ± 3.15 | 27.74 ± 2.04 | 17.05 ± 1.2 |
| | pubchem + NP (Yi et al., 2023) | 41.13 ± 4.46 | 44.69 ± 14.09 | 45.51 ± 7.91 | 41.97 ± 6.91 | 25.7 ± 1.99 | 17.14 ± 1.0 |
| | rdkit + NP (Yi et al., 2023) | 43.85 ± 3.37 | 39.21 ± 19.76 | 47.92 ± 11.32 | 50.55 ± 3.96 | 25.7 ± 1.89 | 15.59 ± 1.08 |
| | standard + NP (Yi et al., 2023) | 44.64 ± 6.02 | 46.13 ± 13.78 | 47.94 ± 13.87 | 48.47 ± 9.85 | 27.46 ± 1.1 | 17.55 ± 0.73 |
| | Subgraph + NP | 46.33 ± 1.26 | 37.79 ± 21.22 | 31.81 ± 12.68 | **53.93 ± 4.97** | 27.61 ± 1.47 | 17.27 ± 0.51 |
| | SubDyve | **51.78 ± 3.38** | **69.5 ± 11.81** | **62.53 ± 14.84** | 52.66 ± 5.91 | **29.48 ± 2.37** | **18.15 ± 0.90** |
| 150 | avalon + NP (Yi et al., 2023) | 54.73 ± 2.42 | 65.0 ± 15.73 | 70.85 ± 9.83 | 60.72 ± 4.7 | 31.0 ± 0.55 | 19.59 ± 0.37 |
| | cdk-substructure + NP (Yi et al., 2023) | 48.25 ± 3.74 | 75.75 ± 9.08 | 66.61 ± 10.79 | 53.76 ± 7.26 | 25.7 ± 1.09 | 16.48 ± 0.91 |
| | estate + NP (Yi et al., 2023) | 40.42 ± 5.07 | 51.37 ± 17.43 | 48.78 ± 2.63 | 47.69 ± 10.35 | 21.48 ± 3.35 | 14.28 ± 2.02 |
| | extended + NP (Yi et al., 2023) | 51.87 ± 3.8 | 55.4 ± 6.54 | 56.87 ± 12.75 | 60.28 ± 5.39 | 30.18 ± 1.52 | 18.53 ± 0.76 |
| | fp2 + NP (Yi et al., 2023) | 50.99 ± 5.85 | 47.39 ± 15.42 | 56.12 ± 15.32 | 59.05 ± 7.79 | 29.24 ± 1.14 | 18.12 ± 0.71 |
| | fp4 + NP (Yi et al., 2023) | 48.8 ± 3.46 | 74.15 ± 6.03 | 62.71 ± 8.39 | 53.38 ± 7.34 | 26.78 ± 1.63 | 17.38 ± 0.55 |
| | graph + NP (Yi et al., 2023) | 52.85 ± 5.55 | 76.98 ± 15.73 | 70.09 ± 16.19 | 54.23 ± 8.76 | 29.78 ± 1.79 | 18.36 ± 0.77 |
| | hybridization + NP (Yi et al., 2023) | 52.69 ± 5.27 | 70.17 ± 24.31 | 69.22 ± 19.13 | 57.05 ± 8.56 | 28.55 ± 0.97 | 17.79 ± 0.33 |
| | maccs + NP (Yi et al., 2023) | 55.22 ± 4.39 | **79.99 ± 15.80** | 71.65 ± 13.30 | 60.69 ± 6.59 | 30.6 ± 1.29 | 18.85 ± 0.48 |
| | pubchem + NP (Yi et al., 2023) | 46.74 ± 5.38 | 62.37 ± 19.0 | 58.63 ± 11.73 | 48.9 ± 7.76 | 28.01 ± 1.63 | 18.44 ± 0.7 |
| | rdkit + NP (Yi et al., 2023) | 50.82 ± 3.79 | 52.69 ± 6.75 | 54.62 ± 10.48 | 54.62 ± 7.24 | 29.5 ± 1.59 | 17.79 ± 0.95 |
| | standard + NP (Yi et al., 2023) | 51.59 ± 4.93 | 60.85 ± 11.29 | 63.42 ± 13.66 | 55.39 ± 8.09 | 29.78 ± 1.85 | 18.61 ± 0.75 |
| | Subgraph + NP | 55.08 ± 1.52 | 44.39 ± 22.83 | 61.29 ± 10.07 | **67.17 ± 7.24** | 30.07 ± 1.38 | 18.22 ± 0.93 |
| | SubDyve | **59.07 ± 2.25** | 74.67 ± 7.46 | **73.55 ± 10.51** | 66.72 ± 5.29 | **32.26 ± 1.04** | **19.73 ± 0.36** |
| 250 | avalon + NP (Yi et al., 2023) | 61.29 ± 2.44 | 97.18 ± 13.25 | **86.96 ± 9.16** | 68.05 ± 4.42 | 31.14 ± 0.52 | 19.51 ± 0.48 |
| | cdk-substructure + NP (Yi et al., 2023) | 54.07 ± 4.05 | 95.87 ± 20.51 | 81.39 ± 13.84 | 61.09 ± 7.94 | 26.52 ± 0.43 | 16.97 ± 0.91 |
| | estate + NP (Yi et al., 2023) | 44.34 ± 5.83 | 64.97 ± 13.25 | 66.81 ± 12.27 | 50.14 ± 8.31 | 22.16 ± 2.67 | 15.18 ± 1.32 |
| | extended + NP (Yi et al., 2023) | 57.48 ± 3.71 | 64.79 ± 10.11 | 75.67 ± 10.89 | 64.35 ± 5.22 | 30.99 ± 1.26 | 18.85 ± 0.54 |
| | fp2 + NP (Yi et al., 2023) | 56.88 ± 5.26 | 67.45 ± 16.53 | 75.57 ± 15.28 | 65.15 ± 8.3 | 30.19 ± 1.26 | 18.52 ± 0.61 |
| | fp4 + NP (Yi et al., 2023) | 55.04 ± 3.77 | 91.76 ± 18.18 | 81.27 ± 12.86 | 62.75 ± 6.11 | 27.33 ± 0.66 | 18.03 ± 0.79 |
| | graph + NP (Yi et al., 2023) | 58.68 ± 5.4 | 93.5 ± 19.79 | 78.84 ± 12.62 | 62.79 ± 9.89 | 30.19 ± 1.75 | 18.44 ± 0.83 |
| | hybridization + NP (Yi et al., 2023) | 58.94 ± 4.32 | 99.75 ± 15.48 | 87.76 ± 17.54 | 65.2 ± 8.07 | 30.05 ± 0.8 | 18.36 ± 0.26 |
| | maccs + NP (Yi et al., 2023) | 60.94 ± 4.57 | 102.66 ± 15.71 | 84.48 ± 12.88 | 67.21 ± 6.9 | 30.6 ± 1.67 | 19.01 ± 0.66 |
| | pubchem + NP (Yi et al., 2023) | 51.92 ± 5.47 | 71.7 ± 15.79 | 73.95 ± 17.13 | 55.82 ± 8.04 | 29.78 ± 1.51 | 18.69 ± 0.87 |
| | rdkit + NP (Yi et al., 2023) | 58.4 ± 2.09 | 70.29 ± 7.84 | 70.65 ± 9.75 | 68.89 ± 5.38 | 30.59 ± 1.14 | 18.52 ± 0.76 |
| | standard + NP (Yi et al., 2023) | 57.08 ± 4.39 | 71.4 ± 15.11 | 76.42 ± 18.16 | 61.09 ± 8.56 | 31.0 ± 1.26 | 19.02 ± 0.42 |
| | Subgraph + NP | 61.96 ± 3.24 | 41.01 ± 13.89 | 86.31 ± 11.97 | **80.31 ± 4.60** | 30.20 ± 1.44 | 18.49 ± 0.85 |
| | SubDyve | **66.73 ± 2.71** | **97.69 ± 16.55** | 85.44 ± 12.82 | 78.19 ± 3.38 | **32.85 ± 0.60** | **19.72 ± 0.36** |

Table 21: Ablation study on the small number of seed compounds (5, 10, 15) on the PU dataset. For each seed size, the baseline of all generic fingerprint performance is shown. For each number, the best value is highlighted in **bold**, and the second-best is underlined.

| No. of Seeds | Method | BEDROC (%) | EF | | | | |
|---|---|---|---|---|---|---|---|
| | | | 0.30% | 0.50% | 1% | 3% | 5% |
| 5 | avalon + NP (Yi et al., 2023) | 29.22 ± 3.32 | 13.54 ± 7.41 | 25.23 ± 11.02 | 24.46 ± 8.16 | 23.11 ± 1.93 | 14.77 ± 0.94 |
| | cdk-substructure + NP (Yi et al., 2023) | 26.47 ± 2.42 | 48.73 ± 7.95 | 33.37 ± 2.97 | 29.35 ± 4.55 | 14.69 ± 2.13 | 10.04 ± 1.54 |
| | estate + NP (Yi et al., 2023) | 23.78 ± 5.8 | 35.14 ± 12.35 | 28.52 ± 6.31 | 26.91 ± 8.98 | 14.14 ± 2.87 | 9.3 ± 1.96 |
| | extended + NP (Yi et al., 2023) | 26.67 ± 3.97 | 10.87 ± 11.86 | 16.27 ± 5.74 | 18.75 ± 7.0 | 23.12 ± 1.55 | 14.85 ± 1.17 |
| | fp2 + NP (Yi et al., 2023) | 24.4 ± 4.26 | 12.13 ± 9.88 | 13.82 ± 9.48 | 16.71 ± 6.24 | 22.99 ± 2.13 | 15.1 ± 0.58 |
| | fp4 + NP (Yi et al., 2023) | 25.63 ± 2.17 | 21.69 ± 6.67 | 22.79 ± 8.38 | 25.66 ± 2.75 | 16.99 ± 1.61 | 11.75 ± 1.35 |
| | graph + NP (Yi et al., 2023) | 21.94 ± 2.42 | 14.86 ± 9.93 | 26.03 ± 7.99 | 24.05 ± 2.72 | 13.6 ± 3.55 | 10.12 ± 1.35 |
| | hybridization + NP (Yi et al., 2023) | 21.21 ± 5.06 | 16.29 ± 11.04 | 17.9 ± 8.75 | 20.38 ± 9.38 | 14.0 ± 2.26 | 11.91 ± 1.01 |
| | maccs + NP (Yi et al., 2023) | 28.64 ± 3.88 | 25.7 ± 13.06 | 21.94 ± 9.82 | 25.26 ± 2.78 | 22.3 ± 2.73 | 16.08 ± 1.23 |
| | pubchem + NP (Yi et al., 2023) | 25.96 ± 3.25 | 8.08 ± 7.85 | 10.58 ± 4.14 | 26.48 ± 5.92 | 19.58 ± 2.49 | 13.06 ± 1.21 |
| | rdkit + NP (Yi et al., 2023) | 18.13 ± 3.29 | 12.2 ± 6.64 | 11.39 ± 5.4 | 13.85 ± 4.15 | 14.96 ± 1.97 | 10.94 ± 1.38 |
| | standard + NP (Yi et al., 2023) | 26.77 ± 2.95 | 10.84 ± 6.91 | 11.39 ± 5.97 | 19.98 ± 4.18 | 23.12 ± 1.88 | 15.34 ± 0.76 |
| | Subgraph + NP | 47.93 ± 5.23 | **75.69 ± 11.54** | **64.8 ± 11.11** | 54.6 ± 7.0 | 23.12 ± 2.6 | 15.32 ± 1.13 |
| | SubDyve | **47.98 ± 3.24** | 58.41 ± 17.58 | 58.92 ± 10.07 | **56.02 ± 4.99** | **26.15 ± 4.02** | **16.51 ± 1.7** |
| 10 | avalon + NP (Yi et al., 2023) | 32.24 ± 4.29 | 40.55 ± 14.12 | 30.92 ± 8.36 | 24.86 ± 5.37 | 24.47 ± 1.55 | **17.38 ± 0.88** |
| | cdk-substructure + NP (Yi et al., 2023) | 30.0 ± 9.6 | 47.35 ± 26.79 | 42.32 ± 16.4 | 32.61 ± 13.84 | 15.37 ± 4.07 | 11.18 ± 2.97 |
| | estate + NP (Yi et al., 2023) | 26.79 ± 5.72 | 43.27 ± 19.89 | 34.23 ± 7.6 | 28.95 ± 7.35 | 15.5 ± 4.93 | 10.04 ± 3.45 |
| | extended + NP (Yi et al., 2023) | 32.83 ± 3.97 | 47.55 ± 13.6 | 34.15 ± 11.64 | 28.53 ± 6.44 | 24.48 ± 0.96 | 17.3 ± 1.17 |
| | fp2 + NP (Yi et al., 2023) | 34.52 ± 3.56 | **63.37 ± 7.05** | 47.13 ± 6.08 | 30.16 ± 5.82 | 24.34 ± 1.39 | 17.06 ± 1.28 |
| | fp4 + NP (Yi et al., 2023) | 30.54 ± 7.55 | 51.5 ± 12.6 | 43.14 ± 9.51 | 33.0 ± 9.84 | 16.99 ± 4.69 | 12.08 ± 3.05 |
| | graph + NP (Yi et al., 2023) | 32.52 ± 7.27 | 44.57 ± 3.3 | 38.22 ± 6.05 | 34.64 ± 9.29 | 18.49 ± 4.68 | 11.91 ± 2.69 |
| | hybridization + NP (Yi et al., 2023) | 28.68 ± 4.07 | 54.27 ± 9.56 | 40.7 ± 8.98 | 24.45 ± 5.17 | 16.86 ± 3.14 | 13.55 ± 2.21 |
| | maccs + NP (Yi et al., 2023) | 36.37 ± 6.53 | 50.03 ± 10.02 | 41.44 ± 7.9 | 39.52 ± 9.08 | 20.39 ± 3.94 | 13.14 ± 1.9 |
| | pubchem + NP (Yi et al., 2023) | 30.84 ± 4.56 | 37.75 ± 10.07 | 39.07 ± 3.28 | 29.74 ± 6.4 | 20.94 ± 3.77 | 14.36 ± 1.35 |
| | rdkit + NP (Yi et al., 2023) | 31.63 ± 3.95 | 59.58 ± 9.9 | 45.56 ± 7.01 | 30.14 ± 4.73 | 18.22 ± 4.62 | 12.65 ± 2.69 |
| | standard + NP (Yi et al., 2023) | 32.1 ± 5.18 | 45.99 ± 14.46 | 34.94 ± 10.46 | 28.94 ± 7.23 | 23.11 ± 2.82 | 17.38 ± 0.8 |
| | Subgraph + NP | 43.67 ± 5.04 | 53.64 ± 17.19 | 50.74 ± 9.72 | 45.52 ± 8.51 | 25.2 ± 2.09 | 15.39 ± 1.23 |
| | SubDyve | **44.88 ± 8.43** | 50.76 ± 21.98 | **54.27 ± 12.31** | **46.87 ± 6.5** | **26.48 ± 5.59** | 15.95 ± 2.38 |
| 15 | avalon + NP (Yi et al., 2023) | 23.25 ± 3.56 | 23.02 ± 11.06 | 17.9 ± 9.81 | 17.93 ± 6.75 | 19.03 ± 1.43 | 14.12 ± 0.76 |
| | cdk-substructure + NP (Yi et al., 2023) | 23.0 ± 0.63 | 32.54 ± 11.67 | 26.89 ± 7.56 | 24.06 ± 2.72 | 13.33 ± 1.59 | 9.39 ± 0.93 |
| | estate + NP (Yi et al., 2023) | 19.38 ± 4.78 | 10.81 ± 6.88 | 21.18 ± 3.98 | 24.04 ± 6.74 | 10.47 ± 2.81 | 6.45 ± 1.88 |
| | extended + NP (Yi et al., 2023) | 22.33 ± 3.83 | 12.23 ± 11.7 | 9.75 ± 7.07 | 17.93 ± 7.78 | 19.44 ± 1.8 | 13.46 ± 0.73 |
| | fp2 + NP (Yi et al., 2023) | 20.87 ± 3.92 | 12.16 ± 10.83 | 13.82 ± 9.48 | 13.86 ± 6.75 | 18.36 ± 2.82 | 13.87 ± 1.03 |
| | fp4 + NP (Yi et al., 2023) | 21.78 ± 1.8 | 25.73 ± 11.62 | 17.9 ± 8.38 | 21.59 ± 2.77 | 14.14 ± 1.74 | 12.57 ± 0.47 |
| | graph + NP (Yi et al., 2023) | 24.06 ± 2.18 | 5.4 ± 5.05 | 6.51 ± 4.15 | 30.15 ± 5.67 | 16.32 ± 1.55 | 10.86 ± 1.08 |
| | hybridization + NP (Yi et al., 2023) | 22.39 ± 4.7 | 5.43 ± 5.08 | 13.01 ± 8.66 | 22.82 ± 6.74 | 16.31 ± 3.52 | 14.12 ± 1.2 |
| | maccs + NP (Yi et al., 2023) | 23.25 ± 2.23 | 9.5 ± 8.15 | 6.5 ± 4.15 | 20.36 ± 4.63 | 18.63 ± 2.74 | 12.97 ± 1.25 |
| | pubchem + NP (Yi et al., 2023) | 19.15 ± 2.54 | 4.06 ± 3.31 | 4.88 ± 4.74 | 15.89 ± 3.51 | 17.0 ± 2.65 | 13.06 ± 1.57 |
| | rdkit + NP (Yi et al., 2023) | 20.56 ± 3.31 | 18.95 ± 13.11 | 15.46 ± 5.4 | 15.07 ± 3.54 | 18.22 ± 2.41 | 11.75 ± 1.14 |
| | standard + NP (Yi et al., 2023) | 21.02 ± 2.7 | 6.76 ± 6.04 | 6.5 ± 4.14 | 15.91 ± 6.1 | 19.85 ± 2.49 | 13.46 ± 1.24 |
| | Subgraph + NP | 43.09 ± 5.5 | 48.21 ± 7.75 | 50.86 ± 6.53 | 47.36 ± 7.5 | 22.9 ± 3.74 | 15.11 ± 1.41 |
| | SubDyve | **44.24 ± 7.4** | **55.92 ± 31.8** | **57.39 ± 18.94** | **48.3 ± 4.97** | **23.72 ± 3.25** | **15.4 ± 1.31** |

Table 22: Sensitivity analysis of SubDyve under varying LFDR thresholds ($\tau \in 0.05, 0.10,$ $0.30, 0.50$) and iteration counts (3–10). Each entry reports the relative performance difference (%) from the baseline configuration ($\tau = 0.10$, iterations = 6) across three metrics: BEDROC, $EF_{1\%}$, and $EF_{5\%}$. The **best** and second-best values are highlighted.

| Iterations | $\tau = 0.05$ | | | $\tau = 0.10$ | | | $\tau = 0.30$ | | | $\tau = 0.50$ | | |
|---|---|---|---|---|---|---|---|---|---|---|---|---|
| | BEDROC | $EF_{1\%}$ | $EF_{5\%}$ | BEDROC | $EF_{1\%}$ | $EF_{5\%}$ | BEDROC | $EF_{1\%}$ | $EF_{5\%}$ | BEDROC | $EF_{1\%}$ | $EF_{5\%}$ |
| 3 | +0.15 | +1.67 | -0.26 | -0.00 | -0.30 | -0.41 | -0.73 | +0.67 | -0.61 | -0.53 | +0.41 | -0.45 |
| 5 | +0.21 | +0.99 | -0.12 | +0.02 | -0.59 | -0.17 | -0.28 | -0.44 | -0.13 | +0.04 | +1.20 | -0.50 |
| 6 | +0.04 | +0.40 | -0.48 | +0.00 | +0.00 | +0.00 | -0.07 | -0.33 | -0.04 | **+0.44** | +0.27 | +0.05 |
| 7 | -0.14 | -0.08 | -0.29 | -0.39 | +0.55 | -0.41 | -0.48 | +0.46 | -0.25 | +0.35 | +0.62 | -0.03 |
| 10 | -0.36 | +0.82 | -0.45 | +0.20 | **+3.51** | -0.16 | -0.59 | -0.49 | -0.45 | +0.18 | +0.91 | **+0.12** |
| Max $|\Delta|$ (%) | BEDROC: 0.73%, $EF_{1\%}$: 3.51%, $EF_{5\%}$: 0.61% | | | | | | | | | | | |

Table 23: Overall statistics summarizing the stability of SubDyve across 20 hyperparameter configurations (four LFDR thresholds × five iteration counts). The table reports the mean, standard deviation, coefficient of variation (CV), and performance range for each retrieval metric.

| Metric | Mean | Std Dev | CV (%) | Range (%) |
|---|---|---|---|---|
| BEDROC | 86.2095 | 0.2709 | **0.31** | 0.81 |
| $EF_{1\%}$ | 56.69 | 0.50 | **0.88** | 2.27 |
| $EF_{5\%}$ | 17.04 | 0.04 | **0.23** | 0.60 |
| $EF_{10\%}$ | 8.74 | 0.03 | **0.29** | 0.76 |

## F.5 Ablation Study: Systematic Analysis of LFDR Refinement Stability and Computational Overhead

To examine the behavior of LFDR-guided refinement in a systematic manner, we first evaluated its sensitivity to calibration parameters. We conducted a two-dimensional sweep over both the LFDR threshold $\tau$ and the number of refinement iterations, considering $\tau \in \{0.05, 0.10, 0.30, 0.50\}$ and iteration counts in $\{1, 3, 5, 7\}$. These settings span regimes that prioritize conservative updates, aggressive seed expansion, and intermediate-confidence regions in which many compounds lie between clear positive and negative structural evidence. Appendix Table 22, across all tested configurations, SubDyve's early-recognition metrics (BEDROC, $EF_{1\%}$, $EF_{5\%}$) varied only slightly relative to the default configuration ($\tau = 0.10$, 6 iterations), typically within a 1% deviation band. Appendix Table 23 shows consistently low CV values ($<1\%$) and narrow score ranges, indicating that SubDyve maintains stable behavior under variations in both the LFDR threshold and the number of refinement iterations. This demonstrates the robustness of the LFDR procedure to these hyperparameter choices.

We next examined the computational overhead of LFDR refinement. Although the procedure involves multiple propagation–update cycles, the practical cost remains modest because the subgraph-mining stage extracts highly discriminative structural patterns that govern connectivity within the constructed network. Compounds that do not share meaningful substructures with the seed molecules naturally receive near-zero similarity during network construction, limiting their participation in propagation. Consequently, LFDR primarily operates over a compact and behaviorally relevant neighborhood around the seeds rather than over the full candidate library. This selective propagation behavior aligns with the ablation trends in Table 3, where subgraph fingerprints and LFDR refinement reinforce one another—enhancing early-recognition performance while avoiding unnecessary computation.

## F.6 Ablation Study: False Positive Pressure via LFDR-guided Refinement

Figure 4 – 8 further analyze the effect of the seed-selection rule ($\tau_{\mathrm{FDR}}$) on calibration and retrieval performance, which is considered a controlled false positive (FP)-pressure experiment: A low $\tau_{\mathrm{FDR}}$ imposes stricter FDR control (low FP-pressure), whereas a high $\tau_{\mathrm{FDR}}$ relaxes control (high FP-pressure), thereby including more false positives in the refined seed set. In brief, we evaluate the impact of the LFDR threshold $\tau_{\mathrm{FDR}}$ on calibration and screening performance as shown in Figure 4. As a baseline, we compare against a

probability-based refinement strategy (denoted as PROB), which directly thresholds GNN logits without LFDR estimation.

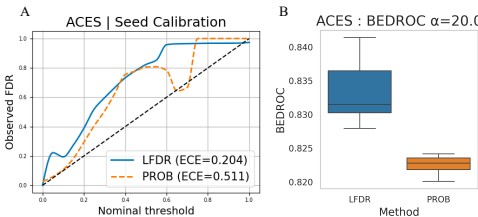

Figure 4: Effect of LFDR Threshold $\tau_{\mathrm{FDR}}$. (A) Seed-calibration curves for LFDR (blue) and PROB thresholding (orange). Calibration quality improves as the curve approaches the diagonal; ECE values are shown for both methods. (B) Boxplot of BEDROC ($\alpha = 20$) scores for LFDR and PROB across threshold values.

Figure 4A shows the seed-calibration curves across thresholds for each method. LFDR achieves substantially better calibration, with an expected calibration error (ECE) of 0.204 compared to 0.511 for PROB. This indicates that LFDR maintains robust performance even under increasing FP-pressure, validating its practical FDR control capability. Figure 4B shows BEDROC scores across five LFDR thresholds ($\tau_{\mathrm{FDR}}$) and five probability thresholds ($\tau_{\mathrm{PROB}}$). LFDR consistently yields higher performance across the threshold range, suggesting robustness against increasing FP-noise. Similar trends for $EF_{1\%}$ and AUPRC are shown in Figure 7 and Figure 8.

- **Figure 5**: Seed-calibration curves comparing LFDR-based and probability-based (PROB) refinement strategies. LFDR yields lower expected calibration error (ECE) across most targets, demonstrating better control of false discovery rates.

- **Figure 6**: BEDROC ($\alpha = 20$) scores evaluated across thresholds. LFDR generally shows higher BEDROC values with reduced variance, reflecting improved early enrichment.

- **Figure 7**: $EF_{1\%}$ plotted as a function of threshold. LFDR consistently outperforms PROB across thresholds for most targets, confirming its robustness under different calibration conditions.

- **Figure 8**: Precision–recall (PR) curves at the best-performing threshold per method. LFDR achieves higher PR-AUC for the majority of targets, especially those with imbalanced label distributions.

Together, these results highlight that the LFDR-guided refinement process controls the false discovery rate and mitigates false positive noise under varying FP-pressures. SubDyve demonstrates strong and stable performance across a variety of target proteins, offering a reliable solution for virtual screening under sparse supervision.

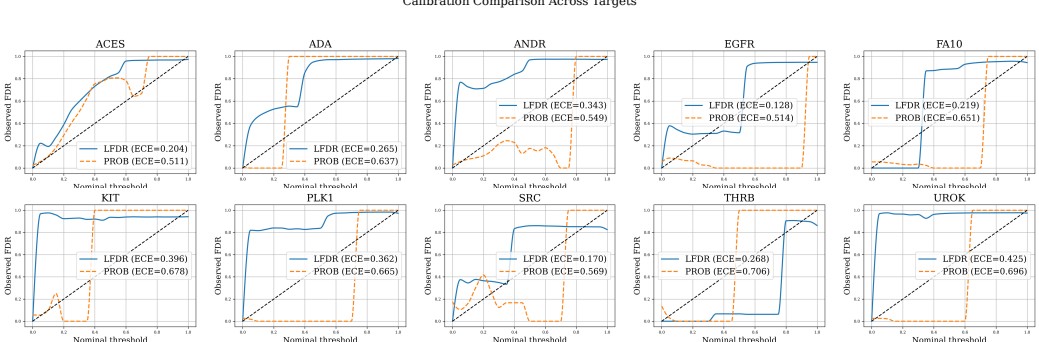

Figure 5: Effect of seed-selection rule on FDR control and early recognition for 10 DUD-E targets. Seed-calibration curves for LFDR (blue) and probability thresholding (orange). The closer the curve lies to the diagonal, the better the calibration. Expected calibration error (ECE) is annotated for both methods.

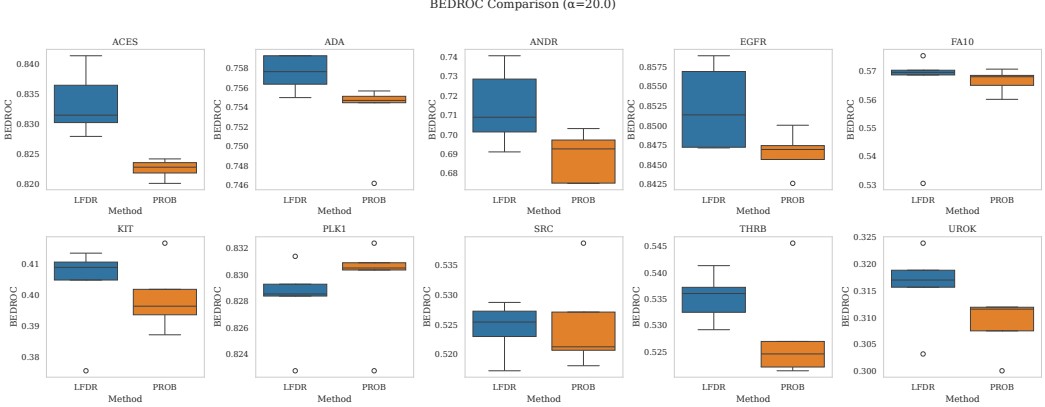

Figure 6: Effect of seed-selection rule on FDR control and early recognition for 10 DUD-E targets. BEDROC ($\alpha = 20$) scores evaluated across the threshold grid; boxes show the interquartile range, whiskers the 5–95 percentiles. Box plot for LFDR (blue) and probability (orange).

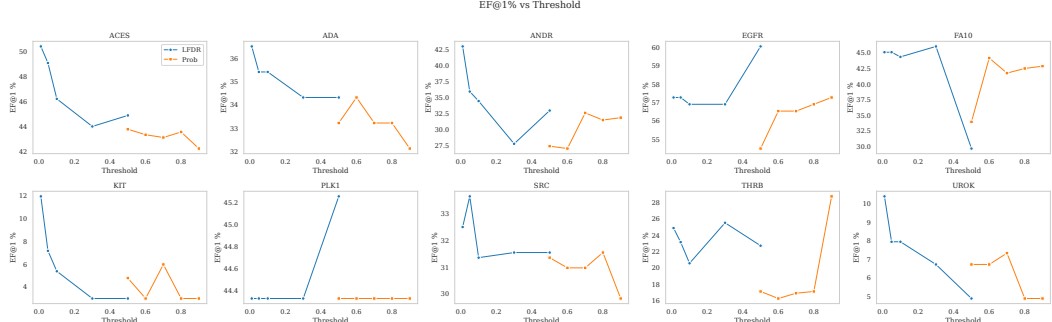

Figure 7: Effect of seed-selection rule on FDR control and early recognition for 10 DUD-E targets. Enrichment factor at the top 1% of the ranking ($EF_{1\%}$) as a function of the threshold. Line plot for LFDR (blue) and probability (orange).

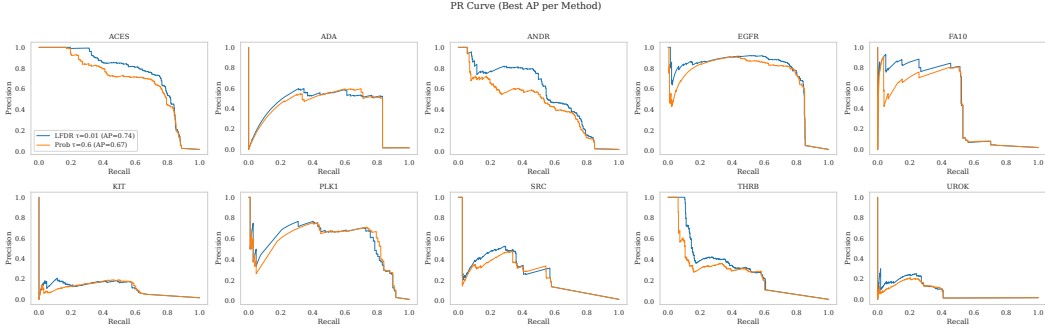

Figure 8: Effect of seed-selection rule on FDR control and early recognition for 10 DUD-E targets. Precision–recall curves at the best threshold for each rule. Legends indicate the chosen $\tau$ and the corresponding PR-AUC.

### F.7 Ablation Study: Varying $d$-dimensional subgraph pattern fingerprint

To evaluate the impact of subgraph pattern size on virtual screening performance, we present the results of SubDyve under varying fingerprint sizes in Figures 9. The number of subgraph patterns mined using the SSM algorithm and selected according to their entropy importance was varied as $d \in \{100, 200, 300, 500, 1000, 2000\}$. The figure shows the average performance for 10 DUD-E targets evaluated using the AUROC, BEDROC, $EF_{1\%}$, $EF_{5\%}$, and $EF_{10\%}$ metrics. As the number of patterns increases, SubDyve shows consistently improved performance across all metrics, indicating that incorporating a broader range of chemically informative substructures improves model representation. For the finalized setting of $d = 2000$, we additionally report target-specific AUROC scores to highlight the consistency of performance across targets.

All results are calculated with 100 bootstrap resamples to obtain confidence intervals and report both the mean and standard deviation. These results highlight the benefits of capturing different subgraph-level features, which contribute substantially to improving screening accuracy in low-label environments.

### F.8 Ablation Study: GNN Model Feature Components

To more thoroughly assess the contribution of each component in GNN model to SubDyve's ranking performance, we performed an extended set of ablation experiments. These experiments isolate the effect of pretrained ChemBERTa embeddings, the hybrid PCA–NP ranking module, and individual scoring elements such as the seed-weight term, the network-

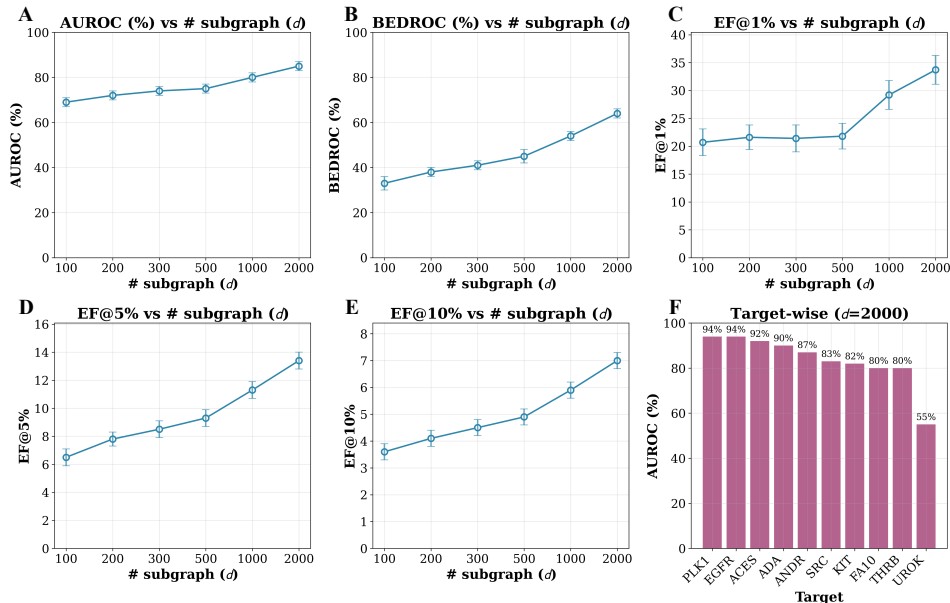

Figure 9: Performance comparison for varying numbers of $d$ in the subgraph pattern removal study. (A-E) Average performance metrics (AUROC, BEDROC, $EF_{1\%}$, $EF_{5\%}$, $EF_{10\%}$) for 10 DUD-E targets as a function of $d$. Error bars represent the standard deviation of 100 bootstrap resamples. (F) AUROC performance per target at $d = 2000$, ranked by performance per target.

Table 24: Ablation study of GNN features on the PU dataset. The full model (with both ChemBERTa features and hybrid ranking) achieves the best performance. Top and second-best results are in **bold** and underline, respectively.

| Setting | BEDROC | $\mathbf{EF}_{1\%}$ |
|---|---|---|
| Full | **83.44 $\pm$ 1.44** | **97.59 $\pm$ 1.44** |
| -Hybrid | 81.73 $\pm$ 3.10 | 91.64 $\pm$ 3.65 |
| -Hybrid (PCA only) | 82.25 $\pm$ 2.92 | 96.08 $\pm$ 2.61 |
| -Hybrid (NP only) | 80.16 $\pm$ 3.88 | 91.14 $\pm$ 4.06 |
| -FP | 81.87 $\pm$ 2.94 | 94.35 $\pm$ 1.93 |
| -PCA | 80.28 $\pm$ 2.84 | 93.93 $\pm$ 2.13 |
| -Seed Weight | 79.40 $\pm$ 3.64 | 89.90 $\pm$ 3.48 |
| -NP Score | 79.84 $\pm$ 3.51 | 90.63 $\pm$ 4.04 |
| -ChemBERTa | 79.82 $\pm$ 4.40 | 88.92 $\pm$ 5.02 |

propagation (NP) score, and the PCA-derived feature projection. Removing each component allows us to quantify its marginal influence on early-recognition metrics.

Across all settings, we observed consistent decreases in BEDROC and $EF_{1\%}$ relative to the full model (Appendix Table 24). Disabling the hybrid ranking module or replacing it with a single scoring scheme led to performance degradation, highlighting the benefit of combining complementary signals. Similarly, eliminating ChemBERTa features resulted in one of the largest drops, indicating the role of pretrained molecular representations in enhancing discriminative power.

Taken together, these results demonstrate that SubDyve's performance gains arise from the coordinated effect of multiple interacting modules. The hybrid ranking mechanism, in particular, provides a synergistic integration of PCA-derived structure features, propagation-based signals, and pretrained embeddings, each of which contributes meaningfully to the overall retrieval quality.

Table 25: Comparison of BEDROC, running time (preprocessing, inference), and memory usage across different models on the PU dataset. Time and memory usage reports average values.

| Model | BEDROC | Preprocessing (sec) | Inference (sec) | Memory Used (MiB) |
|---|---|---|---|---|
| AutoDock Vina | $1.0 \pm 1.3$ | - | 13343.37 | 215.7 |
| DrugCLIP | $2.7 \pm 1.26$ | 2955.93 | 29.53 | 9605.1 |
| GRAB | $40.68 \pm 10.60$ | 1038.97 | 0.03 | 661.1 |
| rdkit + NP | $79.04 \pm 1.96$ | 956.70 | - | 771.7 |
| SubDyve | $83.44 \pm 1.44^{**}$ | 1088.0 | - | 637.9 |

## F.9 TIME AND MEMORY COST

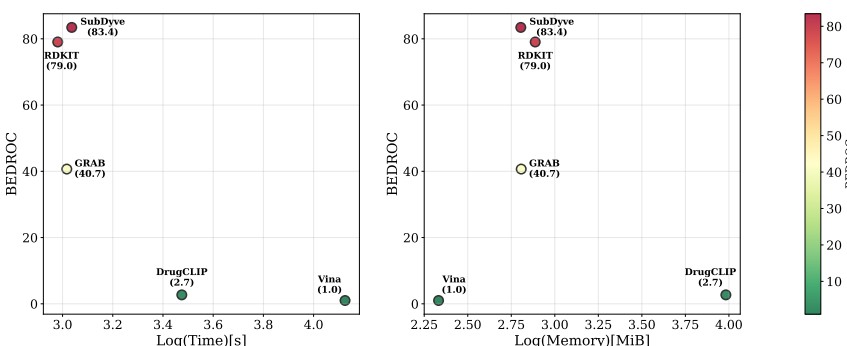

Figure 10: Comparison of BEDROC, computational runtime, and memory consumption across models.

We conducted a comprehensive comparison of the running time and memory consumption across different baseline models on the PU dataset to ensure a fair evaluation. As shown in Table 25, AutoDock Vina required the lowest memory resources but incurred the longest runtime, with the runtime increasing further when advanced functionalities such as multi-ligand docking or force field expansions were used. Here, preprocessing refers to the time required for data preparation and model training, whereas inference denotes the time taken for model prediction. The memory requirement was computed from the features each model uses and the cost of loading the data. In Figure 10, RDKIT refers to the model rdkit+NP, and Vina refers to Autodock Vina.

When the total time, defined as the sum of preprocessing and inference times, is considered, DrugCLIP, in contrast, suffered from substantial preprocessing overhead, requiring $2.74\times$ longer runtime and $15.06\times$ more memory than SubDyve.

SubDyve achieved these computational savings while maintaining moderate resource requirements compared to rdkit+NP and GRAB, which consumed similar memory but benefited from slightly lower runtimes as shown in Figure 10. However, this efficiency in RDKit+NP and GRAB came at the expense of predictive accuracy, as SubDyve delivered significantly higher BEDROC scores ($+4.4\%$ vs. RDKit+NP, $+42.76\%$ vs. GRAB). These findings highlight a clear trade-off between computational efficiency and predictive performance, underscoring that while SubDyve does not minimize runtime or memory usage, it provides state-of-the-art accuracy at an acceptable computational cost.

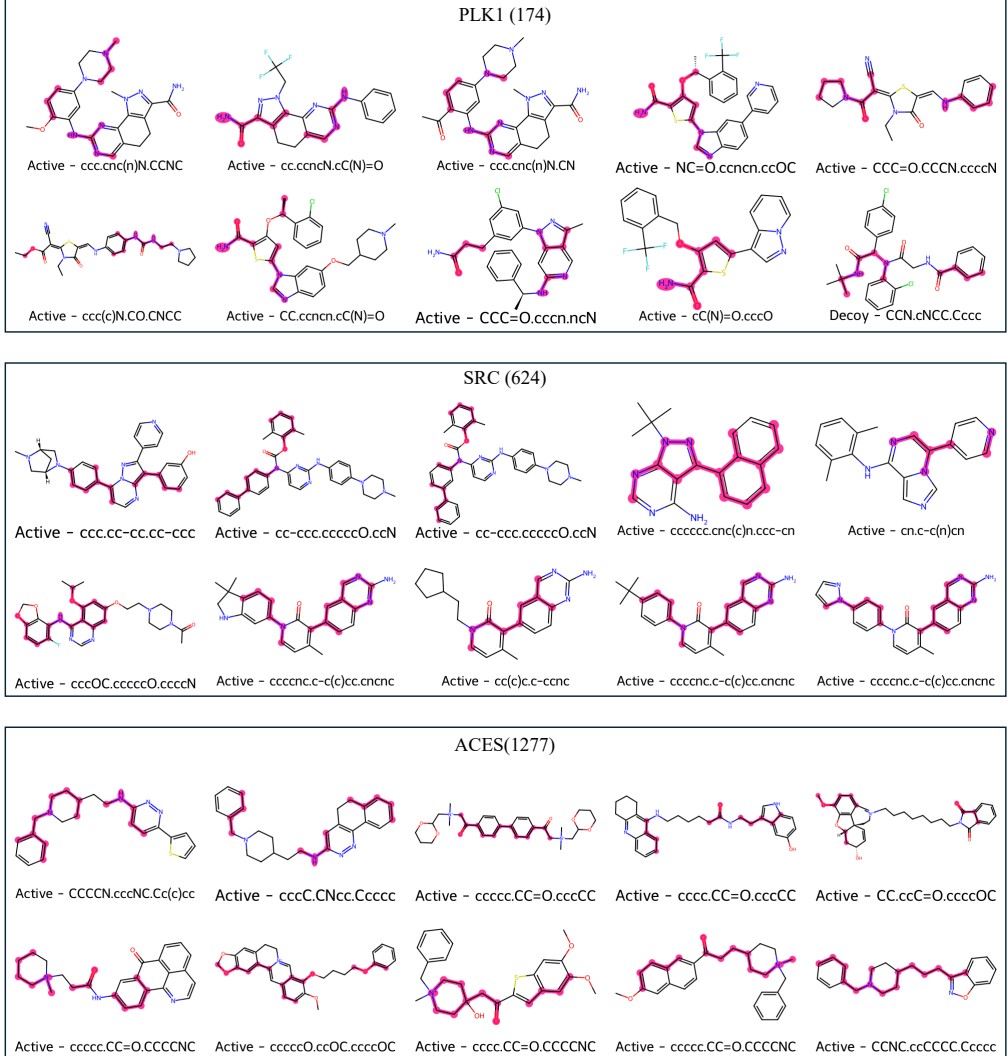

Figure 11: Top-10 matched Active/Decoy on the DUD-E Dataset over the number of seeds utilized.

## G  ADDITIONAL CASE STUDY RESULTS

### G.1  TOP-10 MATCHED ACTIVE/DECOY ON THE DUD-E DATASET OVER THE NUMBER OF SEEDS UTILIZED

To check the distribution of active and decoy sets with subgraphs according to the number of seeds used, we investigated the targets with the lowest, average, and highest number of seeds for the DUD-E dataset in Figure 11. PLK1 (174 seeds utilized), the target with the lowest number of seeds, captured one decoy molecule with a subgraph, while the rest remained active. Even though decoy ranked in the top-10, the distribution of utilized subgraphs varied, with 10 different subgraphs captured. For the average number of SRC (624) and ACES (1277) targets, all molecules were active, and the distribution of subgraphs captured varied. For SRC, 8 subgraph patterns were captured, and for ACES, 9 were captured.

Therefore, this result suggests that a higher number of seeds provides more structural diversity to reflect the subgraphs of active and allows for more reliable structure-based analyses. Nevertheless, the low number of seeds also reveals as much structural information

about the molecule as the number of subgraph patterns, suggesting that the results are comparable to those with a higher number of seeds.

## G.2 Active-Decoy Ranking Gap for DUD-E datasets

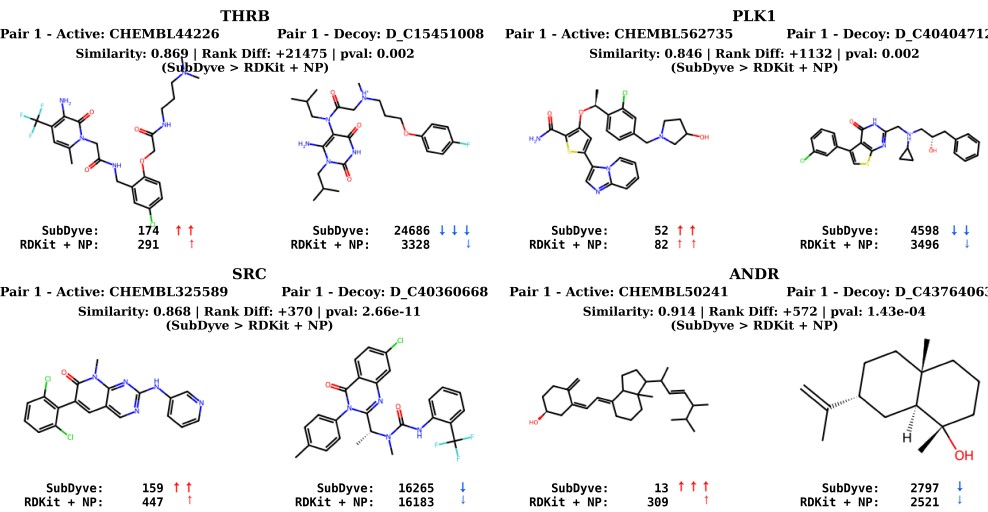

Figure 12: Ranking gap for Active/Decoy on the DUD-E Dataset of 4 targets

To demonstrate the effectiveness of SubDyve's ranking capability, we compare its performance to the best-performing RDKit+NP baseline, which is based on general-purpose molecular fingerprints. We focus on structurally similar active-decoy pairs and calculate the ranking gap between them. As shown in Figure 12, SubDyve consistently ranks the active compounds higher and the decoy compounds lower than the baseline, resulting in a much larger ranking gap. This indicates that even under conditions of high structural similarity, SubDyve is able to distinguish the true active compounds.

To facilitate visual interpretation, we annotated each active compound with an up arrow (one, two, or three arrows for the top 50, top 250, and top 500, respectively) based on its rank in the output of each model. For decoys, we annotated with one, two, or three down arrows when SubDyve rank improved by $< 10\%$, $< 50\%$, or $\geq 50\%$, respectively, compared to the rank difference between SubDyve and RDKit+NP on a percentile basis. The rank difference is calculated as the gap between the active and decoy rankings, and we report the difference in this gap by model. Higher values indicate that SubDyve separates the active from the decoys more than the baseline. Statistical significance is evaluated using the Wilcoxon signed rank test for all matched pairs.

## G.3 Subgraph-Level Characterization of Augmented Seeds on the PU dataset

To reveal the structural properties of compounds added during SubDyve's refinement, we analyze the differences between the initial and augmented seed sets across three perspectives: subgraph motif enrichment, compound-level pattern visualization, and graph-level connectivity.

First, we identify subgraph motifs that are significantly enriched in the augmented seed set. As shown in Figure 13A, multiple motifs exhibit increased presence after refinement, with Fisher's exact test ranking the top 40 patterns by significance. Figure 13B further quantifies these enrichments by measuring the difference in average motif counts, with statistical significance determined using unpaired t-tests and Benjamini–Hochberg correction. These results indicate that SubDyve preferentially amplifies discriminative patterns rather than merely expanding chemical diversity.

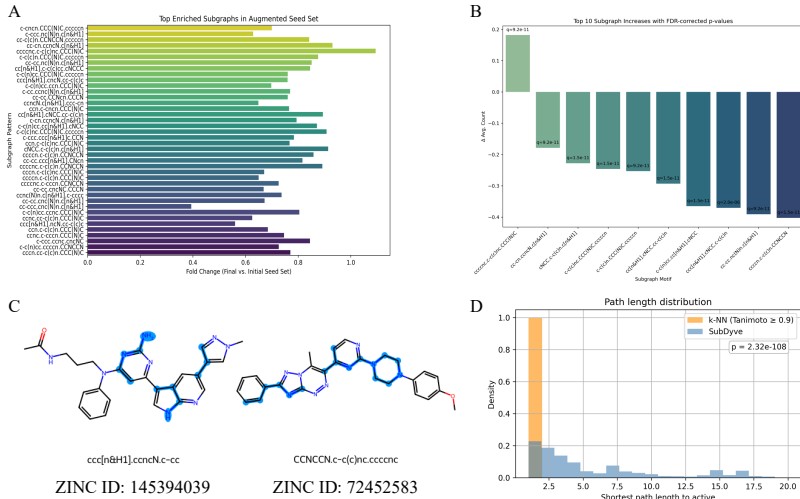

Figure 13: Subgraph-level analysis of augmented seeds on PU dataset. (A) Top 40 subgraph motifs enriched in the augmented seed set, ranked by $p$-values from Fisher's exact test. (B) Subgraphs with the largest mean frequency increase relative to the initial seeds, with $q$-values from unpaired $t$-tests (Benjamini–Hochberg correction). (C) Examples of ZINC compounds containing enriched subgraphs that pass Lipinski's rule of five; matched motifs are highlighted. (D) Distribution of shortest path lengths from initial seeds to retrieved actives in the SubDyve graph versus a Tanimoto $k$-NN baseline.

Second, we examine how enriched substructures manifest in real molecules. Figure 13C presents ZINC compounds from the augmented set that satisfy Lipinski's rule of five and contain representative enriched subgraphs. Highlighted regions confirm that these patterns correspond to chemically meaningful and interpretable substructures rather than artifacts of global structural similarity.

Lastly, we assess whether SubDyve can prioritize structurally distant yet bioactive compounds. We compute the shortest path distances from each retrieved compound to the closest known active in the subgraph-similarity network and compare this distribution to a Tanimoto $k$-NN baseline (similarity $\geq 0.9$). As shown in Figure 13D, SubDyve retrieves candidates with significantly longer path distances ($p = 2.32 \times 10^{-108}$, Mann–Whitney U-test), supporting its ability to generalize beyond immediate structural neighbors while maintaining high enrichment performance.

These results suggest that SubDyve refines the seed set in a substructure-aware and chemically meaningful manner, enabling robust prioritization of active compounds even when they lie outside the reach of traditional fingerprint-based similarity metrics.

### G.4 Chemical and Pharmacophoric Relevance of Mined Subgraph Patterns

To investigate the chemical interpretability and pharmacophoric relevance of the subgraph patterns mined by SubDyve, we conducted an analysis focused on the well-characterized binding-site features of CDK7. Structural and biochemical studies have shown that the ATP-binding pocket of CDK7 contains a predominantly hydrophobic cavity formed by residues such as Phe91 and Leu144, which support $\pi$–$\pi$ stacking, hydrophobic packing, and van der Waals contacts in addition to canonical hinge-binding hydrogen-bond interactions (Düster et al., 2024; Kumar et al., 2021; Peissert et al., 2020). Potent CDK7 inhibitors frequently exploit these interaction motifs, combining aromatic scaffolds with heteroaromatic hinge-binding fragments and aliphatic substituents that occupy adjacent shallow pockets.

The subgraphs extracted by SubDyve show strong alignment with these experimentally established binding requirements. Among the highest-ranked subgraph patterns, we observe frequent occurrences of aromatic ring systems (e.g., ccc, cc-ccc, cc-c(c)c) and nitrogen-containing heterocycles (e.g., ncccn, ccncNC, [n&H1]), which correspond to hinge-binding fragments commonly found across CDK family inhibitors. Additionally, the model identifies aliphatic amine–linked chains (e.g., CCCNC, CNCCC, CCC(C)N) that resemble substituents known to occupy hydrophobic subpockets adjacent to the hinge region in CDK7. These patterns collectively recapitulate the pharmacophoric elements—aromaticity, hydrogen-bonding capability, and hydrophobic chain extension—known to drive CDK7 ligand recognition.

Figure 14 shows (i) a docking simulation of a known active compound from PubChem against the CDK7 target and (ii) the docking pose of a high-ranked compound retrieved from the ZINC library. The green molecule represents the high-ranked candidate identified by SubDyve, while the blue molecule corresponds to ATP bound in the CDK7 pocket. The highlighted subgraph contains two hydrophobic pharmacophore features along with an additional non-pharmacophoric fragment. As illustrated in the figure, the hydrophobic regions of the retrieved compound orient toward the ATP-binding pocket in a manner consistent with known binding preferences. These observations support that SubDyve identifies pharmacophorically meaningful subgraphs and that these patterns contribute to its strong early-recognition performance.

Furthermore, we evaluated the mined subgraphs using RDKit's pharmacophore annotations (donor, acceptor, aromatic, hydrophobe). Across the top ten subgraphs ranked by discriminative score, 60% contained at least one pharmacophore feature, indicating that the extracted structural motifs correspond to chemically meaningful functional groups rather than arbitrary graph fragments.

Taken together, these observations demonstrate that SubDyve retrieves subgraphs that are not only class-discriminative but also chemically interpretable and pharmacophorically coherent, consistently reflecting established principles of CDK7 binding-site engagement as well as broader functional group patterns relevant to small-molecule–protein interactions.

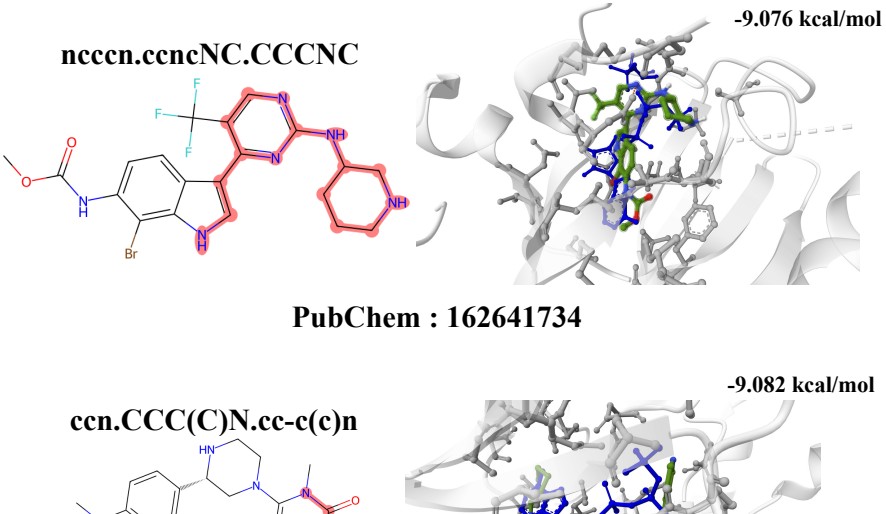

Figure 14: Docking poses of a known CDK7 active compound from PubChem and a high-ranked candidate retrieved from the ZINC library (green), shown in comparison to the bound ATP molecule (blue). The highlighted subgraph motifs exhibit hydrophobic and heteroaromatic features that orient toward the ATP-binding pocket, reflecting known CDK7 hinge-binding and hydrophobic interaction patterns.

# H    LIMITATIONS

SubDyve requires constructing a target-specific chemical similarity network for each protein target, which introduces preprocessing overhead due to repeated subgraph mining and graph construction. While this design enables tailored modeling of bioactivity-relevant structures, it may limit scalability when screening across a large number of targets. Additionally, although LFDR-based seed calibration consistently outperforms probability-based heuristics in terms of expected calibration error (ECE), performance in the mid-range threshold region remains suboptimal.

Despite these limitations, SubDyve offers a promising foundation for scalable virtual screening. Its modular architecture and uncertainty-aware design make it well suited for future extensions to multi-target or multi-omics settings, where integration with transcriptomic profiles or cell line information could further improve prioritization in complex biological contexts.

