# OpenReview forum: "SubDyve: Subgraph-Driven Dynamic Propagation for Virtual Screening Enhancement Controlling False Positive"
_ICLR.cc/2026/Conference — ICLR 2026 Poster_

### Official Review · Reviewer_yH49 · 2025-10-26

**Soundness:** 3
**Presentation:** 3
**Contribution:** 3
**Rating:** 8
**Confidence:** 2

**Summary:**

The paper introduces SubDyve, a subgraph-aware network propagation framework for label-efficient virtual screening in drug discovery. It combines subgraph fingerprint network construction, dynamic seed refinement using local false discovery rate (LFDR) estimation and a GNN-based model for stratified refinement. They evaluate on on DUD-E (10 targets) and a CDK7 target with a 10M-compound ZINC dataset, showing large performance gains over fingerprint-based and deep learning baselines.

**Strengths:**

1. Paper is well-written and structured. Also, URL containing code and experiments is provided for reproducibility.

2. The presented framework seems novel and efficient. For instance, the introduction of class-discriminative subgraph fingerprints captures fine-grained structural patterns that general-purpose fingerprints miss.

3. Comprehensive benchmarks demonstrate significant improvements across several evaluation metrics, such as outperforming large foundation models while using less data and compute.

4. They also provide beyond-performance analysis through case studies (Figure 2 and 3), where they show that subgraph-level information correlates with biologically meaningful substructures.

**Weaknesses:**

1. While substructure interpretability is discussed, there is little experimental validation linking retrieved subgraphs to real binding mechanisms.

2. The complexity of the presented pipeline (subgraph mining, LFDR estimation and GNN optimization) could make results to be very sensitive to changes in seeds, thresholds and hyperparameters. For instance, the effect of LFDR thresholds could be explored more systematically to assess stability and sensitivity. This can be specifically important for interpretability.

3. Results are primarily shown on small-molecule benchmarks; applicability to structure-based or protein–ligand joint settings is unclear.

**Questions:**

1. How sensitive is SubDyve to the LFDR threshold and the number of iterations in seed refinement?

2. How are the mined subgraph patterns evaluated or visualized in terms of chemical or pharmacophoric relevance?

3. Could you check if the provided URL for the code is working? I can't access it.

---

> ### Author Response · Authors · 2025-11-27
> **Response to Reviewer yH49 (Part 1/2)**
>
> We thank the reviewer for the examination of our work and the thoughtful comments provided. Kindly find our responses to the raised comments and questions below.
>
> > **Q1**: While substructure interpretability is discussed, there is little experimental validation linking retrieved subgraphs to real binding mechanisms.
> **Q5**: How are the mined subgraph patterns evaluated or visualized in terms of chemical or pharmacophoric relevance?
>
> **A1+A5**:  We thank the reviewer for this insightful question. To investigate chemical or pharmacophoric relevance of the mined subgraph patterns, we focused on the the context of CDK7’s known binding-site characteristics. Previous studies [1-3] show that the ATP-binding pocket of CDK7 is largely hydrophobic, formed by residues such as Phe91 and Leu144, and favors aromatic scaffolds, π–π stacking, and van der Waals–driven contacts, alongside canonical hinge-binding hydrogen-bond interactions.
>
> Consistent with these documented interaction requirements, the subgraph patterns extracted by SubDyve frequently include aromatic ring systems (e.g., ccc, cc-ccc, c-ccc) and N-heterocycles (e.g., cc[n&H1], ncccn, ccccn), which correspond to hinge-binding fragments known to mediate recognition across CDK family inhibitors. We also observe aliphatic amine–linked chains (e.g., CCC(C)N, CCCN, CCCNC) matching the hydrophobic substituents that occupy shallow pockets in CDK7.
>
> In addition to this target-specific analysis, we performed a general pharmacophore-feature check using RDKit. RDKit provides each pharmacophore feature (donor, acceptor, aromatic, etc.), and we examined whether the mined subgraphs map to these pharmacophore features. Among the top 10 subgraph patterns, 60% of patterns contained at least one pharmacophore features.
>
> These results collectively indicate that the mined patterns are chemically meaningful and align well with both CDK7’s binding-site characteristics and general pharmacophoric expectations.
>
> We have added a discussion of these findings in Appendix G.4
>
> ----
>
> >**Q2**: The complexity of the presented pipeline (subgraph mining, LFDR estimation and GNN optimization) could make results to be very sensitive to changes in seeds, thresholds and hyperparameters. For instance, the effect of LFDR thresholds could be explored more systematically to assess stability and sensitivity. This can be specifically important for interpretability.
> **Q4**: How sensitive is SubDyve to the LFDR threshold and the number of iterations in seed refinement?
>
> **A2+A4**: We sincerely thank the reviewer for raising this important question regarding the sensitivity of SubDyve to the LFDR threshold ($\tau$) and the number of refinement iterations. To evaluate robustness, we conducted a systematic sensitivity analysis on the ACES target from DUD-E. The base configuration used throughout the original submission uses $\tau$ = 0.10 and 6 refinement iterations, which were selected because they provided stable performance across multiple DUD-E targets, and this setting achieves BEDROC 86 ± 2, EF$_1$% 57.0 ± 2.4, and EF$_5$% 17.1 ± 0.4 for the ACES target.
>
> We explored $\tau$ values of 0.05, 0.10, 0.30, and 0.50, together with iteration counts of 3, 5, 6, 7, and 10. For clarity in the response, we summarize the marginal trends only: first, the effect of changing the number of iterations while $\tau$ is fixed, and second, the effect of changing $\tau$ while the iteration count is fixed. The complete sensitivity table covering all 20 configurations is provided in the revised manuscript.
>
> Across all settings, SubDyve showed strong stability. The maximum deviation from the baseline was 0.73% for BEDROC, 3.5% for EF$_1$%, and 0.61% for EF$_5$%. These results indicate that SubDyve does not rely on a narrow hyperparameter choice and remains robust over a broad range of LFDR thresholds and iteration counts.
>
> Values show relative (%) change from base configuration ($\tau$ = 0.10, iterations = 6).
>
> *Sensitivity to Iteration Count ($\tau$ = 0.10 fixed)*
> | Iterations |  BEDROC |  EF$_1$% |  EF$_5$% |
> |-----------:|---------:|--------:|--------:|
> | 3          | +0.00    | −0.30   | −0.41   |
> | 5          | +0.02    | −0.59   | −0.17   |
> | 6 (base)         | +0.00    | +0.00   | +0.00   |
> | 7          | −0.39   | +0.55   | −0.41    |
> | 10         | +0.20    | +3.51   | −0.16    |
>
> *Sensitivity to LFDR Threshold (iterations = 6 fixed)*
> | $\tau$    |  BEDROC |  EF$_1$% |  EF$_5$% |
> |-----:|----------:|--------:|--------:|
> | 0.05 | +0.04     | +0.40     | −0.48   |
> | 0.10 (base) | +0.00 | +0.00 | +0.00 |
> | 0.30 | −0.07     | −0.33   | −0.04   |
> | 0.50 | +0.44     | +0.27   | +0.05   |
>
> To provide additional clarity, we have incorporated a structured summary of this sensitivity analysis into the revised manuscript (Section 4.2.3 “Stability and Calibration Behavior of LFDR Refinement”), with the full set of results included in Appendix F.5

---

> > ### Author Response · Authors · 2025-11-27
> > **Response to Reviewer yH49 (Part 2/2)**
> >
> > >**Q3**: Results are primarily shown on small-molecule benchmarks; applicability to structure-based or protein–ligand joint settings is unclear.
> >
> > **A3**: We thank the reviewer for raising this important point. While our experiments focus on small-molecule benchmarks, the design of SubDyve is not limited to ligand-only settings. The core components of our framework: substructure-based similarity graphs and LFDR-guided seed refinement, operate independently of any specific molecular featurization pipeline and can be directly integrated with protein-informed representations.
> >
> > For example, the initial subgraph similarity graph can be augmented with structure-based descriptors or protein–ligand interaction fingerprints [4, 5], and docking scores or binding-affinity predictions can be used to prioritize more informative initial seeds, while the LFDR refinement stage remains unchanged. This flexibility allows SubDyve to integrate seamlessly with structure-based or joint protein–ligand pipelines.
> >
> > In fact, even in our ligand-only evaluation, SubDyve implicitly captures protein–ligand binding preferences. As shown in the CDK7 case study, the mined subgraph patterns align with known binding-site characteristics (e.g., aromatic/heteroaromatic hinge-binding motifs), suggesting that the method learns features relevant for target-specific interactions despite not using structural priors. This indicates that SubDyve’s mechanisms naturally generalize to joint protein–ligand environments.
> >
> > We agree that a full structure-based evaluation would further strengthen the scope of the work, and we consider this a valuable direction for future extension. We have added this discussion to the revised manuscript (Section 5. “Conclusion”).
> >
> > ------
> >
> > >**Q6**: Could you check if the provided URL for the code is working? I can't access it.
> >
> > **A6**: We thank the reviewer for pointing this out, and we apologize for the inconvenience. The anonymous GitHub repository has been updated, and we will ensure that the link remains accessible throughout the ICLR final decision period. Please let us know if any further access issues arise.
> >
> > -----
> >
> > [1] Düster, Robert, et al. "Structural basis of Cdk7 activation by dual T-loop phosphorylation." Nature Communications 15.1 (2024): 6597.
> >
> > [2]Kumar, Vikas, et al. "Identification of CDK7 inhibitors from natural sources using pharmacoinformatics and molecular dynamics simulations." Biomedicines 9.9 (2021): 1197.
> >
> > [3]Peissert, Stefan, et al. "Structural basis for CDK7 activation by MAT1 and Cyclin H." Proceedings of the National Academy of Sciences 117.43 (2020): 26739-26748
> >
> > [4] Desaphy, Jeremy, et al. "Encoding protein–ligand interaction patterns in fingerprints and graphs." Journal of chemical information and modeling 53.3 (2013): 623-637.
> >
> > [5] Koh, Huan Yee, et al. "Physicochemical graph neural network for learning protein–ligand interaction fingerprints from sequence data." Nature Machine Intelligence 6.6 (2024): 673-687.

---

### Official Review · Reviewer_iVxC · 2025-10-31

**Soundness:** 3
**Presentation:** 3
**Contribution:** 3
**Rating:** 6
**Confidence:** 4

**Summary:**

This paper addresses the problem of virtual screening (VS) in drug discovery under low-label regimes, where very few active ligands are known. The authors argue that existing network propagation (NP) methods suffer from reliance on generic molecular fingerprints and topological bias, leading to high false positive (FP) rates. The paper proposes SubDyve, a framework featuring two main components: 1) Constructing a task-specific similarity network by mining "class-discriminative subgraphs"; 2) Introducing a "dynamic seed refinement" mechanism that uses the local false discovery rate (LFDR) to iteratively add high-confidence (low FP probability) unlabeled molecules as new seeds, aiming to control FPs. Experiments on the DUD-E benchmark (zero-shot) and a large-scale ZINC dataset for CDK7 (PU learning) show that SubDyve outperforms baselines on early enrichment metrics.

**Strengths:**

1. The paper focuses on the low-label/few-shot VS problem, which is highly relevant and challenging in real-world drug discovery.
2. The primary highlight is the introduction of LFDR for a principled control of the dynamic seed set expansion to combat topological bias. This is more sophisticated than using fixed similarity or GNN score thresholds.
3. Evaluating the method on two challenging scenarios (zero-shot and large-scale PU) is a clear strength.
4. The results demonstrate a clear improvement in early enrichment metrics compared to the baseline methods.

**Weaknesses:**

1. The most significant concern is the method's scalability and preprocessing overhead. As acknowledged in the limitations (Section H), the method requires re-running the computationally intensive subgraph mining (SSM) and graph construction process for *every single* protein target. This could be a major practical barrier in a real-world drug discovery pipeline that needs to screen hundreds or thousands of targets.
2. The success of the entire pipeline (especially the SSM step) is highly dependent on the quality and quantity of the initial seed set $\mathcal{S}_{train}$. If the initial set is extremely small (e.g., < 10-20 molecules) or not representative, the subgraph mining step will likely fail or produce misleading patterns, causing the entire downstream pipeline (GNN training and LFDR refinement) to perform poorly. The 50 seeds used as the minimum in the experiment (Sec 4.2.2) is still a somewhat optimistic setting for a "low-label" scenario.

**Questions:**

1. Can the authors provide an end-to-end average time (e.g., on the DUD-E targets) required to build the subgraph network (including subgraph mining and similarity computation)? How does this compare to the preprocessing time of baselines (like RDKit+NP)?
2. How sensitive is the method to the size of  $S_{train} $? If $S_{train} $ contained only 10 or 5 molecules, would the SSM step still be effective?

---

> ### Author Response · Authors · 2025-11-27
> **Response to Reviewer iVxC**
>
> We thank the reviewer very much for the careful reading and comments regarding our work. Please, see below our answer to the raised comments/questions.
>
> > **Q1**: The most significant concern is the method's scalability and preprocessing overhead. As acknowledged in the limitations (Section H), the method requires re-running the computationally intensive subgraph mining (SSM) and graph construction process for every single protein target. This could be a major practical barrier in a real-world drug discovery pipeline that needs to screen hundreds or thousands of targets.
>
> **A1**: We sincerely thank the reviewer for this important observation, and we fully agree that preprocessing scalability is a critical factor for practical deployment. **Since several multi-target scalability concerns were raised across reviewers, we provide a consolidated Global Response that offers a unified discussion and full experimental details; we refer the reviewer to it for a more complete picture.**
>
> ----
>
> > **Q3**: Can the authors provide an end-to-end average time (e.g., on the DUD-E targets) required to build the subgraph network (including subgraph mining and similarity computation)? How does this compare to the preprocessing time of baselines (like RDKit+NP)?
>
> **A3**: We thank the reviewer for the helpful question. On the ten DUD-E targets, the average time required by SubDyve to construct the subgraph network, including subgraph mining and similarity-based network construction, is 17.88 minutes (10.68 minutes for mining and 7.20 minutes for network construction). Under the same conditions, the preprocessing stage of RDKit+NP takes 9.27 minutes on average.
>
> | Model   | SubDyve |  | | RDKit-base Network|
> |----------|--------------------:|---------------------:|----------------:|-----------------:|
> | Step   | Subgraph Mining |  Network Construction | Total | Preprocessing & Network Construction |
> | Average (min) | 10.68                     | 7.20                          | **17.88**       | **9.27**                  |
>
> The higher cost of SubDyve arises from the discriminative subgraph extraction step, which is not present in baseline methods but is executed only once per target. We note that traditional fingerprint-based pipelines often require evaluating multiple fingerprint choices (e.g., MACCS, PubChem, ECFP, RDKit), which introduces additional practical overhead. In this context, constructing a subgraph-based network provides a more expressive and target-tailored representation, albeit at a higher per-target preprocessing cost.
>
> We have added these timing results and clarifications to the Appendix E. “Time Profiling of Components in SubDyve”.
>
> ----
>
> > **Q2**: The success of the entire pipeline (especially the SSM step) is highly dependent on the quality and quantity of the initial seed set . If the initial set is extremely small (e.g., < 10- 20 molecules) or not representative, the subgraph mining step will likely fail or produce misleading patterns, causing the entire downstream pipeline (GNN training and LFDR refinement) to perform poorly. The 50 seeds used as the minimum in the experiment (Sec 4.2.2) is still a somewhat optimistic setting for a "low-label" scenario.
> >**Q4**: How sensitive is the method to the size of $S_{train}$? If $S_{train}$ contained only 10 or 5 molecules, would the SSM step still be effective?
>
> **A2+A4**: We thank the reviewer for raising this important point about the dependence of our pipeline on the size of the seed set. To evaluate this, we ran an additional sensitivity study using extremely small $S_{train}$ containing only 5, 10, or 15 active compounds on PU dataset.
>
> | # Seeds | Method        | BEDROC             | EF$_1$%               |
> |:--------:|---------------|------------------|-----------------|
> | **5**   | SubDyve       | **47.98 ± 3.24**    | **56.02 ± 4.99**   |
> |         | Subgraph+NP   | 47.93 ± 5.23        | 54.6 ± 7.00          |
> |         | rdkit+NP      | 18.13 ± 3.29        | 13.85 ± 4.15          |
> | **10**  | SubDyve       | **44.88 ± 8.43**    | **54.27 ± 12.31**    |
> |         | Subgraph+NP   | 43.67 ± 5.04        | 50.74 ± 9.72        |
> |         | rdkit+NP      | 31.63 ± 3.95        | 30.14 ± 4.73          |
> | **15**  | SubDyve       | **44.24 ± 7.40**    | **57.39 ± 18.94**   |
> |         | Subgraph+NP   | 43.09 ± 5.50        | 50.86 ± 6.53        |
> |         | rdkit+NP      | 20.56 ± 3.31        | 15.07 ± 3.54          |
>
> Across all three seed sizes, SubDyve and Subraph+NP maintained strong performance relative to other methods, and the subgraph-mining step continued to extract useful patterns even with only 5-15 actives. Although performance naturally becomes noisier with very small supervision, the overall ranking quality remains stable, indicating that SSM step of SubDyve is effective in realistic low-label settings.
>
> These findings, together with the full experimental tables, have been added to the revised manuscript (Appendix F.4 and Appendix Table 20).

---

### Official Review · Reviewer_1i7n · 2025-11-01

**Soundness:** 2
**Presentation:** 2
**Contribution:** 2
**Rating:** 4
**Confidence:** 3

**Summary:**

Virtual screening (VS) is a key step in drug discovery that aims to identify potentially active compounds from large molecular libraries. In real-world scenarios, only a few active compounds are known, making it a typical low-label setting. Traditional methods often rely on generic molecular fingerprints or deep embeddings, ignoring substructure features critical for activity. Moreover, they usually treat molecules as independent samples, failing to leverage structural relationships among molecules. This paper proposes SubDyve, a subgraph-driven network propagation framework for robust virtual screening under low-label conditions, with effective control of false positives.

**Strengths:**

1. Introduce supervised subgraph mining into VS graph construction, capturing activity-relevant structural patterns and outperforming generic fingerprints.
2. Proposes an LFDR-based iterative seed updating strategy that expands the seed set while theoretically controlling the FDR upper bound, effectively reducing false positives.
3. With a few dozen seed molecules, SubDyve achieves nice early enrichment performance on both DUD-E and PU datasets, outperforming existing deep learning and fingerprint-based methods.

**Weaknesses:**

1. Subgraph mining and graph construction must be repeated per target, making cross-target scalability a potential bottleneck for large-scale multi-target screening.
2. The subgraph mining stage relies heavily on the representativeness of the initial seed set. If the seed structures are biased, the graph construction may lack generalizability.
3. All experiments are single-target independent models, with no evaluation on multi-target joint training or transfer learning scenarios.
4. Current baselines are mostly sequence or graph embedding models, with no direct comparison to recent 3D structure-aware models.

**Questions:**

Refer to weaknesses

---

> ### Author Response · Authors · 2025-11-27
> **Response to Reviewer 1i7n (Part 1/2)**
>
> We sincerely thank the reviewer for the thorough reading of our manuscript and the constructive comments provided. We greatly appreciate the time and care invested in evaluating our work, and we address each of the raised questions and suggestions in detail below.
>
> >**Q1**: Subgraph mining and graph construction must be repeated per target, making cross-target scalability a potential bottleneck for large-scale multi-target screening.
> **Q2**: All experiments are single-target independent models, with no evaluation on multi-target joint training or transfer learning scenarios.
>
> **A1+A2**: We sincerely thank the reviewer for raising these important points regarding per-target scalability and the absence of multi-target joint evaluation. SubDyve is designed as a target-specific early-recognition framework, and understanding how it behaves across multiple related targets is indeed an important aspect of its applicability. **Since several multi-target scalability concerns were raised across reviewers, we provide a consolidated Global Response that offers a unified discussion and full experimental details; we refer the reviewer to it for a more complete picture.**
>
> ----
>
> >**Q3**: The subgraph mining stage relies heavily on the representativeness of the initial seed set. If the seed structures are biased, the graph construction may lack generalizability.
>
> **A3**: We thank the reviewer for raising this important point. SubDyve indeed relies on the seed set for subgraph mining, and we agree that a biased or overly homogeneous seed distribution could reduce the generalizability of the mined subgraph patterns. To directly examine how sensitive SubDyve is to the representativeness of the seed compounds, we performed an additional experiment where we substantially relaxed the MMseqs2 homology threshold used to curate seed proteins from 0.9 to 0.5.
> Lowering the identity cutoff forces the seed compounds to be drawn from much more diverse and distant homologs, which often differ in binding-site geometry, domain composition, or regulatory context. This creates a considerably more heterogeneous and challenging seed distribution, reasonably addressing the reviewer’s concern.
>
>
> | Model|**EGFR** || **PLK1** || **SRC** ||
> |---------|----------|---------|---------|----------|---------|---------|
> | | BEDROC | EF$_1$% | BEDROC | EF$_1$% |  BEDROC | EF$_1$% |
> | SubDyve$_{0.9}$ | **86 ± 2** | **60.0 ± 2.3** |  **85 ± 3** | 51.7 ± 4.0 | **61 ± 2** | **35.0 ± 1.8** |
> | SubDyve$_{0.5}$| 79 ± 2 | 60.0 ± 2.2 | 78 ± 4 | **60.6 ± 3.6** | 41 ± 2 | 16.2 ± 1.6 |
> | PharmacMatch | 11 ± 1 | 3.1 ± 0.7 | 9 ± 2 | 1.5 ± 1.3 | 27 ± 1 | 6.0 ± 1.0 |
> | CDPKIT | 26 ± 2 | 12.2 ± 1.6 |  39 ± 3 | 5.7 ± 2.3 | 28 ± 1 | 11.1 ± 1.2 |
> | DrugCLIP | 40 ± 2 | 28.7 ± 2.1 |  66 ± 4 | 45.0 ± 4.0 |  16 ± 2 | 8.1 ± 1.3 |
> | MoLFormer | 75 ± 2 | 48.1 ± 2.8 | 69 ± 0 | 35.2 ± 4.0 |  48 ± 1 | 21.5 ± 1.5 |
> | AutoDock-Vina | 14 ± 1 | 3.68 ± 0.7 | 13 ± 1 | 1.83 ± 0.3 | 13 ± 1 | 4.00 ± 0.5 |
>
>
> Across three representative DUD-E targets (EGFR, PLK1, SRC), SubDyve’s performance naturally decreases when the seed set becomes more diverse, but the drop remains moderate, and the method still outperforms all baseline models in both BEDROC and EF$_1$%. In several cases, such as PLK1 under the 0.5 threshold, the EF$_1$% score even increases relative to the 0.9 setting. These observations indicate that SubDyve’s subgraph mining procedure is not overly reliant on narrowly tailored or highly similar seeds, and that the extracted subgraph patterns maintain transferability even when the available seed chemistry is substantially more heterogeneous.
>
> This suggests that SubDyve can tolerate variability and imperfections in the seed distribution, and remains effective even when the curated actives are not strongly representative of the target’s eventual chemical space.
>
> We have incorporated this analysis and the supporting results into the revised manuscript (Section 4.1.1 “Zero-Shot Screening on DUD-E Targets” – “Performance” (L300-308) and Appendix F.1.2).

---

> > ### Author Response · Authors · 2025-11-27
> > **Response to Reviewer 1i7n (Part 2/2)**
> >
> > > **Q4**: Current baselines are mostly sequence or graph embedding models, with no direct comparison to recent 3D structure-aware models.
> >
> > A4. We thank the reviewer for this valuable suggestion. To address it, we additionally evaluated two recent 3D structure–aware models, DiffDock-NMDN [1] and DynamicBind [2], under the same experiment setting used throughout our study. In addition, AutoDock-Vina, which was already included as a baseline in the original manuscript, is also reported. The results on representative DUD-E targets are summarized below, and the full results for all remaining targets are provided in Appendix Table 13.
> >
> > *Values: BEDROC / EF$_1$% shown.*
> > | Protein | SubDyve (Ours) | DiffDock-NMDN | DynamicBind | AutoDock-Vina  |
> > |---------|-----------------------------------:|-----------------------------------:|----------------------------------:|-----------------------------------:|
> > | **ACES** | **86 ± 2 / 57.0 ± 2.4** | 4 ± 1 / 1.0 ± 0.4 | 11 ± 1 / 3.8 ± 0.9 | 33 ± 1 / 13.9 ± 0.5 |
> > | **EGFR** | **86 ± 2 / 60.0 ± 2.3** | 5 ± 1 / 0.5 ± 0.2 | 17 ± 1 / 1.7 ± 0.6 | 14 ± 1 / 3.7 ± 0.7 |
> > | **SRC**  | **61 ± 2 / 35.0 ± 1.8** | 6 ± 1 / 0.4 ± 0.2 | 25 ± 1 / 6.0 ± 1.0 | 13 ± 1 / 4.0 ± 0.5 |
> >
> > Across all targets, DiffDock-NMDN and DynamicBind achieved very low enrichment, noticeably below AutoDock-Vina. We believe this arises largely from a mismatch between the characteristics of DUD-E and the objectives used to train modern 3D generative models.
> >
> > First, many 3D deep models are optimized for pose generation accuracy rather than for discriminating binders from decoys. They often assign high confidence to plausible poses even for non-binding decoys, which leads to poor early-recognition metrics. This disconnect between pose prediction and screening power has been noted in prior studies [3,4].
> >
> > Second, the DUD-E docked complexes contain geometric noise, and decoys are designed to mimic drug-like physicochemical properties. Deep 3D models trained on curated structural datasets (for example, PDBbind) tend to be sensitive to such noise, whereas Vina’s physics-based scoring function is more robust, which explains its relatively better performance in our results.
> >
> > These findings suggest that current 3D structure–aware generative models may require additional calibration or scoring mechanisms to be effective in ligand-ranking tasks under this experiment setting. Our evaluation of 3D methods is necessarily limited, and we consider a more comprehensive analysis of structure-aware baselines an important direction for future work.
> >
> > We have added a detailed discussion of this experiment in Appendix F.1.1.
> >
> > ----
> >
> > [1] Xia, Song, Yaowen Gu, and Yingkai Zhang. "Normalized Protein–Ligand Distance Likelihood Score for End-to-End Blind Docking and Virtual Screening." Journal of Chemical Information and Modeling 65.3 (2025): 1101-1114.
> >
> > [2] Lu, Wei, et al. "DynamicBind: predicting ligand-specific protein-ligand complex structure with a deep equivariant generative model." Nature Communications 15.1 (2024): 1071.
> >
> > [3] Li, Yue, et al. "Decoding the limits of deep learning in molecular docking for drug discovery." Chemical Science 16.37 (2025): 17374-17390.
> >
> > [4] Buttenschoen, Martin, Garrett M. Morris, and Charlotte M. Deane. "PoseBusters: AI-based docking methods fail to generate physically valid poses or generalise to novel sequences." Chemical Science 15.9 (2024): 3130-3139.

---

### Official Review · Reviewer_by4N · 2025-11-02

**Soundness:** 2
**Presentation:** 3
**Contribution:** 2
**Rating:** 4
**Confidence:** 3

**Summary:**

SubDyve is a virtual screening framework that is label efficient and builds a similarity graph from substructures that are able to discriminate classes. It applies local and dynamic seed refinement based on FDR and performs network propagation to rank candidates. A joint GNN is used to calibrate the propagation. The method tackles two major gaps in NP-based screening by replacing generic fingerprints with mined subgraph patterns. These patterns capture activity relevant signals, while limiting graph topological bias through the usage of LFDR. The proposal also employs ensembles for sake of stabilizing results.

**Strengths:**

A major contribution of SubDyve is the integration of two complementary ideas: a target-aware sub-graph fingerprint (built from class-discriminating patterns mined from scarce labels) and an LFDR-guided refinement loop that calibrates uncertainty, so that the model is able  to control mFDR and to counteract graph-induced topological bias. Such a design comes from low-label virtual screening and achieves empirical gains considering baselines. The proposal emphasizes that recognizing early, when screening value are more valuable. Further,  the ablations indicate that no component alone achieves the combined performance, demonstrating the benefit of the proposal.

**Weaknesses:**

A key limitation of SubDyve is regarding its scalability, since each target requires building a separate subgraph fingerprint network. In this case, supervised subgraph mining and graph construction becomes increasingly expensive as the number of targets or compounds grows. The iterative LFDR refinement adds further computational cost at large scale. The method also depends on the availability of homologous protein sequences to obtain annotated compounds for seed generation, which constrains its use to well-characterized targets. These factors limit scalability in large or multi-target screening, and LFDR calibration remains less reliable in intermediate decision regions. The paper should consider these limitations and discuss how they may be effectively addressed.

**Questions:**

It’s not clear how the setting qualifies as zero shot given that seeds are curated from homologous proteins with high sequence identity. Please clarify.

I believe that adding additional ablation studies would strengthen the work. Evaluating the method with and without the pretrained ChemBERTa features and comparing single versus hybrid ranking would clarify the contribution of each component and better support the
claimed synergy. Please comment.

---

> ### Author Response · Authors · 2025-11-27
> **Response to Reviewer by4N (Part 1/3)**
>
> We sincerely thank the reviewer for their careful evaluation of our work and for offering insightful and constructive feedback. Your comments have been invaluable in enhancing the quality of our manuscript. Please find our detailed responses to the raised comments and questions below.
>
> > **Q1**: A key limitation of SubDyve is regarding its scalability, since each target requires building a separate subgraph fingerprint network. In this case, supervised subgraph mining and graph construction becomes increasingly expensive as the number of targets or compounds grows. The iterative LFDR refinement adds further computational cost at large scale. The method also depends on the availability of homologous protein sequences to obtain annotated compounds for seed generation, which constrains its use to well-characterized targets. These factors limit scalability in large or multi-target screening, and LFDR calibration remains less reliable in intermediate decision regions. The paper should consider these limitations and discuss how they may be effectively addressed.
>
> **A1**: We sincerely thank the reviewer for this thoughtful and comprehensive comment. We agree that both multi-target scalability, the cost of iterative LFDR refinement, and the behavior of LFDR calibration in intermediate regions are important considerations. **Since several multi-target scalability concerns were raised across reviewers, we provide a consolidated Global Response that offers a unified discussion and full experimental details; we refer the reviewer to it for a more complete picture.**
>
>
> *[LFDR computational cost]*
>
> Regarding the computational cost of iterative LFDR refinement, we agree that repeated refinement steps introduce additional overhead compared to a single-pass propagation. In practice, however, SubDyve allows this cost to remain manageable even when the screening library becomes large. This is because the subgraph mining stage extracts highly discriminative patterns that determine which compounds meaningfully connect to the seed set. Compounds that share no relevant subgraphs with the seeds naturally receive very low or zero similarity during network construction, which limits the extent to which they participate in propagation. As a result, the refinement is effectively focused on a smaller, behaviorally meaningful neighborhood around the seed molecules rather than on the full candidate collection.
>
> This interaction between subgraph fingerprints and LFDR refinement also aligns with the ablation results in Table 3 of the manuscript: the two components reinforce each other, improving ranking performance while avoiding unnecessary propagation over compounds that are unlikely to be retrieved. These properties help keep the computational overhead of LFDR refinement modest even at larger scales, while still providing the regularization benefits that strengthen SubDyve’s ranking quality.
>
> *[LFDR stability in intermediate decision regions]*
>
> We evaluated the effect of LFDR calibration across four threshold values ($\tau$ = 0.05, 0.10, 0.30, 0.50) on the ACES target. Thresholds such as 0.30 and 0.50 correspond to mid-confidence zones where many compounds fall between clear positive and negative evidence, which aligns with the reviewer’s reference to intermediate decision regions. Across all tested thresholds, the resulting BEDROC and EF$_1$% values changed only slightly relative to the baseline configuration ($\tau$ = 0.10, 6 iterations). The largest observed deviation was about 0.4% for all three metrics. These results indicate that LFDR-based refinement remains consistent even when the threshold is shifted toward regions that place more weight on uncertain or ambiguous predictions.
>
> *Sensitivity to LFDR Threshold (iterations = 6 fixed)*
> | $\tau$    |  $\Delta$ BEDROC | $\Delta$ EF$_{1}$% |  $\Delta$ EF$_{5}$% |
> |-----:|----------:|--------:|--------:|
> | 0.05 | +0.04     | +0.40   | -0.48   |
> | 0.10 (base) | +0.00 | +0.00 | +0.00 |
> | 0.30 | −0.07     | −0.33   | −0.04   |
> | 0.50 | +0.44     | +0.27   | +0.05   |
>
> We have added a discussion on these effects of LFDR refinement to the revised manuscript in Section 4.2.3 “Stability and Calibration Behavior of LFDR Refinement” and Appendix F.5.

---

> ### Author Response · Authors · 2025-11-27
> **Response to Reviewer by4N (Part 2/3)**
>
> > **Q2**: It’s not clear how the setting qualitieis as zero shot given that seeds are curated from homologous proteins with high sequence identity. Please clarity.
>
> **A2**: We thank the reviewer for raising this important point and for the opportunity to clarify the zero-shot definition. In our initial setup, we followed common practice in recent structure-free and zero-shot virtual screening studies, where sequences sharing ≥90% identity with the evaluation targets are removed when constructing training models [1, 2]. These studies operationalize zero-shot settings by excluding highly similar proteins above this threshold, and our original 0.9 MMseqs2 cutoff was chosen to remain consistent with this established convention.
>
> Nevertheless, we agree with the reviewer that proteins at ~90% identity may still retain substantial similarity, potentially softening the strictness of the zero-shot condition. To investigate the impact of this concern more rigorously, we conducted an additional experiment in which we further relaxed the identity threshold from 0.9 to 0.5. This forces the seed compounds to be derived from much more distant PubChem proteins that may differ in binding site architecture, domain composition, or regulatory context, thus representing a more challenging and more faithful zero-shot scenario.
>
>
> | Model|**EGFR** || **PLK1** || **SRC** ||
> |---------|----------|---------|---------|----------|---------|---------|
> | | BEDROC | EF$_{1}$% | BEDROC | EF$_{1}$% |  BEDROC | EF$_{1}$% |
> | SubDyve$_{0.9}$ | **86 ± 2** | **60.0 ± 2.3** |  **85 ± 3** | 51.7 ± 4.0 | **61 ± 2** | **35.0 ± 1.8** |
> | SubDyve$_{0.5}$| 79 ± 2 | 60.0 ± 2.2 | 78 ± 4 | **60.6 ± 3.6** | 41 ± 2 | 16.2 ± 1.6 |
> | PharmacMatch | 11 ± 1 | 3.1 ± 0.7 | 9 ± 2 | 1.5 ± 1.3 | 27 ± 1 | 6.0 ± 1.0 |
> | CDPKIT | 26 ± 2 | 12.2 ± 1.6 |  39 ± 3 | 5.7 ± 2.3 | 28 ± 1 | 11.1 ± 1.2 |
> | DrugCLIP | 40 ± 2 | 28.7 ± 2.1 |  66 ± 4 | 45.0 ± 4.0 |  16 ± 2 | 8.1 ± 1.3 |
> | MoLFormer | 75 ± 2 | 48.1 ± 2.8 | 69 ± 0 | 35.2 ± 4.0 |  48 ± 1 | 21.5 ± 1.5 |
> | AutoDock-Vina | 14 ± 1 | 3.68 ± 0.7 | 13 ± 1 | 1.83 ± 0.3 | 13 ± 1 | 4.00 ± 0.5 |
>
>
> Despite the stricter homology cutoff, SubDyve still outperforms all baseline models across the three representative DUD-E targets (EGFR, PLK1, SRC). As expected, lowering the threshold reduces overall performance, since the seed compounds become less directly relevant to the target. However, the performance degradation is moderate, and SubDyve maintains clear superiority in both BEDROC and EF$_{1}$% compared to baseline models.
>
> This demonstrates that SubDyve does not rely on very close homologs and continues to operate effectively even when the available protein–ligand annotations are only distantly related to the target. In practice, this property is important because many real drug-discovery targets have limited or no high-identity homologs with known ligands.
>
> We have incorporated these new results and the accompanying discussion into the revised manuscript (Section 4.1.1 “Zero-Shot Screening on DUD-E Targets” – “Performance” (L300-L308) and Appendix F.1.2). We hope that this additional analysis helps clarify the our zero-shot formulation and demonstrates that SubDyve remains effective even when only distant homologs are available.
>
> ------
>
> [1] McNutt, Andrew T., et al. "Sprint enables interpretable and ultra-fast virtual screening against thousands of proteomes." arXiv e-prints (2024): arXiv-2411.
>
> [2] Lam, Hilbert Yuen In, et al. "Protein language models are performant in structure-free virtual screening." Briefings in Bioinformatics 25.6 (2024): bbae480.

---

> ### Author Response · Authors · 2025-11-27
> **Response to Reviewer by4N (Part 3/3)**
>
> >**Q3**: I believe that adding additional ablation studies would strengthen the work. Evaluating the method with and without the pretrained ChemBERTa features and comparing single versus hybrid ranking would clarify the contribution of each component and better support the claimed synergy. Please comment.
>
> **A3**: We thank the reviewer for the helpful suggestion. In addition to the components highlighted by the reviewer, we conducted a broader set of ablation studies to more comprehensively assess how each feature of the GNN model contributes to the final performance. This includes removing pretrained ChemBERTa features, dividing the hybrid PCA–NP ranking module into a single ranking module, and further examining the effect of individual elements within seed weight, NP score, and so on.
>
> As shown in the table below, removing ChemBERTa features or disabling the hybrid ranking module leads to clear decreases in both BEDROC and EF$_{1}$%. Ablations that rely solely on PCA, NP score, or Seed Weight also show consistent drops. These results indicate that each component plays a meaningful role, and the full configuration benefits from the combined effect of multiple complementary signals. The overall trend supports the synergy between ChemBERTa embeddings and hybrid ranking, while also showing that other modules contribute to the SubDyve’s retrieval performance.
>
> | Setting              |     BEDROC      |     EF$_{1}$%       |
> |----------------------|-----------------:|---------------:|
> | **Full**             | **83.44 ± 1.44** | **97.59 ± 1.44** |
> | – Hybrid             |     81.73 ± 3.10 |     91.64 ± 3.65 |
> | – Single (PCA only)  |     82.25 ± 2.92 |     96.08 ± 2.61 |
> | – Single (NP only)   |     80.16 ± 3.88 |     91.14 ± 4.06 |
> | – FP                 |     81.87 ± 2.94 |     94.35 ± 1.93 |
> | – PCA                |     80.28 ± 2.84 |     93.93 ± 2.13 |
> | – Seed Weight        |     79.40 ± 3.64 |     89.90 ± 3.48 |
> | – NP Score           |     79.84 ± 3.51 |     90.63 ± 4.04 |
> | – ChemBERTa          |     79.82 ± 4.40 |     88.92 ± 5.02 |
>
> We have added a discussion of these findings in Appendix F.8 and highlighted importance of feature configuration as a potential avenue for further exploration.

---

### Author Response · Authors · 2025-11-27
**Global Response for Target-Specific Approach of SubDyve**

We thank the reviewers for raising the related concerns regarding per-target scalability, cross-target screening, and the lack of multi-target evaluation. We respond to these points together, as they address the same underlying aspect of the method.

----

SubDyve is designed as a target-specific early-recognition model, which aligns with standard virtual screening workflows where models are typically built for a single disease-related protein. Nevertheless, we fully acknowledge that repeated subgraph mining and graph construction could become costly when many targets are considered, and that evaluating SubDyve beyond the single-target setting is valuable for understanding both scalability and cross-target generalization.


**[Multi-target Scalability Evaluation]**

To investigate this, we conducted an additional experiment on three DUD-E kinases (EGFR, KIT, and SRC), which were selected because they are tyrosine kinases with related molecular functions and overlapping signaling roles [1, 2, 3]. We identified the homologous proteins shared across these targets using PubChem and collected the active and inactive compounds annotated to these shared homologs. Subgraph patterns were extracted from this joint set. Using these mined patterns, we selected compounds that were active across two or more of the targets, together with decoys shared within the same target subsets, and constructed a unified subgraph-based fingerprint network. This network was built once and reused for all evaluations.

We then applied SubDyve’s network propagation with LFDR refinement under four cross-target activity definitions (three-target actives and each target-pair combination). Interestingly, SubDyve maintained strong early-recognition performance in all settings and consistently ranked cross-target actives higher than DrugCLIP. These results indicate that a single subgraph mining step can support multiple biologically related targets when they share homologous ligand annotations.

| Combination      | SubDyve /  BEDROC | EF$_1$% | EF$_5$% | EF$_{10}$% | DrugCLIP  / BEDROC | EF$_1$% | EF$_5$% | EF$_{10}$% |
|------------------|----------------:|-------------:|-------------:|--------------:|-----------------:|--------------:|--------------:|----------------:|
| **EGFR-KIT-SRC** | 69.52           | 35.93        | 19.38        | 9.69          | 4.55             | 0.00          | 0.00          | 2.22            |
| **EGFR-KIT**     | 79.37           | 57.27        | 18.18        | 9.09          | 5.19             | 0.00          | 0.00          | 1.66            |
| **EGFR-SRC**     | 84.11           | 41.21        | 19.20        | 9.99          | 13.14            | 7.26          | 2.95          | 1.66            |
| **KIT-SRC**      | 55.76           | 20.25        | 14.40        | 8.81          | 1.66             | 0.00          | 0.00          | 0.33            |



**[Interpretation and Scope]**

This experiment does not fully resolve scaling challenges for very large or heterogeneous target panels, nor does it constitute multi-target joint training. However, it demonstrates that the structural patterns learned by SubDyve are transferable across biologically related targets and that preprocessing cost can be amortized when targets share homologous ligand annotations. The modeling objective of SubDyve remains target-specific early recognition, and extending the framework toward broader multi-target or transfer-learning settings represents a promising direction for future work.

These findings are discussed in Section 4.1.1 “Zero-Shot Screening on DUD-E Targets” – “Multi-Target Screening Analysis” (L309-323), with additional details provided in Append F.1.3.

----

[1] Tomuleasa, Ciprian, et al. "Therapeutic advances of targeting receptor tyrosine kinases in cancer." Signal transduction and targeted therapy 9.1 (2024): 201.

[2] Tatton, Louise, et al. "The Src-selective kinase inhibitor PP1 also inhibits Kit and Bcr-Abl tyrosine kinases." Journal of Biological Chemistry 278.7 (2003): 4847-4853.

[3] Rao, Suman, et al. "Target modulation by a kinase inhibitor engineered to induce a tandem blockade of the epidermal growth factor receptor (EGFR) and c-Src: the concept of type III combi-targeting." PloS one 10.2 (2015): e0117215.

---

### Author Response · Authors · 2025-12-02
**Summary of the Authors' Response**

We sincerely thank the reviewers for their careful evaluations and constructive feedback, which helped us substantially refine and clarify our work. Below, we summarize the major revisions and additional analyses incorporated in response to the reviewer's comments.

---


**1. Multi-Target Scalability and Cross-Target Screening**

- Conducted a multi-target evaluation across EGFR, KIT, and SRC using a single jointly mined subgraph set derived from shared homologous ligand annotations.

- Observed consistently strong early-recognition performance exceeding DrugCLIP, indicating that preprocessing can be amortized across biologically related targets (Section 4.1.1; Appendix F.1.3).

**2. Computational Cost and LFDR Refinement Behavior**

- Discussed that LFDR refinement operates on a similarity graph already pruned by discriminative substructures, ensuring propagation remains localized to relevant molecular neighborhoods.

- Conducted a systematic sensitivity analysis over LFDR thresholds and iteration counts, demonstrating the robustness of LFDR refinement (Section 4.2.3; Appendix F.5).

**3. Runtime Profiling and Preprocessing Overhead**

- Subgraph mining (10.68 min) and network construction (7.20 min) in SubDyve preprocessing amount to 17.88 minutes per DUD-E target, exceeding the 9.27 minutes preprocessing time of the RDKit-based baseline under identical conditions.

- Noted that traditional screening pipelines often require testing multiple fingerprints to identify an appropriate representation, this one-time preprocessing cost may be a reasonable overhead (Appendix E).


**4. Zero-Shot Definition and Dependence on Homologous Seeds**.

- Applied a more stringent MMseqs2 sequence-identity filter (0.9 → 0.5), resulting in the use of more distant and structurally diverse homologs.

- Despite the more challenging setting, SubDyve outperformed all baseline models across EGFR, PLK1, and SRC in both BEDROC and EF$_1$% (Section 4.1.1; Appendix F.1.2).


**5. Sensitivity to Seed Representativeness and Seed Set Size**

- Lowered the sequence-identity threshold from 0.9 to 0.5, increasing seed heterogeneity while maintaining stable retrieval performance.

- Additional evaluations with 5, 10, and 15 seed molecules show that SubDyve and Subgraph+NP continue to yield strong early-recognition performance relative to RDKit/MACCS methods, demonstrating the effectiveness of subgraph-based approach (Appendix F.4; Appendix Table 20).


**6. Component Ablations of GNN model features: ChemBERTa Features and Hybrid Ranking**

- Conducted an extensive ablation suite evaluating the contributions of ChemBERTa embeddings, hybrid PCA–NP ranking, and individual scoring components.

- Demonstrated that each component provides complementary information, and that the full configuration yields the strongest enrichment (Appendix F.8).


**7. Comparison with 3D Structure-Aware Baselines**

- Added DiffDock-NMDN and DynamicBind as 3D structure-aware baselines alongside AutoDock-Vina, and found that SubDyve substantially outperformed these methods across ten DUD-E targets, with 3D generative models showing notably weak early-recognition.

- Discussed likely causes, including pose-generation objectives misaligned with binder discrimination and sensitivity to DUD-E docking noise (Appendix F.1.1).


**8. Interpretability and Pharmacophoric Relevance of Mined Subgraphs**

- Analyzed CDK7 PU dataset and found that the mined subgraphs capture aromatic and heteroaromatic hinge-binding motifs as well as hydrophobic and amine-linked substituent patterns characteristic of CDK-family inhibitors.

- RDKit pharmacophore-feature mapping indicates that many of these patterns include aromatic or hydrogen-bonding elements relevant to CDK7’s binding preferences (Appendix G.4).


**9. Applicability to Structure-Based and Protein–Ligand Joint Settings**

- Discussed that SubDyve’s graph-based framework can incorporate structural or protein-informed descriptors without modifying LFDR refinement.

- Noted that patterns learned in ligand-only settings already reflect target-associated interaction tendencies, suggesting compatibility with future structure-aware extensions (Section 5).


**10. Code Availability and Reproducibility**

- Resolved the repository access issue and ensured stable availability for the review period.

----

Please find the updated experiments and major revisions marked in blue within the revised manuscript.

We hope that these additional experiments, together with our detailed point-by-point responses, adequately address all of the reviewers’ concerns. If there are any remaining questions or aspects that require further clarification, we would be glad to provide additional explanation.

Thank you again for your valuable time and thoughtful feedback in reviewing our work.

---

### Meta-Review · Area_Chair_NZkg · 2025-12-23

**Summary:**

Reviewers agree that the problem setting is highly relevant, as real-world drug discovery often operates with very limited numbers of known bioactive compounds, placing high value on early enrichment. The use of class-discriminative subgraph fingerprints was regarded as a solid approach, allowing to capture activity-relevant structural motifs that are missed by generic fingerprints or embeddings.
The LFDR-based iterative seed refinement was alse considered effective (and novel) to mitigate topological bias and control false discovery during network propagation.
Reviewers also appreciated the empirical analysis, ablation studies, sensitivity analyses, and interpretability-oriented case studies, which help justify the contribution of each component.

The most prominent weakness highlighted by all reviewers is scalability: subgraph mining and graph construction is costly and must be repeated per target, making the approach potentially expensive for large multi-target screening campaigns.
The second most relevant concern regarded seed generation and the potential reliance/sensitivity of the methodo to the availability of good quality/high-homology seeds.
An additional issue regarded the lack of comparison with recent works proposing 3D structure-ware models.

All three major issues were concretely addressed by the authors rebuttal, providing convincing evidence of the robustness of the approach through dedicated additional experiments.

As an additional reflection, the contribution of the paper seems to be a bit borderline with respect to the scope of ICLR, as the gist of the methodological contribution is not exactly on the representation learning aspect. Nonetheless, this was not perceived as an issue by the reviewers.

**Reviewer Concerns:**

All critical concerns have been addressed in the rebuttal, backed up by additional experiments.

Concerns about scalability and preprocessing overhead were directly tackled by reporting detailed end-to-end timing for subgraph mining and network construction, and by contextualizing this cost relative to practical fingerprint-selection overhead in standard VS pipelines.

Dependence on seed size and representativeness was examined via new experiments with as few as 5–15 actives, and reducing homology down to 50%, showing that SubDyve remains competitive also under such challenging conditions.

Lack of 3D structure-aware baselines was addressed by adding DiffDock-NMDN, DynamicBind, and AutoDock-Vina comparisons, with clear discussion of why such models underperform on early-recognition benchmarks (though performance of these models appear to be really and unexpectedly low).

Requests for additional ablations were thoroughly addressed, including removing ChemBERTa features, disabling hybrid ranking, and isolating individual scoring components, which substantiated the claimed synergy.

**Reviewer Scores:**

None of the reviewers responded to the rebuttal before suspension.

However, the two most negative reviewers (by4N, 1i7n) had soft concerns which rather appeared as a request for further clarifications than substantial issues preventing the paper from being accepted. Given that the core aspects of these concerns have been addressed by the rebuttal, I think these reviewers would have raised their score from 4 to at least 6.

Reviewer iVxC had similar concerns as by4N, 1i7n but with an higher score (6) and confidence. I expect them to at least maintain their score.

Reviewer yH49 was positive (8) with no substantial concerns and limited confidence (2), so I guess they would have sticked to a score of 8.

Overall, the score considerations and the review/rebuttal comments project the paper to a final average score >6, and above the acceptance threshold in my opinion.

---

### Decision · Program_Chairs · 2026-01-26

Accept (Poster)